# RL4CO: an Extensive Reinforcement Learning for Combinatorial Optimization Benchmark

**Federico Berto**[*1], **Chuanbo Hua**[*1], **Junyoung Park**[*1], **Laurin Luttmann**[*2],
**Yining Ma**[3], **Fanchen Bu**[1], **Jiarui Wang**[4], **Haoran Ye**[5], **Minsu Kim**[1],
**Sanghyeok Choi**[1], **Nayeli Gast Zepeda**[6], **André Hottung**[6], **Jianan Zhou**[3], **Jieyi Bi**[3],
**Yu Hu**[7], **Fei Liu**[8], **Hyeonah Kim**[1], **Jiwoo Son**[15], **Haeyeon Kim**[1],
**Davide Angioni**[9], **Wouter Kool**[10], **Zhiguang Cao**[11], **Qingfu Zhang**[8], **Joungho Kim**[1],
**Jie Zhang**[3], **Kijung Shin**[1], **Cathy Wu**[12], **Sungsoo Ahn**[13], **Guojie Song**[5]
**Changhyun Kwon**[1,14], **Kevin Tierney**[6], **Lin Xie**[2,14], **Jinkyoo Park**[1,15]

[1]KAIST, [2]Leuphana University, [3]Nanyang Technological University,
[4]Southeastern University, [5]Peking University, [6]Bielefeld University,
[7]Soochow University, [8]City University of Hong Kong, [9]University of Brescia,
[10]ORTEC, [11]Singapore Management University, [12]MIT, [13]POSTECH,
[14]Twente University, [15]OMELET, AI4CO⬡

## Abstract

Deep reinforcement learning (RL) has recently shown significant benefits in solving combinatorial optimization (CO) problems, reducing reliance on domain expertise, and improving computational efficiency. However, the field lacks a unified benchmark for easy development and standardized comparison of algorithms across diverse CO problems. To fill this gap, we introduce RL4CO, a unified and extensive benchmark with in-depth library coverage of 23 state-of-the-art methods and more than 20 CO problems. Built on efficient software libraries and best practices in implementation, RL4CO features modularized implementation and flexible configuration of diverse RL algorithms, neural network architectures, inference techniques, and environments. RL4CO allows researchers to seamlessly navigate existing successes and develop their unique designs, facilitating the entire research process by decoupling science from heavy engineering. We also provide extensive benchmark studies to inspire new insights and future work. RL4CO has attracted numerous researchers in the community and is open-sourced at https://github.com/ai4co/rl4co.

## 1 Introduction

Combinatorial optimization (CO) focuses on finding optimal solutions for problems with discrete variables and has broad applications, including vehicle routing [89, 60], scheduling [128], and hardware device placement [53]. Given that the combinatorial space expands exponentially and exhibits NP-hard characteristics, the operations research (OR) community has traditionally tackled these challenges through the development of mathematical programming algorithms [35] and handcrafted heuristics [27]. Despite their success, these methods still face significant limitations: mathematical programming struggles with scaling, while handcrafted heuristics require significant domain-specific adjustments for different CO problems.

---

[*]Equal contribution.

⬡Work made with contributions from the AI4CO open research community.

Submitted to the 38th Conference on Neural Information Processing Systems (NeurIPS 2024) Track on Datasets and Benchmarks. Do not distribute.

Recently, to address these limitations, neural combinatorial optimization (NCO) [7] has emerged. It employs deep neural networks to automate the problem-solving process and significantly reduces the computation demands and the need for domain expertise. Recent NCO works mainly leverage the reinforcement learning (RL) paradigm, making significant strides in improving exploration efficiency [62, 54], relaxing the needs of obtaining optimal solutions, and extending to various CO tasks [128, 89, 60, 53]. Although supervised learning (SL) methods [29] are shown to be effective in NCO, they require the availability of high-quality solutions, which is unrealistic for large instances or theoretically hard problems. Therefore, we focus on the widespread RL paradigm in this paper.

Despite the growing popularity and advancements in using reinforcement learning for solving combinatorial optimization, there remains a lack of a unified benchmark for analyzing past works under consistent implementations and conditions. The absence of a standardized benchmark hinders NCO researchers' efforts to make impactful advancements and leverage existing successes, as it becomes challenging to determine the superiority of one method over another. Moreover, the significance of NCO lies in its potential for generalizability across multiple problems without extensive problem-specific knowledge. Variations in implementation can make it difficult for new researchers to engage with the NCO community, and inconsistent comparisons obstruct straightforward performance evaluations. These issues pose significant challenges and underscore the need for a comprehensive benchmark to streamline research and foster consistent progress.

**Contributions.** To bridge this gap, we introduce RL4CO, the first comprehensive benchmark with multiple baselines, environments, and boilerplate from the literature, all implemented in a *modular*, *flexible*, *accelerated*, and *unified* manner. Our aim is to facilitate the entire research process for the NCO community with the following key contributions: 1) **Simplifying development** through modularizing 27 environments and 23 existing baseline models, allowing for flexible and automated combinations for effortless testing, switching, and achieving state-of-the-art performance; 2) **Enhancing the training and testing efficiency** through the customized unified pipeline tailored for the NCO community based on advanced libraries such as TorchRL [15], PyTorch Lightning [31], Hydra [123], and TensorDict [15]; 3) **Standardizing evaluation** to ensure fair and comprehensive comparisons, enabling researchers to automatically test a broader range of problems from diverse distributions and gather valuable insights using our testbed. Overall, RL4CO eliminates the need for repetitive heavy engineering in the NCO community and fosters seamless future development by building on existing successes, enabling advanced innovation and progress in the field.

## 2  Related Works

**Neural Combinatorial Optimization.** Neural combinatorial optimization (NCO) utilizes machine learning techniques to automatically develop novel heuristics for solving NP-hard CO problems. We classify the majority of NCO research from the following perspectives: 1) *Learning Paragiams*: researchers have employed supervised learning [115, 108, 29, 75] to approximate optimal solutions to CO instances. Further research leverages reinforcement learning [6, 89, 60, 62], and unsupervised learning [39, 84] to ease the difficulty of obtaining (near-)optimal solutions. 2) *Models*: various deep learning architectures such as recurrent neural networks [115, 22, 68], graph neural networks [48, 84], Transformers [60, 62], diffusion models [108], and GFlowNets [129, 56] have been employed. 3) *Problems*: NCO has demonstrated great success in various problems, including vehicle routing problems (VRPs) (e.g., traveling salesman problem and capacitated VRP), scheduling problems (e.g., job shop scheduling problems [128]), hardware device placement [53], and graph-based CO problems (e.g., maximum independent set [23, 2] and maximum cut [129]). 4) *Heuristic Types*: generally, the learned heuristics can be categorized as *constructive* in an autoregressive [60] or non-autoregressive [48] way, and *improvement* heuristics, which leverage traditional heuristics [120, 80] and meta-heuristics [105]. We refer to Bengio et al. [7] for a comprehensive survey. In this paper, we focus on the reinforcement learning paradigm due to its effectiveness and flexibility. Notably, the proposed RL4CO is versatile to support most combinations of models, problems and heuristic types, making it an apt library and benchmark for future research in NCO.

Table 1: Comparison of libraries in reinforcement learning for combinatorial optimization.

| Library | Environments # | Baselines[†] # | Hardware Acceleration | Availability | Modular Baselines | Open Community |
|---|---|---|---|---|---|---|
| ORL [4] | 3 | 1 | × | × | × | × |
| OR-Gym [42] | 9 | - | × | ✓ | × | × |
| Graph-Env [12] | 2 | - | × | ✓ | × | × |
| RLOR [116] | 2 | 2 | × | ✓ | ✓ | × |
| RoutingArena [111] | 1 | 8 | ✓ | × | × | × |
| Jumanji [14] | 22 | 3 | ✓ | ✓ | × | × |
| RL4CO (ours) | 27[‡] | 23 | ✓ | ✓ | ✓ | ✓ |

[†] We consider as *baselines* ad-hoc network architectures (i.e., policies) and RL algorithms from the literature.
[‡] We also consider the possible 16 combinations of environments generated by the unified Multi-Task VRP, as they have been historically considered separate environments in the NCO literature.

**Related Benchmark Libraries.** Despite the variety of general-purpose RL software libraries [18, 70, 96, 119, 24, 33, 81], there is a lack of a unified and extensive benchmark for CO problems. Balaji et al. [4] propose an RL benchmark for Operations Research (OR) with a PPO baseline [100]; Hubbs et al. [42], Biagioni et al. [12] provide a collection of OR environments. Wan et al. [116] propose a general-purpose library for OR, and benchmarks the canonical TSP and CVRP environments. However, a major downside of the above libraries is that they cannot be massively parallelized due to their reliance on the OpenAI Gym API, which can only run on CPU, unlike RL4CO, which is based on the TorchRL [15], a recent official PyTorch [92] library for RL that enables hardware-accelerated execution of both environments and algorithms. Prouvost et al. [94] introduces a library specialized for CO problems that work in combination with traditional MILP [71] solvers. We also mention Routing Arena [111], whose scope is different from RL4CO, namely, comparing NCO and classical solvers only for the CVRP. The most related work is Jumanji [14], which provides a variety of CO environments written in JAX [16] that can be hardware-accelerated alongside an actor-critic baseline. While Jumanji is an RL environment suite, RL4CO is a full-stack library that integrates environments, policies, RL algorithms under a unified framework.

## 3 RL4CO: Taxonomy

We describe the RL4CO taxonomy, categorizing components into *Environments, Policies,* and *RL Algorithms*. Then we translate the taxonomy to implementation in § 4.

**Environments.** Given a CO problem instance $x$, we formulate the solution-generating procedure as a Markov Decision Process (MDP) characterized by a tuple $(\mathcal{S}, \mathcal{A}, \mathcal{T}, \mathcal{R}, \gamma)$ as follows. **State $\mathcal{S}$** is the space of states that represent the given problem $x$ and the current partial solution being updated in the MDP. **Action $\mathcal{A}$** is the action space, which includes all feasible actions $a_t$ that can be taken at each step $t$. **State Transition $\mathcal{T}$** is the deterministic state transition function $s_{t+1} = \mathcal{T}(s_t, a_t)$ that updates a state $s_t$ to the next state $s_{t+1}$. **Reward $\mathcal{R}$** is the reward function $\mathcal{R}(s_t, a_t)$ representing the immediate reward received after taking action $a_t$ in state $s_t$. Finally, $\gamma \in [0,1]$ is a discount factor that determines the importance of future rewards. Since the state transition is deterministic, we represent the solution for a problem $x$ as a sequence of $T$ actions $\boldsymbol{a} = (a_1, \ldots, a_T)$. Then the total return $\sum_{t=1}^{T} \mathcal{R}(s_t, a_t)$ translates to the negative cost function of the CO problem.

**Policies.** The policies can be categorized into constructive policies, which generate a solution from scratch, and improvement policies, which refine an existing solution.

*Constructive policies.* A policy $\pi$ is used to construct a solution from scratch for a given problem instance $x$. It can be further categorized into autoregressive (AR) and non-autoregressive (NAR) policies. An AR policy is composed by an encoder $f$ that maps the instance $x$ into an embedding space $\boldsymbol{h} = f(x)$ and by a decoder $g$ that iteratively determines a sequence of actions $\boldsymbol{a}$ as follows:

$$a_t \sim g(a_t|a_{t-1}, ..., a_0, s_t, \boldsymbol{h}), \quad \pi(\boldsymbol{a}|x) \triangleq \prod_{t=1}^{T-1} g(a_t|a_{t-1}, \ldots, a_0, s_t, \boldsymbol{h}). \tag{1}$$

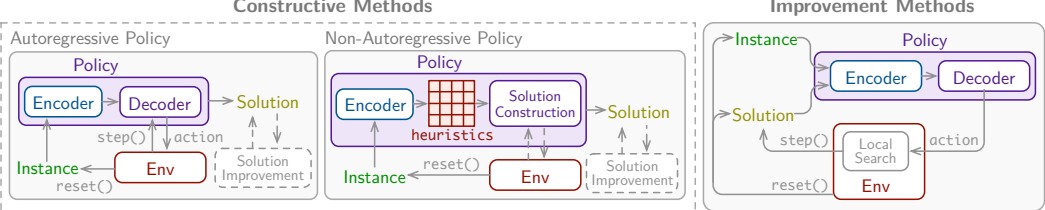

Figure 1: Overview of different types of policies and their modularization in RL4CO.

A NAR policy encodes a problem $\boldsymbol{x}$ into a heuristic $\mathcal{H} = f(\boldsymbol{x}) \in \mathbb{R}_+^N$, where $N$ is the number of possible assignments across all decision variables. Each number in $\mathcal{H}$ represents a (unnormalized) probability of a particular assignment. To obtain a solution $\boldsymbol{a}$ from $\mathcal{H}$, one can sample a sequence of assignments from $\mathcal{H}$ while dynamically masking infeasible assignments to meet problem-specific constraints. It can also guide a search process, e.g., Ant Colony Optimization [28, 125, 56], or be incorporated into hybrid frameworks [127]. Here, the heuristic helps identify promising transitions and improve the efficiency of finding an optimal or near-optimal solution.

*Improvement policies.* A policy can be used for improving an initial solution $\boldsymbol{a}^0 = (a_0^0, \ldots, a_{T-1}^0)$ into another one potentially with higher quality, which can be formulated as follows:

$$\boldsymbol{a}^k \sim g(\boldsymbol{a}^0, \boldsymbol{h}), \quad \pi(\boldsymbol{a}^K | \boldsymbol{a}^0, \boldsymbol{x}) \triangleq \prod_{k=1}^{K-1} g(\boldsymbol{a}^k | \boldsymbol{a}^{k-1}, ..., \boldsymbol{a}^0, \boldsymbol{h}), \tag{2}$$

where $\boldsymbol{a}^k$ is the $k$-th updated solution and $K$ is the budget for number of improvements. This process allows continuous refinement for a long time to enhance the solution quality.

**RL Algorithms.** The RL objective is to learn a policy $\pi$ that maximizes the expected cumulative reward (or equivalently minimizes the cost) over the distribution of problem instances:

$$\theta^* = \underset{\theta}{\mathrm{argmax}} \, \mathbb{E}_{\boldsymbol{x} \sim P(\boldsymbol{x})} \left[ \mathbb{E}_{\pi(\boldsymbol{a}|\boldsymbol{x})} \left[ \sum_{t=0}^{T-1} \gamma^t \mathcal{R}(s_t, a_t) \right] \right], \tag{3}$$

where $\theta$ is the set of parameters of $\pi$ and $P(\boldsymbol{x})$ is the distribution of problem instances. Eq. (3) can be solved using algorithms such as variations of REINFORCE [109], Advantage Actor-Critic (A2C) methods [59], or Proximal Policy Optimization (PPO) [100]. These algorithms are employed to train the policy network $\pi$, by transforming the maximization problem in Eq. (3) into a minimization problem involving a loss function, which is then optimized using gradient descent algorithms. For instance, the REINFORCE loss function gradient is given by:

$$\nabla_\theta \mathcal{L}_a(\theta | \boldsymbol{x}) = \mathbb{E}_{\pi(\boldsymbol{a}|\boldsymbol{x})} \left[ (R(\boldsymbol{a}, \boldsymbol{x}) - b(\boldsymbol{x})) \nabla_\theta \log \pi(\boldsymbol{a}|\boldsymbol{x}) \right], \tag{4}$$

where $b(\cdot)$ is a baseline function used to stabilize training and reduce gradient variance. We also distinguish between two types of RL (pre)training: 1) *inductive* and 2) *transductive* RL. In inductive RL, the focus is on learning patterns from the training dataset to generalize to new instances, thus amortizing the inference procedure. Conversely, transductive RL (or test-time optimization) optimizes parameters during testing on target instances. Typically, a policy $\pi$ is trained using inductive RL, followed by transductive RL for test-time optimization.

## 4 RL4CO: Library Structure

RL4CO is a unified reinforcement learning (RL) for Combinatorial Optimization (CO) library that aims to provide a *modular*, *flexible*, and *unified* code base for training and evaluating RL for CO methods with extensive benchmarking capabilities on various settings. As shown in Fig. 2, RL4CO decouples the major components of an RL pipeline, prioritizing their reusability in the implementation. Following also the taxonomy of § 3, the main components are: (§ 4.1) Environments, (§ 4.2) Policies, (§ 4.3) RL algorithms, (§ 4.4) Utilities, and (§ 4.5) Environments & Baselines Zoo.

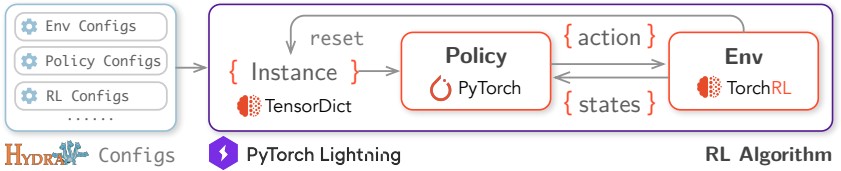

Figure 2: Overview of the RL4CO pipeline: from configurations to training a policy.

## 4.1 Environments

Environments in RL4CO fully specify the CO problems and their logic. They are based on the `RL4COEnvBase` class that extends from the `EnvBase` in TorchRL [15]. A modular `generator` can be provided to the environment. The generator provides CO instances to the environment, and different generators can be used to generate different data distributions. Static instance data and dynamic variables, such as the current state $s_t$, current solution $a^k$ for improvement environments, policy actions $a_t$, rewards, and additional information are passed in a *stateless* fashion in a `TensorDict` [86], that we call td, through the environment `reset` and `step` functions. Additionally, our environment API contains several functions, such as `render`, `check_solution_validity`, `select_start_nodes` (i.e., for POMO-based optimization [62]) and optional API as `local_search` solution improvement.

It is noteworthy that RL4CO enhances the efficiency of environments when compared to vanilla TorchRL, by overriding and optimizing some methods in TorchRL `EnvBase`. For instance, our new `step` method brings a decrease of up to 50% in latency and halves the memory impact by avoiding saving duplicate components in the stateless `TensorDict`.

## 4.2 Policies

Policies in RL4CO are subclasses of PyTorch's `nn.Module` and contain the encoding-decoding logic and neural network parameters $\theta$. Different policies in the RL4CO "zoo" can inherit from metaclasses like `ConstructivePolicy` or `ImprovementPolicy`. We modularize components to process raw features into the embedding space via a parametrized function $\phi_\omega$, called *feature embeddings*. 1) *Node Embeddings* $\phi_n$: transform $m_n$ node features of instances $x$ from the feature space to the embedding space $h$, i.e., $[B, N, m_n] \rightarrow [B, N, h]$. 2) *Edge Embeddings* $\phi_e$: transform $m_e$ edge features of instances $x$ from the feature space to the embedding space $h$, i.e., $[B, E, m_e] \rightarrow [B, E, h]$, where $E$ is the number of edges. 3) *Context Embeddings* $\phi_c$: capture contextual information by transforming $m_c$ context features from the current decoding step $s_t$ from the feature space to the embedding space $h$, i.e., $[B, m_c] \rightarrow [B, h]$, for nodes or edges. Overall, Fig. 3 illustrates a generic constructive AR policy in RL4CO, where the feature embeddings are applied similarly to other types of policies. Embeddings can be automatically selected by RL4CO at runtime by simply passing the `env_name` to the policy. Additionally, we allow for granular control of any higher-level policy component independently, such as encoders and decoders.

## 4.3 RL Algorithms

RL algorithms in RL4CO define the process that takes the `Environment` with its problem instances and the `Policy` to optimize its parameters $\theta$. The parent class of algorithms is the `RL4COLitModule`, inheriting from PyTorch Lightning's `pl.LightningModule` [31]. This allows for granular support of various methods including the `[train, val, test]_step`, automatic logging with several logging services such as Wandb via `log_metrics`, automatic optimizer configuration via `configure_optimizers` and several useful callbacks for RL methods such as `on_train_epoch_end`. RL algorithms are additionally attached to an `RL4COTrainer`, a wrapper we made with additional optimizations around `pl.Trainer`. This module seamlessly supports features of modern training pipelines, including logging, checkpoint management, mixed-precision training, various hardware acceleration supports (e.g., CPU, GPU, TPU, and Apple Silicon), and multi-device hardware accelerator in distributed settings [69]. For instance, using mixed-precision

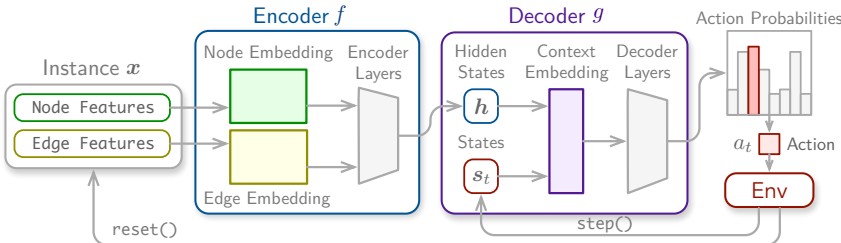

Figure 3: Overview of modularized RL4CO policies. Any component such as the encoder/decoder structure and feature embeddings can be replaced and thus the model is adaptable to various new environments.

training significantly decreases training time without sacrificing much convergence and enables us to leverage recent routines, e.g., FlashAttention [26, 25], which we investigate in Appendix.

## 4.4 Utilities

**Configuration Management.** Optionally, but usefully, we adopt Hydra [123], an open-source Python framework that enables hierarchical config management, making it easier to manage complex configurations and experiments with different settings as shown in Appendix. Hydra additionally allows for automatically parsing parameters (un-)defined in configs - i.e., `python run.py experiment=routing/pomo env=cvrp env.generator_params.num_loc=50` launches an experiment defined under `routing/pomo` and changes the environment to CVRP with 50 locations.

**Decoding Schemes.** Decoding schemes handle the logic of model logits $z$ by applying preprocessing, such as masking of infeasible actions and/or additional techniques to select better actions during training and testing. We implement the model and problem-agnostic decoding schemes under the `DecodingStrategy` class in the RL4CO codebase that can be easily reused: 1) *Greedy*, which selects the action with the highest probability; 2) *Sampling*, which samples `n_samples` solutions from the current masked probability distribution of the policy, incorporating sampling strategies like 2.a) Softmax Temperature $\tau$, 2.b) top-k sampling [61], and 2.c) top-p (or Nucleus) sampling [38] (more details in Appendix); 3) *Multistart*, which enforces diverse starting actions as demonstrated in POMO [62], such as starting from different cities in the Traveling Salesman Problem (TSP) with N nodes; 4) *Augmentation*, which applies transformations to instances, such as random rotations and flipping in Euclidean problems [55], to create an augmented set of problems.

**Documentation, Tutorials, and Testing.** We release extensive documentation to make it as accessible as possible for both newcomers and experts. RL4CO can be easily installed by running `pip install rl4co` with open-source code available at https://github.com/ai4co/rl4co. Several tutorials and examples are also available under the `examples/` folder. We thoroughly test our library via continuous integration on multiple Python versions and operating systems. The following code snippet shows minimalistic code that can train a model in a few lines:

```python
from rl4co.envs.routing import TSPEnv, TSPGenerator
from rl4co.models import AttentionModelPolicy, POMO
from rl4co.utils import RL4COTrainer
# Instantiate generator and environment
generator = TSPGenerator(num_loc=50, loc_distribution="uniform")
env = TSPEnv(generator)
# Create policy and RL model
policy = AttentionModelPolicy(env_name=env.name, num_encoder_layers=6)
model = POMO(env, policy, batch_size=64)
# Instantiate Trainer and fit
trainer = RL4COTrainer(max_epochs=10, accelerator="gpu", precision="16-mixed")
trainer.fit(model)
```

### 4.5 Environments & Baselines Zoo

**Environments.** We include benchmarking from the following environments, divided into four areas. 1) **Routing**: Traveling Salesman Problem (TSP) [65], Capacitated Vehicle Routing Problem (CVRP) [13], Orienteering Problem (OP) [64, 21], Prize Collecting TSP (PCTSP) [5], Pickup and Delivery Problem (PDP) [50, 99] and Multi-Task VRP (MTVRP) [72, 131, 9] (which modularizes with 16 problem variants including the basic VRPTW, OVRP, VRPB, VRPL and VRPs with their constraint combinations); 2) **Scheduling**: Flexible Job Shop Scheduling Problem (FJSSP) [17], Job Shop Scheduling Problem (JSSP) [97] and Flow Shop Scheduling Problem (FJSP); 3) **Electronic Design Automation**: multiple Decap Placement Problem (mDPP) [53]; 4) **Graph**: Facility Location Problem (FLP) [30] and Max Cover Problem (MCP) [51].

**Baseline Zoo.** Given that several works contribute to both new policies and new RL algorithm variations, we list the papers we reproduce. For 1) **Constructive AR** methods, we include the Attention Model (AM) [60], Ptr-Net [115], POMO [62], MatNet [63], HAM [67], SymNCO [55], PolyNet [41], MTPOMO [72], MVMoE [131], L2D [128], HGNN [106] and DevFormer [53]. For 2) **Constructive NAR** methods, we benchmark Ant Colony Optimization-based DeepACO [125] and GFACS [56] as well as the hybrid NAR/AR GLOP [127]. 3) **Improvement methods** include DACT [78], N2S [79] and NeuOpt [80]. We also include 4) **General-purpose RL** algorithm from the literature, including REINFORCE [109] with various baselines, Advantage Actor-Critic (A2C) [59] and Proximal Policy Optimization (PPO) [100] that can be readily be combined with any policy. Finally, we include 5) **Active search** (i.e., Transductive RL) methods AS [6] and EAS [40].

## 5 Benchmarking Study

We perform several benchmarking studies with our unified RL4CO library. Given the limited space, we invite the reader to check out the Appendix for supplementary material.

### 5.1 Flexibility and Modularity

**Changing policy components.** The integration of many state-of-the-art methods in RL4CO from the NCO field in a modular framework makes it easy to implement and improve upon state-of-the-art neural solvers for complex CO problems with only a few lines of code and improve upon them.[2] We demonstrate this in Table 2 for the FJSSP by gradually replacing or adding elements to the original SotA policy [106]. First, replacing the HGNN encoder with the more expressive Mat-Net encoder [63] already improves the average makespan by around 7%. Further improvements can be achieved by replacing the MLP decoder with the Pointer mechanism in the AM decoder [60] with gaps to BKS around $3\times$ lower compared to the original policy in Song et al. [106] even with greedy performance.

Table 2: Solutions obtained with RL4CO for the FJSSP with different model configurations.

| Encoder / Decoder | | FJSSP | |
| --- | --- | --- | --- |
| | | $10 \times 5$ | $20 \times 5$ |
| HGNN + MLP (g.) [106] | Obj. | 111.82 | 211.21 |
| | Gap | 15.8% | 12.1% |
| MatNet + MLP (g.) | Obj. | 103.91 | 197.92 |
| | Gap | 7.6% | 5.0% |
| MatNet + Pointer (g.) | Obj. | 101.17 | 196.3 |
| | Gap | 4.8% | 4.2% |
| MatNet + Pointer (s. x128) | Obj. | 98.31 | 192.02 |
| | Gap | 1.8% | 1.9% |

### 5.2 Constructive Policies

**Mind Your Baseline.** In on-policy RL, which is often employed in RL4CO due to fast reward function evaluations, several different REINFORCE baselines have been proposed to improve the performance. We benchmark several RL algorithms training constructive policies for routing problems of node size 50, whose underlying architecture is based on the encoder-decoder Attention Model [60] and whose main difference lies in how the REINFORCE baseline is calculated (we additionally train the AM with PPO as further reference). For a fair comparison, we run all baselines

---

[2]The different model configurations shown here can be obtained by simply changing the Hydra configuration file like the one shown in Appendix.

in controlled settings with the same number of optimization steps and report results in Table 3. We note that A2C generally underperforms other baselines. Such performance can be attributed to the fact that since in routing problems, the rewards are sparse (i.e., can only be calculated upon solving an entire problem), estimating the value of an entire instance $x$ is inherently a challenging task. Interestingly, while POMO [62], which takes as a baseline the shared baseline of all routes forcing each starting node to be different, may work well as baselines for problems in which near-optimal solutions can be constructed from any node (e.g., TSP), this may not be true for other problems such as the Orienteering Problem (OP): the reason is that in OP only a *subset*

Table 3: Optimality Gap obtained via greedy decoding.

| Method | TSP | CVRP | OP | PCTSP | PDP |
|---|---|---|---|---|---|
| A2C | 2.22 | 7.09 | 8.64 | 14.96 | 10.02 |
| AM-Rollout | 1.41 | 5.30 | 4.40 | 2.46 | 9.88 |
| POMO | 0.89 | 3.99 | 14.26 | 11.61 | 10.64 |
| Sym-NCO | 0.47 | 4.61 | 3.09 | 2.12 | 7.73 |
| AM-PPO | 0.92 | 4.60 | 3.05 | 2.45 | 8.31 |

of nodes should be selected in an optimal solution, while several states will be discarded. Hence, forcing the policy to select all of them makes up for a poor baseline. We remark that while SymNCO (whose shared baseline involves symmetric rotations and flips) [55] may perform well in Euclidean problems, this is not applicable in non-Euclidean CO, including asymmetric routing problems and scheduling. We found similar trends regarding actor-critic methods as A2C and PPO in the EDA mDPP problem [53], which we report in Appendix. Namely, a greedy rollout baseline [60] can do better than value-based methods due to the challenging task of instance value estimation.

**Decoding Schemes.** The solution quality of NCO solvers often shows significant improvements in performance to different decoding schemes, even with the exact NCO solvers. We evaluate the trained solver with different decoding schemes and settings as shown in Fig. 4.

**Generalization.** Using RL4CO, we can easily evaluate the generalization performance of existing baselines by employing supported environments that incorporate various VRP variant tasks and instance distributions (termed MTPOMO and MDPOMO, respectively). Empirical results on CVRPLib, shown in Table 4, reveal that training on different tasks significantly enhances generalization performance.

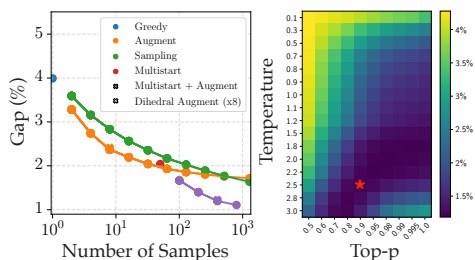

Figure 4: Decoding schemes study of POMO on CVRP50. [Left]: Pareto front of decoding schemes by the number of samples; [Right]: performance of sampling with different temperatures $\tau$ and $p$ values for top-$p$ sampling.

This finding underscores the necessity of building foundational models across diverse CO domains.

**Large-Scale Instances.** We evaluate large-scale CVRP instances of thousands of nodes, with more visualizations and scaling in Appendix. The last row of Table 5 illustrates the performance of the hybrid NAR/AR GLOP [127], while others refer to reproduced results from Ye et al. [127]. Our implementation in RL4CO improves the performance in not only speed but also solution quality.

### 5.3 Combining Construction and Improvement: Best of Both Worlds?

While constructive policies can build solutions in seconds, their performance is often limited, even with advanced decoding schemes such as sampling or augmentations. On the other hand, improvement methods are more suitable for larger computing budgets. We benchmark models on TSP with 50 nodes: the AR constructive method POMO [62] and the improvement methods DACT [78] and NeuOpt [80]. In the original implementation, DACT and NeuOpt started from a solution constructed randomly. To further demonstrate the flexibility of RL4CO, we show that bootstrapping improvement methods with constructive ones enhance convergence speed. Fig. 5 shows that bootstrapping with a pre-trained POMO policy significantly enhances the convergence speed. To further investigate the performance, we report the Primal Integral (PI) [8, 113, 111], which evaluates the evolution of solution quality over time. Improvement methods alone, such as DACT and NeuOpt, achieve 2.99 and 2.26 respectively, while sampling from POMO achieves 0.08. This shows that the "area under the curve" can be better even if the final solution is worse for constructive methods. Bootstrapping

| Benchmark | POMO | | MTPOMO | | MDPOMO | |
|---|---|---|---|---|---|---|
| | Obj. | Gap | Obj. | Gap | Obj. | Gap |
| Set A | 1075 | 3.13% | 1076 | 3.20% | **1074** | **2.97%** |
| Set B | 996 | 3.41% | 1003 | 4.06% | **995** | **3.26%** |
| Set E | 761 | 5.04% | **760** | **4.82%** | 762 | 5.07% |
| Set F | 813 | 13.52% | **798** | **12.09%** | 825 | 13.66% |
| Set M | 1259 | 16.37% | **1234** | **13.58%** | 1263 | 16.03% |
| Set P | 620 | 6.72% | **608** | **3.72%** | 613 | 5.04% |
| Set X | 73953 | 16.80% | **73763** | **16.69%** | 81848 | 23.69% |

Table 4: Results on CVRPLIB instances with models trained on $N = 50$. Greedy multi-start decoding is used.

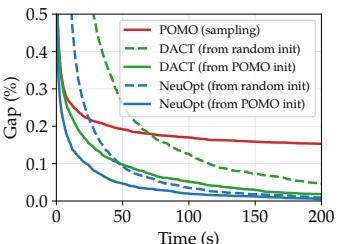

Figure 5: Bootstrapping improvement with constructive methods.

with POMO then improves DACT and NeuOpt to $0.08$ and $0.04$ respectively, showing the benefits of modularity and hybridization of different components.

# 6 Discussion

## 6.1 Limitations and Future Directions

While RL4CO is an efficient and modular library specialized in CO problems, it might not be suitable for any other task due to a number of area-specific optimizations, and we do not expect it to seamlessly integrate with, for instance, OpenAI Gym wrappers without some modifications. Another limitation of the library is its scope so far, namely RL. In fact, extending the library to support supervised methods and creating a comprehensive "AI4CO" library could benefit the whole NCO community. We

Table 5: Performance on large-scale CVRP instances.

| | CVRP1K | | CVRP2K | | CVRP7K | |
|---|---|---|---|---|---|---|
| | Obj. | Time (s) | Obj. | Time (s) | Obj. | Time (s) |
| LKH-3 | 46.4 | 6.2 | 64.9 | 20 | 245.0 | 501 |
| AM | 61.4 | 0.6 | 114.4 | 1.9 | 354.3 | 26 |
| TAM(AM) | 50.1 | 0.8 | 74.3 | 2.2 | 233.4 | 26 |
| TAM(LKH-3) | 46.3 | 1.8 | 64.8 | 5.6 | 196.9 | 33 |
| GLOP-G(AM)* | 47.1 | 0.4 | 63.5 | 1.2 | 191.7 | 2.4 |
| GLOP-G(LKH-3)* | 45.9 | 1.1 | 63.0 | 1.5 | 191.2 | 5.8 |
| GLOP-G(AM) | 46.9 | **0.3** | 64.7 | **0.7** | 190.9 | **2.0** |
| GLOP-G(LKH-3) | **45.5** | 0.5 | **62.8** | 0.8 | **190.1** | 3.9 |

additionally identify in Foundation Models[3] for CO and related scalable architectures a promising area of future research to overcome generalization issues across tasks and distributions, for which we provided some early clues.

## 6.2 Long-term Plans

Our long-term plan is to become the go-to RL for CO benchmark library. While not strictly tied to implementation and benchmarking, we are committed to helping resolve issues and questions from the community. For this purpose, we created a Slack workspace (link available in the online documentation) that by now has attracted more than 130 researchers. It is our hope that our work will ultimately benefit the NCO field with new ideas and collaborations.

# 7 Conclusion

This paper introduces RL4CO, a modular, flexible, and unified Reinforcement Learning (RL) for Combinatorial Optimization (CO) benchmark. We provide a comprehensive taxonomy from environments to policies and RL algorithms that translate from theory to practice to software level. Our benchmark library aims to fill the gap in unifying implementations in RL for CO by utilizing several best practices with the goal of providing researchers and practitioners with a flexible starting point for NCO research. We provide several experimental results with insights and discussions that can help identify promising research directions. We hope that our open-source library will provide a solid starting point for NCO researchers to explore new avenues and drive advancements. We warmly welcome researchers and practitioners to actively participate and contribute to RL4CO.

---

[3] https://github.com/ai4co/awesome-fm4co

## Acknowledgements

We want to express our gratitude towards anonymous reviewers of previous submissions who greatly helped us improve our paper. Even though rejections were not easy at first, they helped us refine our benchmark. Importantly, through our journey, we got to know several outstanding researchers in the community, who gave us even more motivation and meaning behind our work. We would also like to thank people in the AI4CO open research community who have contributed, and those who will, to RL4CO. We also thank OMELET for supporting us with additional compute. We invite practitioners and researchers to join us and contribute with bug reporting, feature requests, or collaboration ideas. A special thanks also goes to the TorchRL team for helping us in solving issues and improving the library.

## Potential Broader Impact

This paper presents work in the field of AI4CO. The main consequene may be that AI methods to solve CO problems may become accessible to the broad public, as our librabry is open source and readily available on GitHub. We do not see potential negative societal consequences as of today.

## Funding

This work was supported by a grant of the KAIST-KT joint research project through AI2XL Laboratory, Institute of Convergence Technology, funded by KT [Project No. G01210696, Development of Multi-Agent Reinforcement Learning Algorithm for Efficient Operation of Complex Distributed Systems] and by the Institute of Information & communications Technology Planning & Evaluation (IITP) grant funded by the Korean government(MSIT)[2022-0-01032, Development of Collective Collaboration Intelligence Framework for Internet of Autonomous Things].

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
