# RL4CO: an Extensive Reinforcement Learning for Combinatorial Optimization Benchmark
## *Supplementary Material*

## Table of Contents

# A RL4CO Library: Additional Material

## A.1 Why Choosing the RL4CO Library?

RL4CO, is a *unified* and *extensive* benchmark the RL-for-CO research area. We intend RL4CO to be used by researchers and practitioners alike of various levels of experience.

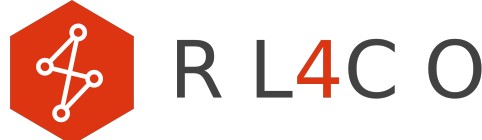

Figure 6: RL4CO benchmark logo.

**Availability and Future Support** RL4CO can be installed through PyPI[4]. We adhere to continuous integration, deployment, and testing to ensure reproducibility and accessibility.[5]

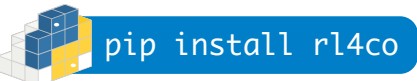

Figure 7: Installing the RL4CO package using `pip`.

**Open License** We adopt the open MIT license for all content contained in RL4CO with source code available at https://github.com/ai4co/rl4co. We ascribe to the principles of *libre software*[6]. Most reimplementations are from original authors and are re-licensed under the MIT license. Data and baseline-specific licenses are reported in Appendix A.3.

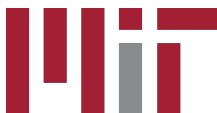

Figure 8: Unofficial - but widely used - open MIT license logo.

**Open Community** Through our journey, we started the AI4CO community[7], which is a non-profit, cross-institution, inclusive, and open research community. AI4CO originally started out as a Slack channel for discussing the RL4CO but evolved into a broader-visioned and inclusive space to communicate with other researchers about general NCO. The RL4CO library can be discussed in the AI4CO Slack [8] under the `#library-rl4co` channel. We warmly invite all interested people to join us.

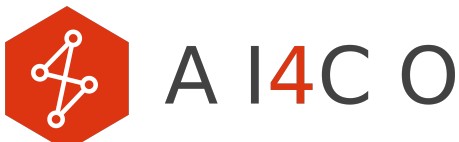

Figure 9: AI4CO community logo.

---

[4]https://pypi.org/project/rl4co/
[5]https://rl4co.readthedocs.io/en/latest/
[6]https://www.gnu.org/philosophy/free-sw.en.html
[7]Community Github: https://github.com/ai4co
[8]Slack invitation link: https://bit.ly/ai4co-slack

## A.2 On the Choice of the Software

During the development of RL4CO, we wanted to make it as simple as possible to integrate repro-ducible and standardized code adhering to the latest guidelines. As a main template for our codebase, we use Lightning-Hydra-Template [9] which we believe is a solid starting point for reproducible deep learning. We further discuss framework choices below.

**PyTorch** PyTorch [110] is a popular open-source deep-learning framework that has gained signif-icant traction in the research community. We chose PyTorch as the primary framework for RL4CO due to its intuitive API, dynamic computational graphs, strong community support, and seamless integration with the Python ecosystem. These features make PyTorch well-suited for rapid proto-typing and experimentation, which are essential in research settings. Moreover, most of the existing research in NCO has been implemented. It is currently being implemented using PyTorch, making it not only easier to build upon and compare with previous work but also easier for newcomers and experienced researchers.

**TorchRL and TensorDict** One of the software hindrances in RL is the bottleneck between CPU and GPU communication, majorly due to CPU-based operating environments. For this reason, we did not opt for OpenAI Gym [23] since, although it includes some level of parallelization, this does not happen on GPU and would thus greatly hinder performance. Kool et al. [74] creates *ad-hoc* environments in PyTorch to handle batched data efficiently. However, it could be cumbersome to integrate into standardized routines that include `step` and `reset` functions. As we searched for a better alternative, we found that TorchRL library [20], an official PyTorch project that allows for efficient batched implementations on (multiple) GPUs as well as functions akin to OpenAI Gym. We also employ the TensorDict [20] to handle tensors efficiently on multiple keys (i.e. in CVRP, we can directly operate transforms on multiple keys as locations, capacities, and more). This makes our environments compatible with the models in TorchRL, which we believe could further spread interest in the CO area.

**PyTorch Lightning** PyTorch Lightning [39] is a useful tool for abstracting away the boilerplate code, allowing researchers and practitioners to focus more on the core ideas and innovations. It features a standardized training loop and an extensive set of pre-built components, including auto-mated checkpointing, distributed training, and logging. PyTorch Lightning accelerates development time and facilitates scalability. We employ PyTorch Lightning in RL4CO to integrate with the Py-Torch ecosystem - which includes TorchRL- enabling us to leverage the rich set of tools and libraries available.

**Hydra** Hydra [148] is a powerful open-source framework for managing complex configurations in machine-learning models and other software. Hydra facilitates creating hierarchical configurations, making it easy to manage even very large and intricate configurations. Moreover, it integrates with command-line interfaces, allowing the execution of different configurations directly from the com-mand line, thereby enhancing reproducibility. We found Hydra to be effective when dealing with multiple experiments since configurations are saved both locally, as `yaml` files, and can be uploaded to monitoring software as Wandb [10] (or to any of the monitoring software supported by PyTorch Lightning).

## A.3 Licenses

We summarize the license of software that we employ in RL4CO in a non-exhaustive list in Table 6. Original environments and models from the authors are acknowledged through their respective cita-tions, with several links available in the library. RL4CO is licensed under the MIT license.

---

[9] https://github.com/ashleve/lightning-hydra-template
[10] https://wandb.ai/

Table 6: Reference code licenses and links.

| Type | Asset | License | Link |
|------|-------|---------|------|
| Library | PyTorch [110] | BSD-3 License | link |
| | PyTorch Lightning [39] | Apache-2.0 License | link |
| | TorchRL+TensorDict [20] | MIT License | link |
| | Hydra [148] | MIT License | link |
| Dataset | TSPLIB [116] | Available for any non-commercial use | link |
| | CVRPLib [86] | Available for any non-commercial use | link |
| | DPP PDNs [108] | Apache-2.0 | link |
| Solver | PyVRP [144] | MIT | link |
| | LKH3 [46] | Available for any non-commercial use | link |
| | OR-Tools [111] | Apache 2.0 License | link |

## B  Environments

This section provides an overview of the list of environments we experimented with at the time of writing. We organize environments by categories, which, at the time of writing, are:

1. **Routing (B.1)**

2. **Scheduling (B.2)**

3. **Electronic Design Automation (B.3)**

4. **Graph (B.4)**

### B.1  Routing

Routing problems are perhaps the most known class of CO problems. They are problems of great practical importance, not only for logistics, where they are more commonly framed, but also for industry, engineering, science, and medicine. The typical objective of routing problems is to minimize the total length of the paths needed to visit some (or all) the nodes in a graph. In the following section, we present each of these variants with details of their implementations.

**Common instance generation details**   Following the standard protocol of NCO for routing, we randomly sample node coordinates from the 2D unit square (i.e., $[0, 1]^2$). To ensure reproducibility in our experiments, we use specific random seeds for generating validation and testing instances. For the 10,000 validation instances, we use a random seed of 4321. For the 10,000 testing instances, we use a random seed of 1234. All protocols, including seed selection, align with the practices outlined by Kool et al. [74].

#### B.1.1  Traveling Salesman Problem (TSP)

The Traveling Salesman Problem (TSP) is a fundamental routing problem that aims to find the Hamiltonian cycle of minimum length. While the original TSP formulation employs mixed-integer linear programming (MILP), in the NCO community, the solution-finding process of TSP is differently formulated for constructive and improvement methods. For constructive methods, the TSP solution is generated by autoregressive solution decoding (i.e., the construction process) in line with Kool et al. [74]. In each step of node selection, we preclude the selection of nodes already picked in previous rounds. This procedure ensures the feasibility of constructed solutions and also allows for the potential construction of an optimal solution for any TSP instance. For improvement methods, it starts with an initial solution and iteratively searches for an optimal one using local search. In each step, the solution is locally adjusted based on a specified local search operator. We support two representative operators for TSP variants, including the 2-opt in line with Ma et al. [96] and the flexible k-opt in line with Ma et al. [98]. The former selects two nodes in the current solution and reverses the solution segment between them to perform a 2-opt exchange. The latter selects $k$ nodes

so that a k-opt is performed. Both methods ensure the feasibility of the solutions by masking invalid actions. The best solution after a set number of iterations is the final output.

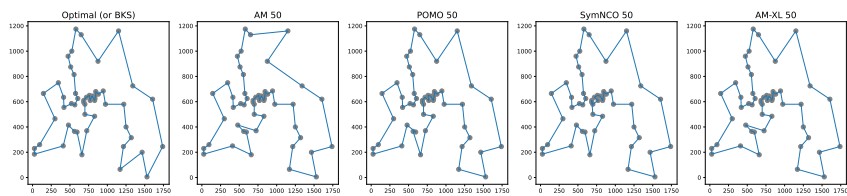

Figure 10: Sample TSP tours on TSPLib's Berlin 52 with different autoregressive models.

### B.1.2 Capacitated Vehicle Routing Problem (CVRP)

The Capacitated Vehicle Routing Problem (CVRP) is a popular extension of TSP, applicable to a variety of real-world logistics/routing problems (e.g., delivery services). In CVRP, each node has its own demand, and the vehicle visiting them has a specific capacity and always leaves from a special node called "depot". The vehicle can visit new nodes while their demand fits in its residual capacity (i.e. the total capacity decreased by the sum of the demands visited in the current path). When no nodes can be added to the path, the vehicle returns to the depot, and its full capacity is restored. Then, it embarks on another tour. The process is repeated until all nodes have been visited. By applying a similar logic to that of the TSP environment, we can reformulate CVRP as a sequential node selection problem, taking into account demands and capacity.

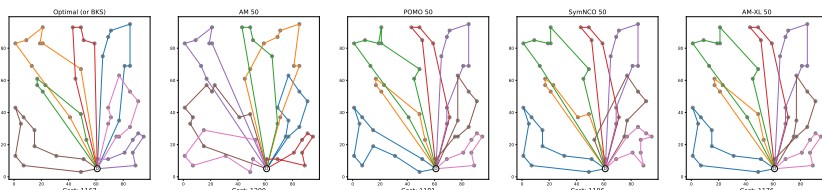

Figure 11: Sample CVRP tours on CVRPLib's A-n54-k7 instance with different autoregressive models.

**Additional generation details** To generate the demand, we randomly sample integers between 1 and 10. Without loss of generality, we fix the capacity of the vehicle at 1.0. Then, we normalize the demands by multiplying them by a constant that varies according to the size of the CVRP. The specific constant can be found in our implementation.

### B.1.3 Orienteering Problem (OP)

The Orienteering Problem (OP) is a variant of the TSP. In the OP, each node is assigned a prize. The objective of the OP is to find a tour, starting and ending at the depot, that maximizes the total prize collected from visited nodes, while abiding by a maximum tour length constraint. The OP can be framed as a sequential decision-making problem by enforcing the "return to depot" action when no nodes are visitable due to the maximal tour length constraint.

**Additional generation details** To generate the prize, we use the prize distribution proposed in Fischetti et al. [41], particularly the distribution that allocates larger prizes to nodes further from the depot.

### B.1.4 Prize Collecting TSP (PCTSP)

In the Prize Collecting TSP (PCTSP), each node is assigned both a prize and a penalty. The objective is to accumulate a minimum total prize while minimizing the combined length of the tour and

 the penalties for unvisited nodes. By making a minor adjustment to the PCTSP, it can model dif-
ferent subproblems that arise when using the Branch-Price-and-Cut algorithms for solving routing
problems.

### B.1.5 Pickup and Delivery Problem (PDP)

The Pickup and Delivery Problem (PDP) is an extension of TSP in the literature Helsgaun [46], Ma
et al. [97].[11] In PDP, a pickup node has its own designated delivery node. The delivery node can be
visited only when its paired pickup node has already been visited. We call this constraint *precedence
constraint*. The objective of the PDP is to find a complete tour with a minimal tour length while
starting from the depot node and satisfying the precedence constraints. We assume that *stacking*
is allowed, meaning that the traveling agent can visit multiple pickups prior to visiting the paired
deliveries. For constructive methods, the PDP solution construction is similar to that of TSP but
must obey precedence constraints. For improvement methods, we consider the ruin and repair local
search operator presented by Ma et al. [96]. In each step, a pair of pickup and delivery nodes are
removed from the current solution and then reinserted back into the solution with potentially better
positions. Invalid actions that violate precedence constraints are masked out to ensure the feasibility
of PDP solutions.

**Additional generation details** To generate the positions of the depot, pickups, and deliveries, we
sample the node coordinates from the 2D unit square. The first $N/2$ generated nodes are pickups,
and the remaining $N/2$ are their respective deliveries. The pickups and deliveries are paired. For a
pickup node $i$, its respective delivery is $i + N/2$ (excluding the depot index).

### B.1.6 Multi-Task VRP (MTVRP)

This environment introduces the 16 VRP variants in Liu et al. [89], Zhou et al. [157] with additional
enhancements, such as support for any number of variants in the same batch, as done in Berto et al.
[13]. The base logic is the same as CVRP: each node has a demand, and the vehicle has a specific
capacity by which it can deliver to nodes and return to the depot to replenish its capacity, with
the goal of minimizing the total tour distance. We report each modular constraint definition in the
following paragraphs according to Berto et al. [13], Wouda et al. [144]. Table 7 reports the list of all
variants and Fig. 12 illustrates the meaning of each MTVRP component.

| *VRP Variant* | Capacity (C) | Open Route (O) | Backhaul (B) | Duration Limit (L) | Time Windows (TW) |
|---|---|---|---|---|---|
| CVRP | ✓ | | | | |
| OVRP | ✓ | ✓ | | | |
| VRPB | ✓ | | ✓ | | |
| VRPL | ✓ | | | ✓ | |
| VRPTW | ✓ | | | | ✓ |
| OVRPTW | ✓ | ✓ | | | ✓ |
| OVRPB | ✓ | ✓ | ✓ | | |
| OVRPL | ✓ | ✓ | | ✓ | |
| VRPBL | ✓ | | ✓ | ✓ | |
| VRPBTW | ✓ | | ✓ | | ✓ |
| VRPLTW | ✓ | | | ✓ | ✓ |
| OVRPBL | ✓ | ✓ | ✓ | ✓ | |
| OVRPBTW | ✓ | ✓ | ✓ | | ✓ |
| OVRPLTW | ✓ | ✓ | | ✓ | ✓ |
| VRPBLTW | ✓ | | ✓ | ✓ | ✓ |
| OVRPBLTW | ✓ | ✓ | ✓ | ✓ | ✓ |

Table 7: The 16 VRP variants that are modeled by the MTVRP environment. All variants include the base
Capacity (C). The $k = 4$ features O, B, L, and TW can be combined into any subset, including the empty set
and itself (i.e., a *power set*) with $2^k = 16$ possible combinations.

---

[11]PDP is also called PDTSP (pickup and delivery TSP).

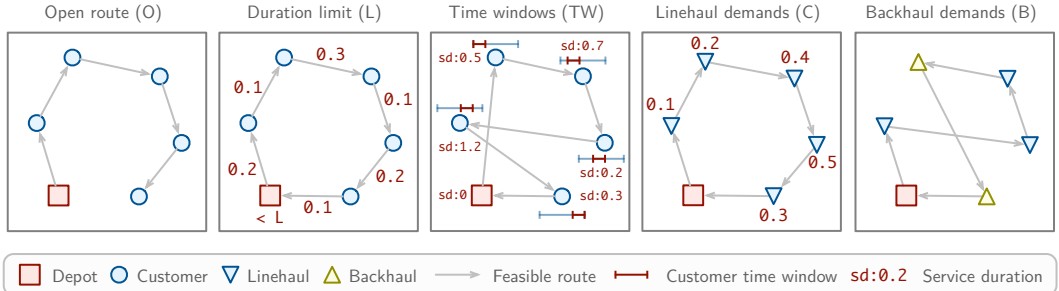

Figure 12: Different VRP attributes. Open routes (O) and duration limits (L) are *global attributes*, whereas time windows (TW), capacitated vehicles for linehaul demands (C) and backhaul demands (B) are *node attributes*. Attributes may be combined in different ways to define VRP variants.

677  *(C) Demand and Vehicle Capacity* $[q \in [0, Q]]$: Every node $i$, except the depot, has a demand $q_i$ that
678  must be satisfied by the vehicle with a uniform capacity of $Q > 0$. The sum of the demands served
679  by a vehicle in the same path must not exceed its capacity $Q$ at any point along its route.

680  *(O) Open Routes* $[o \in \{0, 1\}]$: With open routes, the distance between the last node and the depot
681  is not counted in the total path length. This represents the scenarios where vehicles are not required
682  to return to the depot after serving all assigned customers. Open routes are commonly found in
683  scenarios involving third-party drivers, who are typically compensated only for the deliveries they
684  complete, without the need to return to the depot [80].

685  *(B) Backhauls* $[p \in [0, Q]]$: Backhauls extend the concept of demand to include both delivery and
686  pickup requests, thus increasing vehicle utilization and leading to savings. Nodes are categorized as
687  either linehaul or backhaul nodes.[12] Linehaul nodes require delivery of demand $q_i$ from the depot
688  to the node $i$ (similar to CVRP), while backhaul nodes require a pickup of an amount $p_i$ to be trans-
689  ported from the node back to the depot. A vehicle can serve both linehaul and backhaul customers in
690  a single route, but all linehaul customers must be served before any backhaul customers. A typical
691  example of a backhaul problem is a laundry service for hotels that has to deliver clean towels and
692  pick up dirty ones, in which the precedence constraint of linehaul nodes is important due to possible
693  contamination [26].

694  *(L) Duration Limits* $[l \in [0, L]]$: Imposes a limit $L$ on the total travel duration (or distance) of
695  each vehicle route, ensuring a fair distribution of workload among different paths. This limit is
696  consistently applied to all routes in the problem.

697  *(TW) Time Windows* $[e, s, l \in [0, T]^3]$: Each node $i$, except for the depot, has an associated time
698  window $[e_i, l_i]$, which specifies the earliest and latest times at which it can be visited. When visiting
699  node $i$, the vehicle must wait for a time $s_i$ before leaving. The vehicle must arrive at customer $i$
700  before the end of its time window $l_i$, but if they arrive before the start of the time window $e_i$, they
701  must wait at the customer's location until the time window begins before starting the service. When
702  the vehicle returns to the depot, the time is reset to 0.

703  **Additional generation details**    We introduce the data generation details as follows:

704  *Locations*: We generate $n + 1$ locations randomly with $x_i$ and $y_i \sim U(0, 1), \forall i \in \{0, \dots, n\}$, where
705  $[x_0, y_0]$ represents the depot and $[x_i, y_i], i \in \{1, \dots, n\}$ are the other $n$ nodes.

*Capacity*: The capacity $C$ of the vehicle is determined based on the following calculation:

$$C = \begin{cases} 30 + \left\lfloor \frac{1000}{5} + \frac{n-1000}{33.3} \right\rfloor & \text{if } 1000 < n \\ 30 + \left\lfloor \frac{n}{5} \right\rfloor & \text{if } 20 < n \leq 1000 \ . \\ 30 & \text{otherwise} \end{cases}$$

---

[12]Note that another name of this problem, as adopted in LKH3 [46], is VRP with Pickup and Deliveries (VRPPD). However, we align with PyVRP [144] and do not use this name to prevent confusion with the *one-to-one PDP*, as we described before, where there is strict precedence between each pair of pickup and delivery.

*Open route*: the open route is an instance-wise flag: when set to $1$, the route is open, when $0$ is closed. We sample the flag from a uniform distribution with the same probability of the route being open or closed.

*Linehaul and Backhaul demands*: We generate demands according to the following schema:

1. Generate linehaul demands $q_i \in \{0, \ldots, Q\}$ for all nodes $i \in \{i, \ldots, n\}$. These are needed for both backhaul and linehaul scenarios.

2. Generate backhaul demands $p_i \in \{0, \ldots, Q\}$ for all nodes $i \in \{i, \ldots, n\}$.

3. For each node $i \in \{i, \ldots, n\}$, there is a probability of $0.2$ that it is assigned a backhaul demand, otherwise, its backhaul demand is set to be $0$.

Note that even in a backhaul setting, usually not all nodes are backhaul nodes, i.e., we need to consider both linehaul and backhaul demands in backhaul problem settings. All demands, both linehauls and backhauls, are scaled to $[0, 1]$ through division by the vehicle capacity.

*Duration limits*: Each route is assigned a fixed duration limit $L$ with a default value of $3$. We check that $2 * d_{0i} < L$ to make sure there is a feasible route for any customer.

*Time Windows*: We generate the time windows for each node $i \in \{1, \ldots, n\}$ according to the following steps:

1. Generate service time $s_i \in [0.15, 0.18]$.

2. Generate time window length $t_i \in [0.18, 0.2]$.

3. Calculate distance $d_{0i}$ from node to depot.

4. Calculate the upper bound for the start time $h_i = \frac{t_{max} - s_i - t_i}{d_{0i}} - 1$, where $t_{max}$ is the maximum time with a default value of $4.6$.

5. Calculate the start time as $e_i = (1 + (h_i - 1) \cdot u_i) \cdot d_{0i}$ with $u_i \sim U(0, 1)$.

6. Calculate the end time as $l_i = e_i + t_i$.

**Classical solvers**  We employ the SotA HGS implementation in PyVRP [144] and OR-Tools [111]. We make these solvers conveniently available through the `solve` API of the environment.

## B.2  Scheduling

Scheduling problems are a fundamental class of problems in operations research and industrial engineering, where the objective is to optimize the allocation of resources over time. These problems are critical in various industries, such as manufacturing, computer science, and project management. Currently, RL4CO implements three central scheduling problems, namely the flexible flow shop (FFSP), the job shop (JSSP), and the flexible job shop problem (FJSSP). Each of these problems has unique characteristics and complexities that need to be translated into the environment classes that we will describe hereafter.

### B.2.1  Job Shop Scheduling Problem (JSSP)

The job shop scheduling problem is a well-known combinatorial optimization problem. It is widely used in the operations research community as well as many industries, such as manufacturing and transportation. In the JSSP, a set of jobs $J$ must be processed by a set of machines $M$. Each job $J_i \in J$ consists of a set of $n_i$ operations $O_i = \{o_{ij}\}_{j=1}^{n_i}$ which must be processed one after another in a given order. The goal of the JSSP is to construct a valid schedule that adheres to the precedence order of the operations and minimizes the makespan, i.e., the time until the last job is finished. One example of such a schedule is shown in Fig. 13.

We formulate the JSSP as a sequential decision problem following the implementation of Tassel et al. [132]. Here, the environment iterates through distinct time steps $t = 1, \ldots, T$. At each time

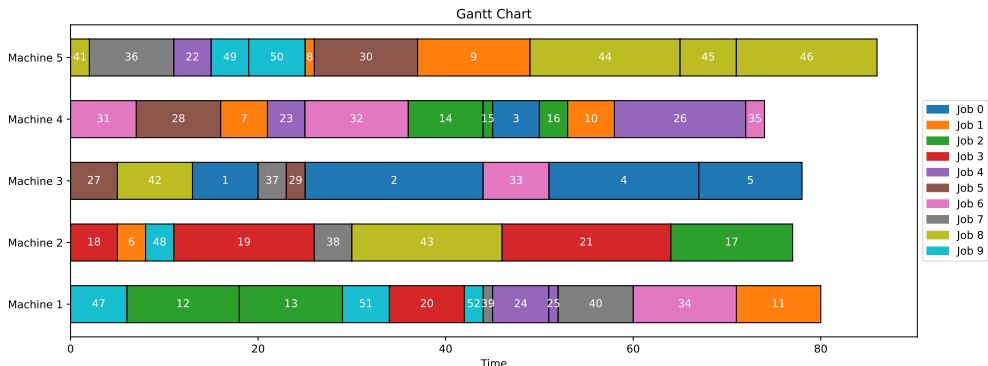

Figure 13: Example Schedule for the JSSP

step, the agent decides for each machine whether and which job to process next until all machines are busy or all jobs are being processed. In this case, the environment transitions to the next time step at which a machine becomes idle.

**Instance Generation** We follow the instance generation method described by Zhang et al. [153], which assumes that each job has exactly one operation per machine, i.e. $n_i = |M|$. Further, processing times for all operations are sampled iid. from a uniform distribution, with parameters specified in Table 8.

### B.2.2 Flexible Job Shop Scheduling Problem (FJSSP)

The flexible job shop scheduling problem is very similar to the JSSP. However, while in the classical JSSP, each operation $o_{ij} \in O$ has a specified machine and processing time $p_{ij}$, the flexible job shop scheduling problem (FJSSP) relaxes this assumption by allowing each operation to be processed by multiple eligible machines $M_k \subseteq M$, potentially with different processing times $p_{ijk}$ associated with the respective operation-machine pair. As a consequence, the agent does not only need to decide which job to process next, but also on which machine it should be processed.

**Instance Generation** We follow the instance generation method described by Song et al. [125], who sample $n_i$ operations for each job $J_i$ from a uniform distribution. Further, an average processing time $\bar{p}_{ij}$ is drawn for each operation $o_{ij} \in O$, and the actual processing time per eligible operation-machine pair is subsequently sampled from $U(0.8 \cdot \bar{p}_{ij}, 1.2 \cdot \bar{p}_{ij})$. The parameters used for instance generation can be found in Table 8.

Table 8: Instance generation parameters

|  | JSSP | | | | FJSSP | | | |
|---|---|---|---|---|---|---|---|---|
|  | $6 \times 6$ | $10 \times 10$ | $15 \times 15$ | $20 \times 20$ | $10 \times 5$ | $20 \times 5$ | $15 \times 10$ | $20 \times 10$ |
| $|J|$ | 6 | 10 | 15 | 20 | 10 | 20 | 15 | 20 |
| $|M|$ | 6 | 10 | 15 | 20 | 5 | 5 | 10 | 10 |
| $n_i$ | 6 | 10 | 15 | 20 | $U(4,6)$ | $U(4,6)$ | $U(8,12)$ | $U(8,12)$ |
| $\bar{p}_{ij}$ | $U(1,99)$ | $U(1,99)$ | $U(1,99)$ | $U(1,99)$ | $U(1,20)$ | $U(1,20)$ | $U(1,20)$ | $U(1,20)$ |
| $|M_i|$ | 1 | 1 | 1 | 1 | $U(1,5)$ | $U(1,5)$ | $U(1,10)$ | $U(1,10)$ |

### B.2.3 Flexible Flow Shop Problem (FFSP)

The flexible flow shop problem (FFSP) is a complex and widely studied optimization problem in production scheduling. It involves $N$ jobs to be processed in $S$ stages, each containing multiple machines ($M > 1$). Each job must pass through the stages in a specified order, but within each stage, it can be processed by any available machine. A critical constraint is that no machine can process more than one job at a time. The objective is to find an optimal schedule that minimizes the

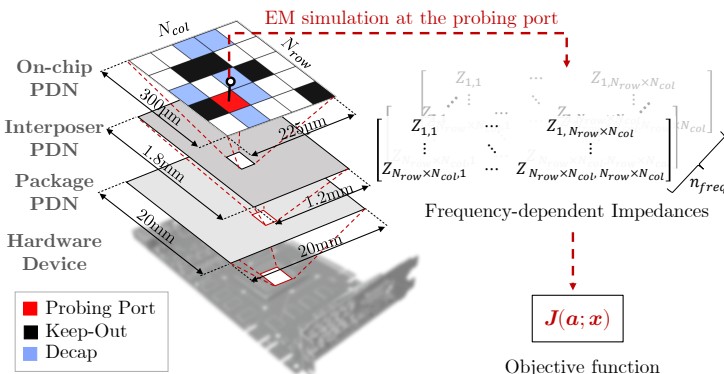

Figure 14: Grid representation of the target on-chip PDN for the DPP problem with a single probing port from Kim et al. [67].

total time required to complete all jobs. We formulate the FFSP as a sequential decision process, where at each time step $t = 0, 1, ...$ and for each idle machine, the agent must decide whether and which job to schedule. If all machines are busy or all jobs are currently being processed, the environment moves to the next time step $t + 1$, and the process repeats until all jobs for each stage have been scheduled.

**Instance Generation** We follow the data generation process described by Kwon et al. [77], who sample processing times for each job-machine pair and for every stage independently from a discrete uniform distribution.

## B.3 Electronic Design Automation

Electronic Design Automation (EDA) is a sophisticated process that involves the use of software tools to design, simulate, and analyze electronic systems, particularly integrated circuits (ICs) and printed circuit boards (PCBs). EDA encompasses a wide range of tasks, from schematic capture and layout design to verification and testing. Optimization is a critical aspect of EDA, where the goal is to achieve the best possible performance, power efficiency, and cost within the constraints of the design. This involves solving complex problems that can be either continuous, such as cell placement [52], or combinatorial, like decap placement [67]. RL4CO integrates CO problems in EDA as benchmarking environments.

### B.3.1 Decap Placement Problem (DPP)

The decap placement problem (DPP) is an electronic design automation problem (EDA) in which the goal is to maximize the performance with a limited number of the decoupling capacitor (decap) placements on a hardware board characterized by asymmetric properties, measured via a probing port. The decaps cannot be placed on the location of the probing port or in keep-out regions (which represent other hardware components) as shown in Fig. 14. The optimal placement of a given number of decaps can significantly impact electrical performance, specifically in terms of power integrity (PI) optimization. PI optimization is crucial in modern chip design, including AI processors, especially with the preference for 3D stacking memory systems like high bandwidth memory (HBM) [54]. For comprehensive details, we follow the configuration guidelines provided in [67].

**Baseline solvers** We employ two meta-heuristic baselines commonly used in hardware design as outlined in [67]: random search (RS) and genetic algorithm (GA) [62]. GA has shown promise as a method for addressing the decap placement problem (DPP).

**Instance generation details** We use the same data for simulating the hardware board as Kim et al. [67], with power distribution network (PDN) datasets from Park et al. [108]. We randomly select one

probing port and a number between 1 and 50 keep-out regions sampled from a uniform distribution for generating instances. As in the routing benchmarks, we select seed 1234 for testing the 100 instances.

### B.3.2 Multi-Port Decap Placement Problem (mDPP)

We further consider a more complex and realistic version compared to Kim et al. [67]. The multi-port decap placement problem (mDPP) is a generalization of DPP from Appendix B.3.1 in which measurements from multiple probing ports are performed. The objective function can be either the mean of the reward from the probing ports: 1) (*Maxsum*): the objective is to maximize the average PI among multiple probing ports and 2) (*Maxmin*): maximize the minimum PI between them.

**Instance generation details**    The generation details are the same as DPP, except for the probing port. A number of probing ports between 2 and 5 is sampled from a uniform distribution, and probing ports are randomly placed on the board, just like the other components.

## B.4   Graph

Many CO problems can be (re-)formulated on graphs [64]. In typical CO problems on graphs, actions are defined on nodes/edges, while problem variables and constraints are incorporated in graph topology and node/edge attributes (e.g., weights). The graph-based formulation gives us concise and systematic representations of CO problems. Moreover, existing traditional and machine-learning algorithms for graphs are off-the-shelf tools.

### B.4.1   Facility Location Problem (FLP)

The optimal usage of limited resources is an important problem to consider in many different fields and has various forms. One specific form of such a problem can be formulated as the facility location problem (FLP), where one aims to choose a given number of locations among given candidates, and the objective is to minimize the overall cost of service (e.g., the sum of the distance from the users to the nearest facility) [38].

Many real-world problems can be abstracted as instances of FLP. For example, franchise brands may need to determine where to open new retail stores to maximize accessibility and profitability [120]; governments may need to consider the placement of public facilities (e.g., hospitals and schools) to maximize the convenience for citizens to use them [101]; energy companies may need to determine the best locations for power centers (e.g., power plants and wind farms) to minimize transmission losses [92].

**Formal definition**    We consider the following specific form of the facility location problem (FLP) used in existing NCO literature [141, 25]: (1) given a group of $n$ locations $x_1, x_2, \ldots, x_n \in \mathbb{R}^d$ in a $d$-dimensional space (usually $d = 2$ or $3$) and $k < n$, (2) we aim to choose $k$ locations $x_{i1}, x_{i2}, \ldots, x_{ik}$ among the given $n$ locations as the locations of facilities, (3) to minimize the sum of the distance from all the $n$ locations to the nearest facility, i.e., $\sum_{j=1}^{n} \min_{t=1}^{k} \text{dist}(x_j, x_{it})$. We specially consider the Euclidean distance, i.e., $\text{dist}(x_i, x_j) = \|x_i - x_j\|_2$.

**Instance generation details**    The locations are ($d = 2$)-dimensional generated i.i.d. at random. For each location, each coordinate is sampled i.i.d. uniformly at random between 0 and 1. Each instance contains $n = 100$ locations, and $k = 10$ locations are to be chosen.

**Classical solvers**    We apply two MIP solvers, Gurobi [44] and SCIP [14], to obtain the optimal solutions.

### B.4.2  Maximum Coverage Problem (MCP)

In many real-world scenarios, one needs to allocate limited resources to achieve maximum coverage, which is a fundamental concern across various domains. One specific formulation is called the maximum coverage problem (MCP), where the goal is to select a subset of sets from a given family of sets to maximize the coverage, i.e., the (weighted) size of the union of the selected sets [65].

As a mathematical abstraction, the MCP can be used to represent many real-world problems. For example, radio frequency identification (RFID) system engineers may need to set RFID readers in an optimal way to ensure the maximum coverage of RFID tags [4]; marketers may need to choose proper forms of advertisement to reach the maximum number of customers [126]; in security applications (e.g., deploying security cameras), one may need to select the optimal deployment to maximize the coverage of the protected area [105].

**Formal definition**  We consider the following specific form of the maximum coverage problem (MCP) used in existing NCO literature [141, 25]: (1) given $m$ items (WLOG, $[m] := \{1, 2, 3, \ldots, m\}$), where each item $t$ has weight $w_t$, and a family of $n$ sets $S_1, S_2, \ldots, S_n \subseteq [m]$ for some positive integer $m$ and $k < n$, (2) we aim to choose $k$ sets $S_{i1}, S_{i2}, \ldots, S_{ik}$ among the given $n$ sets, (3) to maximize the total weighted coverage of the $k$ chosen sets, which is the sum of the weights of items contained in any chosen set, i.e., $\sum_{t \in \bigcup_{j=1}^{k} S_{ij}} w_t$.

**Instance generation details**  First, $m = 200$ items are generated, and the item weights are generated i.i.d., where each weight is a random integer sampled between 1 and 10 (inclusive) uniformly at random. Then, $n = 100$ sets are generated i.i.d., where for each set, we first sample its size between 5 and 15 uniformly at random and then choose that number of items uniformly at random. After generation, $k = 10$ locations are to be chosen.

**Classical solvers**  We apply two MIP solvers, Gurobi [44] and SCIP [14], to obtain the optimal solutions.

### B.5  Additional Environments and Beyond

We also include in the library additional environments that have been implemented but not fully benchmarked in this paper yet, such as the ATSP, mTSP, Skill-VRP, SMTWTP, and SPCTSP, to name a few. We did not count these in the total environment count (hence the "conservative" estimate). Moreover, several projects, among which co-authors of this paper, have adapted several new environments to their own tasks, which may be included in the future.

Although RL4CO already contains several environments, we acknowledge that the library can be further extended within new directions, which we briefly describe. One such direction is multi-objective combinatorial optimization [87, 29], which is a recently trending research topic of practical importance. Moreover, providing modular reward evaluators to optimize different objectives (for instance, min-max, tardiness) is another avenue of research that we recommend exploring [109]. Of practical importance is also non-euclidean routing, which so far has received comparatively less attention in this field but is practically important (i.e., DIMACS challenge[13]). Finally, multi-agent CO [40, 130, 131, 15] is another interesting area of research, which recent approaches model as a sequential decision-making process [123, 155].

Implementing new environments is relatively easy: we created a notebook under the `examples/` folder demonstrating how one can implement a custom environment from the base logic to a fully functioning model. We expect to host an even wider variety of environments in the future, thanks to the community, and invite contributors to help us in our journey.

---

[13]http://dimacs.rutgers.edu/programs/challenge/vrp/

## C Baselines

This section provides an overview of the key components and methods implemented in RL4CO that can be used as baselines for comparative evaluation. The term "baselines" broadly refers to both the RL algorithms that define the learning objectives and update rules, as well as the policy architectures that parameterize the agent's behavior in the environment, given that several papers introduce a mix of RL training schemes and policy improvements. We categorize baselines into:

1. **General-purpose RL algorithms (C.1)**
2. **Constructive autoregressive (AR) methods (C.2)**
3. **Constructive non-autoregressive (NAR) methods (C.3)**
4. **Improvement methods (C.4)**
5. **Active search methods (C.5)**

### C.1 General-purpose RL Algorithms

In the following descriptions of RL algorithms, we use the notations of a full problem instance $\boldsymbol{x}$ and a complete solution $\boldsymbol{a}$ for simplicity. However, note that these algorithms are also applicable to the usual notion of the sum of rewards over partial states $s_t$ and actions $a_t$.

#### C.1.1 REINFORCE [128]

REINFORCE (also known as policy gradients in the literature) is an online RL algorithm whose loss function gradient is given by:

$$\nabla_\theta \mathcal{L}_a(\theta|\boldsymbol{x}) = \mathbb{E}_{\pi(\boldsymbol{a}|\boldsymbol{x})} \left[ (R(\boldsymbol{a}, \boldsymbol{x}) - b(\boldsymbol{x})) \nabla_\theta \log \pi(\boldsymbol{a}|\boldsymbol{x}) \right], \quad (5)$$

where $b(\cdot)$ is a baseline function used to stabilize training and reduce gradient variance. The choice of $b(\cdot)$ can greatly influence the final performance.

#### C.1.2 Advantage Actor-Critic (A2C) [73]

A2C is an algorithm that can be used to solve the RL objective in Eq. (3). It consists of an actor (policy network) and a critic (value function estimator). The actor is trained to maximize the expected cumulative reward by following the policy gradient, while the critic is trained to estimate the value function. The advantage function, computed as the difference between the reward $R(\boldsymbol{a}, \boldsymbol{x})$ and the value function $V(\boldsymbol{x})$, is used to weight the policy gradient update for the actor. This can be seen as a modification of the REINFORCE gradient, where the baseline $b(\boldsymbol{x})$ is replaced by the value function $V(\boldsymbol{x})$:

$$\nabla_\theta \mathcal{L}_a(\theta|\boldsymbol{x}) = \mathbb{E}_{\pi(\boldsymbol{a}|\boldsymbol{x})} \left[ (R(\boldsymbol{a}, \boldsymbol{x}) - V(\boldsymbol{x})) \nabla_\theta \log \pi(\boldsymbol{a}|\boldsymbol{x}) \right]. \quad (6)$$

The critic is updated by minimizing the mean-squared error between the estimated value function and the target value, which is the reward for the given problem instance $\boldsymbol{x}$:

$$\mathcal{L}_c = \mathbb{E}_{\boldsymbol{x} \sim P(\boldsymbol{x})} (R(\boldsymbol{a}, \boldsymbol{x}) - V(\boldsymbol{x}))^2. \quad (7)$$

By using the advantage function, A2C reduces the variance of the policy gradient and stabilizes training compared to the standard REINFORCE algorithm.

#### C.1.3 Proximal Policy Optimization (PPO) [119]

PPO is another algorithm that can be used to solve the RL objective in Eq. (3). It is an on-policy algorithm that aims to improve the stability of policy gradient methods by limiting the magnitude of policy updates. To this end, PPO introduces a surrogate objective function that constrains the probability ratio between the target policy $\pi_\theta$ that is optimized and a reference policy $\pi_{\theta_{\text{old}}}$, which is periodically updated. This clipping mechanism prevents drastic changes to the target policy, ensuring more reliable and stable learning. Formally, the PPO objective function is given by:

$$\mathcal{L}_{\text{CLIP}}(\theta) = \mathbb{E}_{\boldsymbol{x} \sim P(\boldsymbol{x})} \Big[ \mathbb{E}_{\boldsymbol{a} \sim \pi_{\theta_{\text{old}}}(\boldsymbol{a}|\boldsymbol{x})} \big[ \min(\frac{\pi_\theta(\boldsymbol{a}|\boldsymbol{x})}{\pi_{\theta_{\text{old}}}(\boldsymbol{a}|\boldsymbol{x})} A^{\pi_{\theta_{\text{old}}}}(\boldsymbol{x}, \boldsymbol{a}),$$

$$\text{clip}(\frac{\pi_\theta(\boldsymbol{a}|\boldsymbol{x})}{\pi_{\theta_{\text{old}}}(\boldsymbol{a}|\boldsymbol{x})}, 1 - \epsilon, 1 + \epsilon) A^{\pi_{\theta_{\text{old}}}}(\boldsymbol{x}, \boldsymbol{a}))\Big]\Big], \quad (8)$$

where $\theta_{\text{old}}$ represents the parameters of the reference policy, typically a periodically created copy of the parameters $\theta$ of the target policy. Further, $A^{\pi_{\theta_{\text{old}}}}(\boldsymbol{x}, \boldsymbol{a})$ is the advantage function estimated using the reference policy, and $\epsilon$ is a hyperparameter that controls the clipping range, typically set to a small value like 0.2.

The advantage function in PPO is estimated using a learned value function $V_\phi(\boldsymbol{x})$, where $\phi$ represents the parameters of the value function. The advantage is computed as:

$$A^{\pi_{\theta_{\text{old}}}}(\boldsymbol{x}, \boldsymbol{a}) = R(\boldsymbol{a}, \boldsymbol{x}) - V_\phi(\boldsymbol{x}). \quad (9)$$

The value function is learned by minimizing the mean-squared error between the estimated value and the actual return:

$$\mathcal{L}_V(\phi) = \mathbb{E}_{\boldsymbol{x} \sim P(\boldsymbol{x})} \left[ (R(\boldsymbol{a}, \boldsymbol{x}) - V_\phi(\boldsymbol{x}))^2 \right]. \quad (10)$$

An optimization step in PPO updates both, the parameters $\theta$ of the target policy and the parameters $\phi$ of the value function by combining $\mathcal{L}_{\text{CLIP}}$ and $\mathcal{L}_V(\phi)$ in a single loss $\mathcal{L}_{\text{PPO}} = \mathcal{L}_{\text{CLIP}} + \beta \mathcal{L}_V(\phi)$, where $\beta$ is a hyperparameter [119].

## C.2 Constructive Autoregressive (AR)

### C.2.1 Attention Model (AM) [74]

The Attention Model (AM) from Kool et al. [74] is an encoder-decoder architecture based on the self-attention mechanism [136] that is at the heart of several state-of-the-art NCO methods, including RL-based ones [76, 69, 51] as well as (self-)supervised ones [37, 93, 94]. In the original AM, only node features are considered: with abuse of notation from Fig. 3, we consider the `InitEmbedding` as the *node embedding*, and split the *context embedding* into a `ContextEmbedding` which updates the current query and `DynamicEmbedding` that updates the current cached keys and values.

**Multi-Head Attention** Before delving into the encoder and decoder structures, we briefly introduce the notion of Multi-Head Attention (MHA) from Vaswani et al. [136], since it is used across several NCO methods. MHA allows the model to jointly attend to information from different representation subspaces at different positions, enabling it to capture various relationships between the input elements. Importantly, it is flexible in handling a variable number of elements.

In the MHA operation, the input sequences $Q$ (queries), $K$ (keys), and $V$ (values) are linearly projected to $H$ different subspaces using learned matrices $W_i^Q$, $W_i^K$, and $W_i^V$, respectively, where $H$ is the number of attention heads:

$$Q_i = QW_i^Q \quad (11)$$
$$K_i = KW_i^K \quad (12)$$
$$V_i = VW_i^V \quad (13)$$

for $i = 1, \ldots, H$.

The attention weights are computed as the scaled dot-product between the queries and keys, followed by a softmax operation:

$$A_i = \text{Softmax}\left(\frac{Q_i K_i^T}{\sqrt{d_k}} + M\right) \quad (14)$$

where $d_k$ is the dimension of the keys, used as a scaling factor to prevent the dot-products from getting too large, and $M$ is an optional mask matrix that can be used to prevent attention to certain positions (e.g. infeasible actions in a CO problem).

The output of each attention head is computed as the weighted sum of the values, using the attention weights:

$$Z_i = A_i V_i \quad (15)$$

Finally, the outputs of all attention heads are concatenated and linearly projected using a learned matrix $W^O$ to obtain the final output of the MHA operation:

$$\text{MHA}(Q, K, V) = \text{Concat}(Z_1, \ldots, Z_H)W^O \quad (16)$$

This multi-head attention mechanism allows the model to learn different attention patterns and capture various dependencies between the input elements, enhancing the representational power of the model. The queries, keys, and values can come from the same input sequence (self-attention, i.e. $Q = K = V$) or from different sequences (cross-attention), depending on the application. While the attention operation is at the core of much of the current SotA deep learning [134], this scales as $O(L)^2$ where $L$ is the sequence length, such as the number of nodes in a TSP. Thus, an efficient implementation such as FlashAttention [34, 33] is important, as shown in Appendix E.7.2.

**Encoder** The encoder's primary task is to encode input $x$ into a hidden embedding $h$. The structure of $f_\theta$ comprises two trainable modules: the `InitEmbedding` and encoder blocks. The `InitEmbedding` module typically transforms problem features into the latent space and problem-specific compared to the encoder blocks, which often involve plain multi-head attention (MHA):

$$h = f_\theta(x) \triangleq \text{EncoderBlocks}(\text{InitEmbedding}(x)) \qquad (17)$$

Each encoder block in the AM is composed of an Attention Layer, similar to Vaswani et al. [136]. Each layer $\ell$ is composed of multi-head attention (MHA) for message passing and a Multi-Layer Perceptron (MLP, also known as *feed-forward network (FFN)*), with skip-connections and normalization (Norm):

$$\hat{h} = \text{Norm}\left(h^{(\ell-1)} + \text{MHA}(h^{(\ell-1)}, h^{(\ell-1)}, h^{(\ell-1)})\right) \qquad (18)$$

$$h^{(\ell)} = \text{Norm}\left(\hat{h} + \text{MLP}(\hat{h})\right) \qquad (19)$$

with $\ell = [1, \ldots, N]$ where $N$ is the number of encoding layers and $h^0 = \text{InitEmbedding}(x)$. In the encoder side, we have $Q = K = V = h^{(\ell-1)}$), hence self-attention.

The original implementation of the AM uses $N = 3$ layers $H = 8$ heads of dimension $d_\text{k} = \frac{d_\text{h}}{M} = 16$, an MLP with one hidden layer of dimension 512 with a ReLU activation function, and a Batch Normalization [56] as normalization.

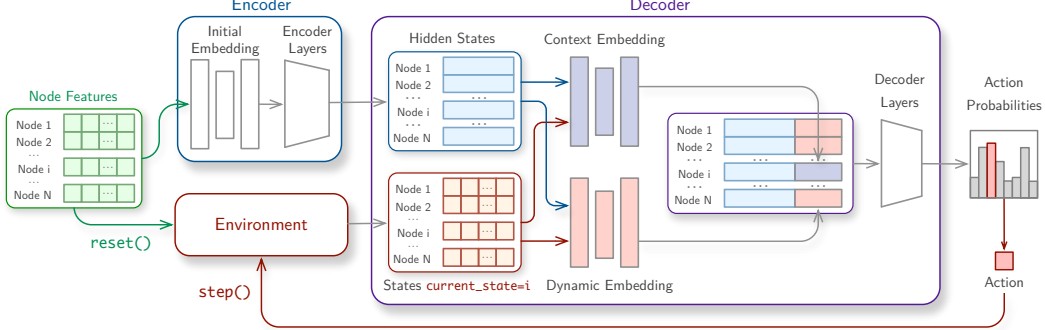

Figure 15: An overview of the modularized Attention Model policy in RL4CO.

**Decoder** The decoder $g_\theta$ autoregressively constructs the solution based on the encoder output $h$ and the state at current step $t$, $s_t$. The solution decoding involves iterative steps until a complete solution is constructed: at each step, starting from the current node's $i$ query $q_t^i$

$$q_t^i = \text{ContextEmbedding}(h, s_t), \qquad (20)$$

$$h_t^c = \text{MHA}(q_t^i, K_t^g, V_t^g, M_t), \qquad (21)$$

$$z = \frac{V_t^p h_t^c}{\sqrt{d_k}} \qquad (22)$$

where $M_t$ is the set of feasible actions (i.e. the `action_mask`), projections $K_t^g, V_t^g, V_t^p = W_k^g h, W_v^g h, W_v^p h$ can either be precomputed once as cache or updated via a dynamic embedding $K_t^g, V_t^g, V_t^p = \text{DynamicEmbedding}(W_k^g h, W_v^g h, W_v^p h, s_t, h, x),$, depending on the problem. We note that Eq. (22) is usually referred to as the pointer mechanism (in the codebase, we refer to Eq. (21) and Eq. (22) as the `PointerAttention`). Finally, logits $z$ (unnormalized output of policy

$\pi$) are transformed into a probability distribution over the action space:

$$p = \text{Softmax}\left(C \cdot \tanh(\boldsymbol{z})\right) \tag{23}$$

where logits $\boldsymbol{z}$ for infeasible actions can be set to $-\infty$ to avoid choosing them; and the $C$ value (called *tanh clipping*, usually set to 10) serves in improving the exploration [8]. We note that Eq. (23) can also include additional operations such as temperature scaling, top-k, and top-p filtering.

**Baseline** Kool et al. [74] additionally introduces the *rollout* baseline $b$ for Eq. (5). At the end of each epoch, a greedy rollout of a baseline policy $\pi_{BL}$ is executed for each of the sampled instances $\boldsymbol{x}$, whose values become baselines for REINFORCE. The algorithm compares the current training policy with a saved baseline policy (similar to the DQN target network [103]) at the end of every epoch, and replace the parameters of $\pi_{BL}$ with the current trained $\pi$ if the improvement is significant with a paired t-test of (i.e., $5\%$ in the original paper).

### C.2.2  Ptr-Net [139]

The original Pointer Network (Ptr-Net) is introduced in Vinyals et al. [139] and further refined to be trained with RL in [8]. The base architecture predates the AM [74]: an attention mechanism is employed to select outputs of variable length, thus "pointing" at them. The baseline architecture additionally uses an LSTM [47], which in practice has less expressivity than full-fledged attention.

### C.2.3  POMO [76]

POMO introduces the *shared* baseline to lower the REINFORCE variance. The key idea is that one can sample rollouts when decoding by forcing diverse starting nodes, which is a powerful inductive bias for certain problems, such as the TSP, in which multiple optimal initial starting points exist. The baseline $b_{\text{shared}}$ is the average of all rollouts:

$$b_{\text{shared}}(s) = \frac{1}{N} \sum_{j=1}^{N} R(\boldsymbol{a_j}, \boldsymbol{x}) \tag{24}$$

where $N$ is the number of sampled trajectories (typically set as the number of nodes).

### C.2.4  SymNCO [69]

SymNCO considers the symmetric nature of combinatorial problems and solutions. There are two major symmetries in combinatorial optimization: 1) *Problem symmetries*: The representation of the input 2D coordinates should have equivalent optimal solution sets and 2) *Solution symmetries*: Multiple permutations can represent an identical cyclic line graph. To reflect this symmetric nature, SymNCO augments the AM architecture by incorporating an auxiliary invariant representation loss function to ensure input 2D symmetries. Additionally, SymNCO employs a shared baseline as Eq. (24) similar to POMO but samples rollouts from both different symmetric problem inputs and solutions together. The implementation is not vastly different from AM and POMO; the primary addition is the symmetric-aware augmentation functions.

### C.2.5  PolyNet [51]

The PolyNet method proposed by Hottung et al. [51] enables the learning of a set of complementary solution strategies within a single model. This facilitates the easy sampling of diverse solutions at test time, resulting in improved exploration of the search space and, consequently, enhanced overall performance. Unlike many other approaches, PolyNet does not artificially increase exploration by forcing diverse starting actions, as initially proposed by Kwon et al. [76]. Instead, PolyNet utilizes its inherent diversity mechanism, based on its novel architecture and the Poppy loss [43, 27]:

$$\nabla_\theta \mathcal{L} = \mathbb{E}_{\pi(\boldsymbol{a}^*|\boldsymbol{x})} \left[ (R(\boldsymbol{a}^*, \boldsymbol{x}) - b_\circ(\boldsymbol{x})) \nabla_\theta \log \pi_\theta(\boldsymbol{a}^*|\boldsymbol{x}) \right], \tag{25}$$

to facilitate exploration during the search process, where $a^*$ is the *best* solution of $K$ PolyNet samples and $b_\circ(\boldsymbol{x})$) is the average reward of the $K$ samples. This can improve performance for problems in which the first action greatly influences the performance.

### C.2.6 HAM [82]

The Heterogeneous Attention Model (HAM) [82] is a model specialized for Pickup and Delivery problems (PDP, Appendix B.1.5), characterized by hard one-to-one precedence constraints. To differentiate between pickup and delivery pairs, it introduces *ad hoc* encoder blocks with a specialized attention mechanism that can differentiate between pickup and delivery pairs.

### C.2.7 MTPOMO [89]

The MTPOMO developed by Liu et al. [89] proposes to adopt a unified model to learn across various VRP variants. It is motivated by the fact that the diverse VRPs are different combinations of several shared underlying attributes. By training on a limited number of VRPs with basic attributes, the model is capable of generalizing to a vast array of VRP variants, each representing different combinations of these attributes. This approach extends POMO [76] by incorporating an attribute composition block, facilitating learning across different problems. The cross-problem learning demonstrates promising zero-shot generation performance on unseen VRPs and benefits out-of-distribution performance.

### C.2.8 MVMoE [157]

The MVMoE architecture proposed by Zhou et al. [157] incorporates mixture-of-experts (MoEs) [57, 60, 121] into attention-based model (e.g., POMO [76]), such that the model capacity can be greatly enhanced without a proportional increase in computation. For the *encoder* part, MVMoE replaces a feed-forward network (FFN) with an MoE layer, which typically consists of 1) $m$ experts $\{E_1, E_2, \ldots, E_m\}$, each of which is also an FFN with independent trainable parameters, and 2) a gating network $G$ parameterized by $W_G$, which decides how the inputs are distributed to experts. Given a single input $x$, $G(x)$ and $E_j(x)$ denote the output of the gating network (i.e., an $m$-dimensional vector), and the output of the $j_{\text{th}}$ expert, respectively. The output of an MoE layer is calculated as:

$$\text{MoE}(x) = \sum_{j=1}^{m} G(x)_j E_j(x). \tag{26}$$

The gating algorithm follows the node-level input-choice gating proposed by Shazeer et al. [121], which leverages a sparse gating network: $G(x) = \text{Softmax}(\text{Top}K(x \cdot W_G))$. In this way, only $k$ experts with partial model parameters are activated, hence saving the computation. For the *decoder* part, MVMoE replaces the final linear layer of MHA with an MoE layer, including $m$ linear layers and a gating network $G$. To balance the empirical performance and computational complexity, a hierarchical gating mechanism is further proposed to utilize MoEs during decoding efficiently. In this case, the MoE layer in the decoder includes two gating networks $\{G, G'\}$, $m$ experts $\{E_1, E_2, \ldots, E_m\}$, and a dense layer $D$. Given a batch of inputs $X$, the hierarchical gating routes them in two stages. In the first stage, $G'$ decides to distribute inputs $X$ to either the sparse or dense layer. In the second stage, if $X$ is routed to the sparse layer, the gating network $G$ is activated to route nodes to experts on the node level by using the default gating algorithms, i.e., the input-choice gating. Otherwise, $X$ is routed to the dense layer $D$ and transformed into $D(X)$. In summary, the hierarchical gating learns to output $G'(X)_0 \sum_{j=1}^{m} G(X)_j E_j(X)$ or $G'(X)_1 D(X)$. Empirically, hierarchical gating has been found to be more efficient, albeit with a slight sacrifice in in-distribution performance, while demonstrating superiority with out-of-distribution data.

### C.2.9 L2D [153]

Learning to Dispatch (L2D) proposed by Zhang et al. [153] is a DRL method to solve the JSSP. It comprises of the usual encoder-decoder structure, where a graph convolution network (GCN) is employed to extract hidden representations from the JSSP instance. To this end, L2D formulates the JSSP as a disjunctive graph, with nodes reflecting the operations of the problem instance. Nodes of operations that belong to the same job are connected via directed arcs, specifying their precedence relation. Moreover, operations to be processed on the same machine are connected using undirected

arcs. Using the resulting neighborhood $\mathcal{N}$ of the nodes, the GCN performs massage passing between adjacent operations to construct their hidden representations. Formally, let $\boldsymbol{h}^0$ be the initial embeddings of operations $O$ and $\tilde{\boldsymbol{A}}$ the adjacency matrix with added self-loops of operations, then a graph convolutional layer can be described as follows:

$$\boldsymbol{h}^{(l+1)} = \sigma \left( \tilde{\boldsymbol{D}}^{-\frac{1}{2}} \tilde{\boldsymbol{A}} \tilde{\boldsymbol{D}}^{-\frac{1}{2}} \boldsymbol{h}^{(l)} \boldsymbol{W}^{(l)} \right)$$

Here, $\boldsymbol{h}^{(l)}$ are the operation embeddings at layer $l$, $\boldsymbol{W}^{(l)}$ is a trainable weight matrix at layer $l$, and $\sigma(\cdot)$ is an activation function such as ReLU. Further, $\tilde{\boldsymbol{D}}$ is the diagonal degree matrix of $\tilde{\boldsymbol{A}}$, ensuring appropriate scaling of the features.

Given the operation embeddings, the decoder of L2D first extracts for each job the embedding of the operation that needs to be scheduled next and then feeds them to an MLP $f : \mathbb{R}^{J \times d} \to \mathbb{R}^{J \times 1}$ to obtain logits for each job $j \in (1, ..., J)$. In contrast to Kool et al. [74] for example, who encode the CO problem once and then generate actions autoregressively using only the decoder, Zhang et al. [153] use the GCN encoder after each step to generate new hidden representations that reflect the current state of the problem.

### C.2.10 HGNN [125]

The heterogeneous graph neural network (HGNN) is a neural network architecture proposed by [125] to solve the FJSSP. Similar to L2D, HGNN considers an FJSSP instance as a graph. However, instead of treating an FJSSP instance as a disjunctive graph, Song et al. [125] formulate it as heterogeneous graph with operations and machines posing different node types. Again, operations are connected to each other via directed arcs that specify the precedence relation. Machines are only connected to operations that they are able to process, and the edge weights indicate the respective processing times. To encode the graph, HGNN first projects operations $O \in x$ and machines $M \in x$ into a mutual embedding space $\mathbb{R}^d$ using type-specific transformations $\boldsymbol{W}^O$ and $\boldsymbol{W}^M$, respectively. Given the initial hidden representations $\boldsymbol{h}_i^0$ and $\boldsymbol{h}_k^0$ for operations $o_i \in O$ and machines $m_k \in M$, respectively, as well as edge embeddings $\boldsymbol{h}_{ik}$, an HGNN layer conducts weighted message passing between operations and machines using the processing times of operation-machine pairs:

$$\boldsymbol{h}_i^{l+1} = \sum_{j \in \mathcal{N}_i} \epsilon_i \boldsymbol{h}_j^l, \quad \text{where} \tag{27}$$

$$\epsilon_{ij} = \underset{j \in \mathcal{N}_i}{\text{Softmax}}(\boldsymbol{a}^\top [\boldsymbol{h}_j^l || \boldsymbol{h}_{ij}]). \tag{28}$$

Since operations in the FJSSP can be processed by multiple machines, the decoder must specify not only which job to process next but also on which machine the operation of the selected job should be executed. To this end, Song et al. [125] concatenates the hidden representations of every operation with the embeddings of every machine. The resulting embeddings are fed to an MLP $f : \mathbb{R}^{J \times M \times 2d} \to \mathbb{R}^{J \times M \times 1}$, which generates the sampling probabilities for the respective action.

### C.2.11 MatNet [77]

The MatNet architecture proposed by Kwon et al. [77] adjusts the attention model [74] so that it is applicable to bipartite graphs with node types $\mathcal{I}$ and $\mathcal{J}$ as well as a weight matrix $E \in \mathbb{R}^{|\mathcal{I}| \times |\mathcal{J}|}$ corresponding to the edges connecting nodes from the two sets. The novelty of this architecture is that instead of using self-attention as in the attention model, MatNet uses cross-attention to perform message passing between both node sets and augments the resulting attention scores with the weight matrix $E$. Formally, let $\mathcal{Z}$ be the set of all nodes $i \in \mathcal{I} \cup \mathcal{J}$, $\mathcal{Z}_{\phi_i}$ the subset of nodes of the same type as $i$ and $\mathcal{Z}_{\phi_i}^{\complement}$ the set of nodes of the respective type. Then, cross-attention is defined as:[14]

$$\alpha'_{ij} = \frac{\boldsymbol{q}_i^\top \boldsymbol{k}_j}{\sqrt{d_k}}, \qquad \forall i \in \mathcal{Z}, \, j \in \mathcal{Z}_{\phi_i}^{\complement} \tag{29}$$

---

[14]For succinctness, note that we omit head and layer enumeration.

where

$$\boldsymbol{q}_i = W_{\phi_i}^Q \boldsymbol{h}_i^{l-1} \qquad \boldsymbol{k}_j = W_{\phi_i}^K \boldsymbol{h}_j^{l-1} \tag{30}$$

and weight matrices $W_{\phi_i}^Q$ and $W_{\phi_i}^K \in \mathbb{R}^{d_k \times d_h}$ being learned by the update function corresponding to nodes of type $\phi_i$. After that, MatNet augments $\alpha'_{ij}$ with the corresponding edge weight $e_{ij}$ and maps it through a feed-forward neural network FF : $\mathbb{R}^2 \to \mathbb{R}$ to a scalar score, which is then normalized using the softmax function:

$$\alpha_{ij} = \frac{\exp(\epsilon_{ij})}{\sum_{q \in \mathcal{Z}_{\phi_i}^{\mathtt{C}}} \exp(\epsilon_{iq})}, \quad \epsilon_{ij} = \mathrm{FF}\big([\alpha'_{ij}||e_{ij}]\big) \tag{31}$$

The resulting weights are used to compute a weighted average of the embeddings $\boldsymbol{v}_j = W_{\phi_i}^V \boldsymbol{h}_j^{l-1}$ of the nodes in $\mathcal{Z}_{\phi_i}^{\mathtt{C}}$. In the end, skip connections, layer normalization (LN), and feed-forward layers are used as in Vaswani et al. [136]. Besides the original MatNet implementation, RL4CO also implements a version that applies both self- and cross-attention, successively as proposed by Luttmann and Xie [95]. This makes MatNet not only applicable to bipartite graph problems but to the more general class of heterogeneous graphs [95].

### C.2.12 DevFormer [67]

We employ online RL variants of DevFormer [67] (DF), an Attention-Model [74] variant specifically designed for autoregressive construction of DPP solutions from Appendix B.3.1. We note that the DF training scheme was initially designed for offline training; however, in this study, we benchmark DF as a sample-efficient online reinforcement learning approach. We benchmark the DF version for RL with the same node and context embedding structure as the original in Kim et al. [67]. We modify the embeddings in the mDPP environment (Appendix B.3.2) version to include the location of multiple probing ports. Min-max and min-sum mDPP versions utilize the same embeddings and are trained separately.

### C.3 Constructive Non-Autoregressive (NAR)

### C.3.1 DeepACO [150]

Ant Colony Optimization (ACO) is an evolutionary algorithm that has been successfully applied to various COPs. Traditionally, customizing ACO for a specific problem requires the expert design of knowledge-driven heuristics. However, this routine of algorithm customization exhibits certain deficiencies: 1) it requires extra effort and makes ACO less flexible; 2) the effectiveness of the heuristic measure heavily relies on expert knowledge and manual tuning; and 3) designing a heuristic measure for less-studied problems can be particularly challenging, given the paucity of available expert knowledge.

DeepACO is designed to automatically strengthen the heuristic measures of existing ACO algorithms and dispense with laborious manual design in future ACO applications. DeepACO consists of two stages: 1) training a neural model to map a COP instance to its heuristic measures, and 2) incorporating the learned heuristic measures into ACO to bias solution constructions and local search. During the training phase, DeepACO parameterizes the heuristic space with a graph neural network (GNN) [61]. It trains the GNN across COP instances with REINFORCE, towards minimizing the expected objective value of both constructed solutions and solutions refined by local search. During the inference phase, DeepACO utilizes the well-trained GNN to generate heuristic measures for ACO. Optionally, DeepACO interleaves local search with neural-guided perturbation to refine the constructed solutions. For more details, please refer to [150].

DeepACO is the first NAR model implemented in RL4CO, laying the foundation for other NAR models later integrated into RL4CO. DeepACO offers a versatile methodological framework that allows for further algorithmic enhancements in neural architecture, training paradigms, decoding

strategies, and problem-specific adaptations. Notable improvements over DeepACO are introduced by GFACS [70].

### C.3.2 GFACS [70]

While DeepACO [150] provides promising results and opens new doors for pretraining heuristic measures for the ACO algorithm using deep learning, their method is sub-optimal for two major reasons. Firstly, they utilized policy gradient reinforcement learning (RL), which is an on-policy method that cannot leverage powerful off-policy techniques such as local search. Secondly, their method cannot effectively capture the multi-modality of heuristic distribution because the RL method cannot accurately model multi-modal probabilistic distributions considering the symmetric nature of combinatorial space, where multiple trajectories can lead to identical solutions.

The methodology of GFACS shares a very similar structure with DeepACO. The key difference lies in the learning procedure; GFACS employs generative flow networks (GFlowNets) [9, 11] for learning the heuristic matrix. Additionally, they leverage effective off-policy exploration methods using local search. The inference procedure with the learned heuristic matrix remains exactly the same. With the RL4CO modular implementation, both DeepACO and GFACS can run similarly and be comparable at the modular level, allowing future researchers to improve certain modules of training or inference.

### C.3.3 GLOP [152]

Most NCO methods struggle with real-time scaling-up performance; they are unable to solve routing problems involving thousands or tens of thousands of nodes in seconds, falling short of the needs of modern industries. GLOP (**G**lobal and **L**ocal **O**ptimization **P**olicies) is proposed to address this challenge. It partitions a large routing problem into sub-TSPs and further partitions potentially large (sub-)TSPs into small Shortest Hamiltonian Path Problems (SHPPs). It is the first hybrid method to integrate NAR policies for coarse-grained problem partitions and AR policies for fine-grained route constructions, leveraging the scalability of the former and the meticulousness of the latter.

**1) AR (Sub-)TSP Solver.** The (Sub-)TSP Solver in GLOP initializes TSP tours using a Random Insertion heuristic, which greedily inserts nodes to minimize cost. These tours are then improved through a process of decomposition and reconstruction. Specifically, the solver decomposes a complete tour into several subtours, which are treated as instances of the Shortest Hamiltonian Path Problem (SHPP). Each subtour is solved using an AR local policy referred to as a "reviser". These revisers are applied in rounds called "revisions" to enhance the initial tour iteratively. The subtours are normalized and optionally rotated to improve the model's performance. After solving the SHPP instances, the subtours are reassembled into an improved complete tour. This method allows for efficient and parallelizable improvements on large-scale TSPs.

**2) NAR General Routing Solver.** The general routing solver in GLOP additionally implements an NAR global policy that either partitions all nodes into multiple sub-TSPs (e.g., for CVRP) or subsets all nodes to form a sub-TSP (e.g., for PCTSP). The NAR global policy is parameterized by a graph neural network (GNN) that processes sparsified input graphs and outputs a partition heatmap. GLOP clusters or subsets nodes by sequentially sampling nodes based on the partition heatmap while adhering to problem-specific constraints. The sub-TSPs are then solved by the (Sub-)TSP solver. The global policy is trained using REINFORCE to output partitions that could lead to the best-performing final solutions after solving sub-TSPs.

GLOP is integrated into RL4CO as the first hybrid method that combines NAR and AR policies, indicating the versatility of RL4CO in accommodating various methodological paradigms. It is promising to further investigate the emerging possibilities that arise when viewing AR and NAR methods from a unified perspective and combining them synergistically. RL4CO provides a flexible and extensible platform for exploring such hybridization in future research.

## C.4 Improvement methods

Improvement methods leverage RL to train a policy that iteratively performs rewriting exchanges on the current solution, aiming to generate a new solution with potentially lower costs. As in constructive methods, the policy of improvement methods is also based on the encoder-decoder structure.

### C.4.1 DACT [96]

Improvement methods typically take node features and solution features (positional information of nodes in the current solution) as key inputs. Encoding VRP solutions involves processing complex relationships between Node Feature Embeddings (NFEs) and Positional Feature Embeddings (PFEs). However, directly adopting the original Transformer to add the two types of embeddings, as done by Wu et al. [145], can cause mixed attention score correlations and impairing performance. To address this, the Dual-Aspect Collaborative Transformer (DACT) proposes DAC-Att, which processes NFEs and PFEs separately and employs cross-aspect referential attention to understand the consistencies and differences between the two embedding aspects. This approach avoids mixed correlations and allows detailed modeling of hidden patterns. Another key issue is the Positional Encoding (PE) method. While the original Transformer's PE works well for linear sequences, it may not suit the cyclic nature of VRP solutions. To address this, DACT proposes Cyclic Positional Encoding (CPE), inspired by cyclic Gray codes, which generates cyclic real-valued coding vectors to capture the topological structure of VRP solutions and improve generalization. Additionally, DACT redesigns the RL algorithm for improvement methods, introducing a Proximal Policy Optimization with Curriculum Learning (PPO-CL) algorithm to improve training stability and efficiency.

In RL4CO, DACT is implemented and modularized so that other methods can easily reuse components like CPE encoding and the PPO-CL algorithm. It also reuses common parts (such as node embedding initialization, decoding functions, etc) from the implementation of constructive methods, indicating the flexibility of the RL4CO framework.

### C.4.2 N2S [97]

The Neural Neighborhood Search (N2S) method extends the capabilities of improvement methods to pickup and delivery problems (PDP). Expanding on the DACT approach, N2S leverages a tailored MDP formulation for a ruin-repair neighborhood search process. It uses a Node-Pair Removal decoder in the ruin stage and a Node-Pair Reinsertion decoder in the repair stage, allowing efficient operation on pickup-delivery node pairs. However, more complex decoders increase computational costs in the policy network, requiring a balance between encoders and decoders. To address this, N2S introduces Synthesis Attention (Synth-Att), which learns a single set of embeddings and synthesizes attention scores from various node feature embeddings using a Multilayer Perceptron (MLP) module. This promotes lightweight policy networks and enhances model expressiveness. The N2S encoder with the efficient Synth-Att represents a state-of-the-art design of improvement encoder, which is adopted in the latest works [97, 98].

In RL4CO, N2S reuses the CPE encoding and the PPO-CL algorithm implemented in DACT. The efficient N2S encoder is also modularized and designed to be shared among other improvement methods to process the complex relationships between different feature embeddings.

### C.4.3 NeuOpt [98]

A key bottleneck of improvement methods like DACT is their simplistic action space design, which typically uses smaller, fixed $k$ values (2-opt or 3-opt) due to decoders struggling with larger, varying $k$. To address this, the latest improvement method introduces Neural k-Opt (NeuOpt), a flexible solver capable of handling any given $k \geq 2$. NeuOpt employs an action factorization method to break down complex k-opt exchanges into a sequence of basis moves (S-move, I-move, E-move), with the number of I-moves determining the $k$ value. This step-by-step construction allows the model to automatically determine a suitable $k$. Similar to variable neighborhood search, NeuOpt combines varying $k$ values across search steps, balancing coarse-grained and fine-grained searches,

which is crucial for optimal performance. NeuOpt also features a Recurrent Dual-Stream (RDS) decoder with recurrent networks and two decoding streams for contextual modeling and attention computation, effectively capturing the complex dependencies between removed and added edges.

In RL4CO, NeuOpt is implemented by reusing the successful CPE and PPO-CL training modules from DACT, as well as the efficient encoder from N2S. This demonstrates the strength and versatility of the RL4CO coding library, which allows for the easy integration of proven methodologies.

### C.5   Active Search Methods

Active search methods are examples of *transductive* RL, in which an RL algorithm is run to finetune a pre-trained policy on specific test-time instances.

### C.5.1   Active Search (AS) [8]

In active search proposed by Bello et al. [8], a model is fine-tuned to a single test instance. To this end, active search uses the same loss formulation as during the original training of the model. Over the course of the search process, the model's performance on the single test instance improves, leading to the discovery of high-quality solutions. While active search is easy to implement, as the search process closely follows the training process, it is often very slow since all model weights are adjusted for each test instance individually.

### C.5.2   Efficient Active Search (EAS) [50]

Efficient active search (EAS), proposed by Hottung et al. [50], builds upon the idea of active search and trains a model on a single instance at test time to enable a guided search. However, EAS only updates a subset of parameters during the search and allows most operations to be performed in parallel across a batch of different instances. This approach not only reduces computational costs but also results in a more stable fine-tuning process, leading to an overall improvement in solution quality.

## D   Benchmarking Setup

### D.1   Metrics

#### D.1.1   Gap to BKS

The Gap to Best Known Solution (BKS) is a commonly used metric to evaluate the performance of optimization algorithms on benchmark instances. It measures the relative difference between the best solution found by the algorithm and the BKS for a given problem instance. Given a problem instance $i$, let $\boldsymbol{a}_i$ be the objective value of the best solution found by the algorithm, and let $\boldsymbol{a}_i^*$ be the objective value of the BKS for that instance. The Gap to BKS for the $i$-th instance is defined as:

$$\text{Gap to BKS}_i = 100 \times \left( \frac{\boldsymbol{a}_i - \boldsymbol{a}_i^*}{\boldsymbol{a}_i^*} \right) \tag{32}$$

The Gap to BKS is expressed as a percentage, with a value of 0% indicating that the algorithm has found a solution that matches the BKS. A positive Gap to BKS indicates that the algorithm's solution is worse than the BKS, while a negative Gap to BKS (though less common) indicates that the algorithm has found a new best solution for the instance[15].

---

[15]Note that when calculating the gap for a set of instances, one should do an average of gaps, i.e. $\frac{1}{n} \sum_{i=1}^{n} \text{Gap to BKS}_i$, instead of calculating the gap of the average $100 \times \sum \boldsymbol{a}_i / \sum \boldsymbol{a}_i^*$, which might yield similar results in some settings but prone to error especially for certain distributions.

 **D.1.2 Primal Integral**

1292 The Primal Integral (PI) is a metric that evaluates the anytime performance of optimization algo-
1293 rithms by capturing the trade-off between solution quality and computational time [12, 133]. It is
1294 defined as the area under the curve of the incumbent solution value plotted against time, normalized
1295 by the BKS value and the total time budget:

$$PI = 100 \times \left( \frac{\sum_{i=1}^{n} \boldsymbol{a}_{i-1} \cdot (t_i - t_{i-1}) + \boldsymbol{a}_n \cdot (T_{\max} - t_n)}{T_{\max} \cdot \boldsymbol{a}^*} - 1 \right) \tag{33}$$

1296 where $T_{\max}$ is the total time budget, $\boldsymbol{a}_i$ is the incumbent solution value at time $t_i$, and $\boldsymbol{a}^*$ is the
1297 best known solution value. A lower PI percentage indicates better anytime performance. The PI
1298 complements other metrics, such as the Gap to BKS, by providing insights into the temporal aspect
1299 of an algorithm's performance, making it particularly useful for assessing anytime algorithms [58].

1300 **D.1.3 Runtime Measurement**

1301 **Runtime normalization** Comparing the run-time efficiency of different methods across various
1302 hardware configurations can be challenging. In the RL4CO benchmark, we generally run the in-
1303 ference on a single machine; when this is not possible due to resource limitations, we employ the
1304 run-time normalization approach based on the *PassMark* hardware rating[16]. This approach nor-
1305 malizes time budgets and run times during the evaluation process, allowing for a more equitable
1306 comparison of methods. We use the definition of Accorsi et al. [1], Thyssens et al. [133] in normal-
1307 izing: the reference machine combines a single CPU thread and a single GPU, the *PassMark* score
1308 $s$ for GPU-based methods is calculated as:

$$s = \frac{1}{2}(\#\text{CPU} \cdot \text{CPU\_Mark} + \#\text{GPU} \cdot \text{GPU\_Mark}) \tag{34}$$

1309 To normalize the solution time from machine 1 to machine 2, we calculate $\tilde{t}_2 = t_1 \frac{s_1}{s_2}$, where $t_1$ is
1310 the solution time on machine 1, $s_1$ is the *PassMark* score of machine 1, and $s_2$ is the *PassMark* score
1311 of machine 2. Note that in the case of most classical solvers, the GPU\_Mark is simply set to 0 due
1312 to them running on CPU.

1313 **Cross-solver comparisons** Another aspect of NCO evaluation that has to be addressed is the fact
1314 that evaluation between classical and learned solvers is often done on different devices, namely on
1315 (single-threaded) CPUs and GPUs, respectively. Moreover, while multiple instances in NCO can
1316 usually be solved in a batch, this is not usually the case for classical solvers. A more correct way is
1317 to measure the *per-instance* solution time (which we do on large-scale NAR routing), which is more
1318 realistic for real-world applications. For other studies, we employ the standard procedure of NCO of
1319 evaluating times on batches as done in the original methods, making sure to compare "apples with
1320 apples" (i.e., different NCO approaches are compared with the same settings). We note that while
1321 RL4CO focuses on comparisons between NCO solvers and creating an open-source ecosystem for
1322 this specific area, future studies (and possibly works in the RL4CO community) may also include
1323 comparisons with classical solvers under different conditions, which we recognize as an important
1324 research direction.

1325 **D.2 Hardware & Software**

1326 **D.2.1 Hardware**

1327 Most experiments (during testing) were carried out on a machine equipped with two AMD EPYC
1328 7542 32-CORE PROCESSOR CPUs with 64 threads each and four NVIDIA RTX A6000 graphic
1329 cards with 48 GB of VRAM, of which only one is used during inference. We note that, due to the
1330 amount of experiments and contributions, training was performed on a variety of hardware combina-

---

[16]*PassMark*: https://www.passmark.com/ is also used in the 2022 DIMACS challenge: http://dimacs.rutgers.edu/programs/challenge/vrp/.

tions, particularly University clusters. We found RL4CO to be robust and efficient across different combinations of CPU, GPU, and software. Throughout the text, we may report the hardware setting on which testing took place if it differs from the default one. In case different configurations were used or results were reported from previous works, we refer to Appendix D.1.3 for result standardization.

### D.2.2 Software

Software-wise, we used `Python 3.11` and PyTorch 2.3 [110][17], most notably due to the native implementation of `scaled_dot_product_attention`. Given that most models in RL constructive methods for CO generally use attention for encoding states, FlashAttention has some boost on the performance (between $5\%$ and $20\%$ saved time depending on the problem size) when training is subject to mixed-precision training, which we do for all experiments. During decoding, the FlashAttention routine is not called since, at the time of writing, it does not support maskings other than causal; this could further boost performance compared to older implementations. Refer to Appendix A.2 for additional details regarding notable software choices of our library, namely TorchRL, PyTorch Lightning, and Hydra.

### D.3 Hyperparameters

### D.3.1 Common Hyperparameters

Common hyperparameters can be found in the `config/` folder from the RL4CO library, which can be conveniently loaded by Hydra. We provide yaml-like configuration files below, divided by experiments in Listing 1.

### D.3.2 Changing Policy Components

We train the models evaluated in Table 2 using the same number of training instances as well as identical hyperparameters. Specifically, models are trained for 10 epochs on 2.000 training instances using the PPO algorithm with clip range $\epsilon = 0.2$. The training dataset is split into batches of size 100 to construct the replay buffer. For the PPO optimization we sample mini-batches of size 512 from the replay buffer until it is empty and repeat this for $\mathcal{R} = 3$ inner epochs. All models use an embedding dimension $d_h$ of 256. The number of encoder layersis set to $L = 3$ in each case. Further, MatNet and the AM Pointer use $H = 8$ attention heads. The parameters of the models are updated using the Adam optimizer with learning rate $10^{-4}$. Afterwards, the trained policies are evaluated on 1.000 randomly generated test instances. The Hydra config files corresponding to this experiment, which also implement the different model architectures, can be found in the `config/experiment/scheduling` folder from the RL4CO library

### D.3.3 Mind Your Baseline

We run all models to match the original implementation details under *controlled* settings. In particular, we run all models for $250,000$ gradient steps with the same Adam [71] optimizer with a learning rate of $10^{-4}$ and 0 weight decay. For POMO, we match the original implementation details of weight decay as $10^{-6}$. For POMO, the number of multistarts is the same as the number of possible initial locations in the environment (for instance, for TSP50, 50 starts are considered). In the case of Sym-NCO, we use 10 as augmentation for the shared baseline; we match the number of effective samples of AM-XL to the ones of Sym-NCO to demonstrate the differences between models.

---

[17]During development, we also used beta wheels as well as manually installed version of FlashAttention [34, 33]. Note that software version varied in terms of training runs depending on the author who ran experiments (e.g. any range of Python and PyTorch as $[3.9, 3.10, 3.11] \times [2.0, 2.1, 2.2, 2.3]$, which RL4CO can support out of the box on multiple devices and operating systems.

```yaml
1   defaults: # override default configurations under configs/
2     - override /env: tsp.yaml
3     - override /model: am.yaml
4     - override /callbacks: default.yaml
5     - override /trainer: default.yaml
6     - override /logger: wandb.yaml
7
8   # Environment
9   env:
10    generator_params:
11      num_loc: 50
12
13  # RL Algorithm and policy (env passed automatically)
14  model:
15    policy: # override policy parameters to pass to the RL algo
16      _target_: rl4co.models.zoo.am.policy.AttentionModelPolicy
17      embed_dim: 128
18      num_heads: 8
19      num_encoder_layers: 3
20      feedforward_hidden: 128
21      env_name: "${env.name}" # automatically construct env embeddings
22    baseline: "rollout" # REINFORCE baseline
23    batch_size: 512
24    train_data_size: 1_280_000
25    optimizer_kwargs:
26      lr: 1e-4
27
28  # Optional override of checkpoint parameters
29  model_checkpoint:
30    dirpath: ${paths.output_dir}/checkpoints
31    filename: "epoch_{epoch:03d}"
32
33  # Trainer
34  trainer:
35    max_epochs: 100
36    gradient_clip_val: 1.0
37    max_epochs: 100
38    precision: "16-mixed" # allows for FlashAttention
39    strategy: DDPStrategy # efficient for multiple GPUs
40    matmul_precision: "medium" # speeds up calculation
41
42  # Logging
43  logger:
44    wandb:
45      project: "rl4co"
46      name: "am-tsp${env.generator_params.num_loc}"
```

Listing 1: Example `example.yaml` configuration for the AM from the AR routing experiments. Additional parameters are modularized in the actual configs and moved to the other config folders (such as `env/tsp.yaml` so that a single experiment config is not too cluttered. Running this configuration is simple: placed under `configs/experiments/`, it can be called with `python run.py experiment=example`.

The number of epochs for all models is 100, except for AM-XL (500). We also employ learning rate scheduling, in particular, `MultiStepLR` [18] with $\gamma = 0.1$ on epoch 80 and 95; for AM-XL, this applies on epoch 480 and 495.

**PPO for the AM**   We follow other hyperparameters for REINFORCE baselines. We set the number of mini-epochs to 2, mini-batch size to 512, clip range to 0.2, and entropy coefficient $c_2 = 0.01$. Interestingly, we found that normalizing the advantage as done in the Stable Baselines PPO2 imple-

---

[18] https://pytorch.org/docs/stable/generated/torch.optim.lr_scheduler.MultiStepLR

mentation[19] slightly hurt performance, so we set the normalize advantage parameter to `False`. We suspect this is because the NCO solvers are trained on *multiple* problem instances, unlike the other RL applications that aim to learn a policy for a single MDP.

**Sample Efficiency Experiments**  We keep the same hyperparameters as the *mind your baseline*, experiments except for the number of epochs and scheduling. We consider 5 independent runs that match the number of samples *per step* (i.e., the batch size is exactly the same for all models after considering techniques such as the multistart and symmetric baselines). For AM Rollout, we employ half the batch size of other models since it requires double the number of evaluations due to its baseline.

**Search Methods Experiments**  For these experiments, we employ the same models trained in the in-distribution benchmark on 50 nodes. For Active Search (AS), we run 200 iterations for each instance and an augmentation size of 8. The Adam optimizer is used with a learning rate of $2.6 \times 10^{-4}$ and weight decay of $10^{-6}$. For Efficient Active Search, we benchmark EAS-Lay (with an added layer during the single-head computation, `PointerAttention` in our code) with the original hyperparameters proposed by Hottung et al. [50]. The learning rate is set to $0.0041$ and weight decay to $10^{-6}$. The search is restricted to 200 iterations with dihedral augmentation of 8 as well as imitation learning weight $\lambda = 0.013$.

Testing is performed on 100 instances on both TSP and CVRP for $N \in [200, 500, 1000]$, generated with the usual random seed for testing 1234.

### D.3.4  Generalization: Cross-Task and Cross-Distribution

In addition to training on uniformly distributed instances, as is standard for POMO [76], we further train POMO [76] on a mixture of multiple distributions (i.e., the exemplar distributions defined in [16]) and multiple VRP tasks (i.e., CVRP, OVRP, VRPL, VRPB, VRPTW, and OVRPTW, as defined in [89, 157, 13]) with fixed problem size $N = 50$, termed as MDPOMO and MTPOMO, respectively. Note that all the models in Table 4 undergo training across 10,000 epochs, each with a batch size of 512 and 10,000 training instances. The other training setups are consistent with the previous work [76]. The whole training time is within one day. During inference, we evaluate their generalization performance on the benchmark datasets in CVRPLib [86] using greedy rollout with $8\times$ instance augmentation and multiple start nodes following Kwon et al. [76].

### D.3.5  Large-Scale Instances

The GLOP [152] models' global policy are trained on random instances of CVRP1K and CVRP2K, respectively. Both models are trained for 100 epochs, with each epoch comprising 1000 instances. To accelerate the training process, random insertion is utilized as the sub-TSP solver.

For the experiment results presented in Table 5, we evaluate our implementation using the identical instances and setup as those utilized in Ye et al. [152]. The AM revisers involved are directly adopted from Ye et al. [152]. Table 13 reports the generalization performance of the CVRP2K model on 100 CVRP10K instances and 24 CVRP20K instances. These test instances are generated following the procedure in Nazari et al. [106], with the capacities fixed to 1000.

### D.3.6  Combining Construction and Improvement

To test the potential collaboration between constructive and improvement methods (in Appendix E.5 and Section 5.3), we recorded the performance of improvement methods during inference with initial solutions generated either randomly or by leveraging solutions generated greedily by constructive methods. This was done for both TSP and PDP with a fixed problem size of $N = 50$. We used a test set with 1,000 instances for both TSP and PDP and recorded the runtime for all constructive

---

[19]`https://stable-baselines.readthedocs.io/en/master/modules/ppo2.html`

1422 and improvement solvers based on an INTEL XEON GOLD 5317 CPU @ 3.00GHz and one RTX
1423 3090 GPU.

1424 For the constructive models to bootstrap improvement, we used the POMO and HAM (i.e. AM with
1425 rollout baseline, with HAM [82] encoder for construction PDP) directly from Appendix D.3.3. Note
1426 that these models were trained under controlled settings and could see a further boost in performance
1427 with further training. Moreover, while we used simple greedy evaluation, more complex evaluation
1428 schemes may be used, such as combining symmetric augmentation, multistart, or advanced sampling
1429 techniques as nucleus sampling.

1430 For the improvement models, we used both DACT and NeuOpt (with $K = 4$) for TSP, and the N2S
1431 model for PDP. Training for all models was conducted with 200 epochs and 20 batches per epoch,
1432 with a batch size of 512 for TSP and 600 for PDP. The n-step and maximum improvement steps for
1433 training were set to 4 and 200, respectively. Other hyperparameters such as learning rate, curriculum
1434 learning scaler, and gradient norm clip were set as per their original papers.

## D.4 Decoding Schemes

1436 Due to the limited space in the main paper, we further elaborate on the setup of the decoding schemes
1437 (or *strategies* in this section, shown in Fig. 16.

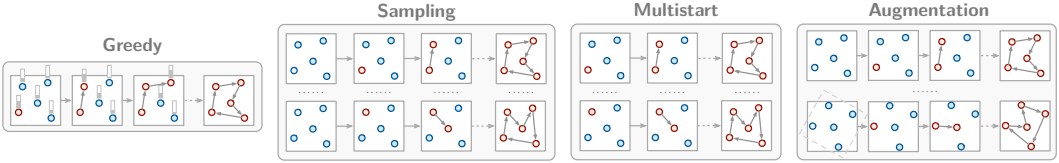

Figure 16: Inference methods we consider in RL4CO. These can also be combined together, such as greedy
multistart with augmentation.

### D.4.1 Augmentations

1439 In RL4CO, we consider as augmentations any transformation $\psi$ that maps an instance $x$
1440 into an instance $x'$ whose (optimal) solution should be the same or close to the original.
1441 Augmentations have been used in various domains, such as computer vision,
1442 where, for example, labels are invariant to rotations. Similarly, in Euclidean
1443 CO, one can apply the *dihedral transformation* of Table 9 to generate a
1444 new instance whose solution is the same as the original one, composed of
1445 4 rotations and 2 flips for a total of $\times 8$ transformation (which is the default
1446 used in POMO-based models as Kwon et al. [76], Liu et al. [89], Zhou et al.
1447 [157]. As introduced in Kim et al. [69] , one may additionally use any angle
1448 $\theta$ to perform a symmetric transformation as follows:

Table 9: Dihedral
transformations [76].

| $\psi(x,y)$ | |
|---|---|
| $(x, y)$ | $(y, x)$ |
| $(x, 1\text{-}y)$ | $(y, 1\text{-}x)$ |
| $(1\text{-}x, y)$ | $(1\text{-}y, x)$ |
| $(1\text{-}x, 1\text{-}y)$ | $(1\text{-}y, 1\text{-}x)$ |

$$\begin{pmatrix} x' \\ y' \end{pmatrix} = \psi(x, y) = \begin{pmatrix} x \cos\theta & -y \sin\theta \\ x \sin\theta & +y \cos\theta \end{pmatrix}$$

1449 where $\theta \in [0, 2\pi]$. Interestingly, we found that, generally, the dihedral augmentation is worse in
1450 terms of sample efficiency compared to randomly augmenting by sampling a $\theta$ value. We note that
1451 other augmentations are possible, including dilation [7] (i.e., rescaling) and possibly new ones such
1452 as *jittering*, which may have a broader application than Euclidean CO.

### D.4.2 Sampling

1454 In most NCO approaches, sampling is performed by simply increasing the evaluation budget but
1455 without additional modifications that can be important for better performance. We include the fol-
1456 lowing techniques in RL4CO: 1) *Sampling with Softmax Temperature*, 2) *Top-k Sampling* and 3)
1457 *Top-p Sampling*, visualized in Fig. 17.

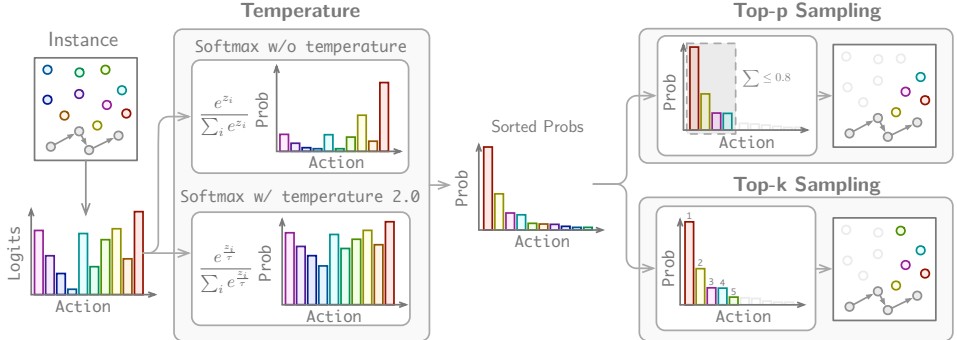

Figure 17: Sampling techniques implemented in RL4CO.

**Sampling with Softmax Temperature** Sampling with softmax temperature is a technique used to control the randomness of the sampling process. The temperature parameter $\tau$ is introduced to the softmax function, which converts the logits $z$ into a probability distribution:

$$p_i = \frac{\exp(z_i/\tau)}{\sum_{j=1}^{N} \exp(z_j/\tau)} \tag{35}$$

where $p_i$ is the probability of selecting the $i$-th action, $z_i$ is the corresponding logit, and $N$ is the total number of actions. A higher temperature $\tau > 1$ makes the distribution more uniform, increasing the chances of selecting less likely actions. Conversely, a lower temperature $0 < \tau < 1$ makes the distribution sharper, favoring the most likely actions.

**Top-k Sampling** Top-k sampling is a method that restricts the sampling space to the $k$ most likely actions. Given the logits $z$, the top-k actions with the highest probabilities are selected, and the probabilities of the remaining actions are set to zero. The probability distribution is then renormalized over the selected actions:

$$p_i = \begin{cases} \frac{\exp(z_i/\tau)}{\sum_{j \in \mathcal{T}_k} \exp(z_j/\tau)} & \text{if } i \in \mathcal{T}_k \\ 0 & \text{otherwise} \end{cases} \tag{36}$$

where $\mathcal{T}_k$ is the set of indices corresponding to the top-k actions. Top-k sampling helps to eliminate the possibility of generating low-probability actions, improving the quality and coherence of the generated output. We note that, however, in CO problems, it may not be as straightforward as in large language models to select the $k$ parameter since neighborhoods and distributions are not homogeneous.

**Top-p Sampling** Top-p sampling, also known as nucleus sampling, is an alternative to top-k sampling that dynamically adjusts the number of actions considered for sampling based on a probability threshold $p$ [48]. The actions are sorted by their probabilities in descending order, and the cumulative probability is calculated. The sampling space is then restricted to the smallest set of actions whose cumulative probability exceeds the threshold $p$:

$$\mathcal{T}_p = \left\{ i : \sum_{j=1}^{i} p_j \leq p \right\} \tag{37}$$

where $\mathcal{T}_p$ is the set of indices corresponding to the actions included in the top-p sampling. The probabilities of the actions in $\mathcal{T}_p$ are renormalized, while the probabilities of the remaining actions are set to zero:

$$p_i = \begin{cases} \frac{\exp(z_i/\tau)}{\sum_{j \in \mathcal{T}_p} \exp(z_j/\tau)} & \text{if } i \in \mathcal{T}_p \\ 0 & \text{otherwise} \end{cases} \tag{38}$$

Top-p sampling provides a more dynamic way to control the diversity and quality of the generated output compared to top-k sampling. In CO, this is also a more structured way of performing training or evaluation since top-p sampling is agnostic of the number of nodes, unlike top-k sampling.

# E    Additional Experiments

## E.1    Mind your Baseline: Further Insights

**Benchmark Setup**   We focus on benchmarking the AR routing NCO solvers under controlled settings, aiming to compare all benchmarked methods as closely as possible in terms of network architectures and the number of training samples consumed.

**Models**   We evaluate the following NCO solvers: 1) *AM* [74] with rollout baseline, 2) *POMO* [76] with the shared baseline to train AM instead of the rollout baseline; we also use six MHA layers and InstanceNorm instead of BatchNorm according to the original implementation, 3) *Sym-NCO* [69] utilizes the symmetric baseline to train AM instead of the rollout baseline and the same encoder as POMO, 4) *AM-XL* is an AM model that adopts *POMO*-style MHA encoder, and trained on the same number of samples as POMO, with the goal of seeing whether training for longer, as done in POMO, can significantly improve the results 5) *A2C*, i.e. AM trained with Advantage Actor-Critic (A2C), 6) *AM-PPO* trained via the Proximal Policy Optimization (PPO, Schulman et al. [119]) algorithm and finally 7) Polynet [51] with shared baseline and setting $K = n$.

For fairness of comparison, we try to match the number of training steps to be the same and adjust the batch size accordingly. Specifically, we train models for 100 epochs as in Kool et al. [74] using the Adam optimizer [71] with an initial learning rate (LR) of 0.001 with a decay factor of 0.1 after the 80th and 95th epochs[20]. We evaluate the trained solvers using the schemes shown in Fig. 16.

### E.1.1    Main In-distribution Results

We first measure the performances of NCO solvers on the same dataset distribution on which they are trained. We first observe that, counter to the commonly known trends that AM < POMO < Sym-NCO, the trends can change to decoding schemes and targeting CO problems. Especially when the solver decodes the solutions with *Augmentation* or *Greedy Multistart + Augmentation*, the performance differences among the benchmarked solvers on TSP and CVRP become less significant. Surprisingly, PolyNet performs well even in the greedy one-shot setting, despite its primary focus on generating diverse solutions. For decoding schemes that generate multiple solutions, PolyNet demonstrates strong performance across various problems. Particularly for decoding schemes without multistarts, PolyNet benefits significantly from its inherent diversity mechanism

We note that the original implementation of POMO [21] is not directly applicable to OP, PCTSP, and PDP. Adapting it to solve new problems is not straightforward due to the coupling between environment and policy implementations. However, owing to the flexibility of RL4CO, we successfully implemented POMO for OP and PCTSP. Our results indicate that POMO underperforms in OP and PCTSP; unlike TSP, CVRP, and PDP, where all nodes need to be visited, OP and PCTSP are not constrained to visit all nodes. Due to such differences, POMO's visiting all nodes strategy may not work as an effective inductive bias. Further, we benchmark the NCO solvers for PDP, which was not originally supported natively by each of the benchmarked solvers. We apply the environment embeddings and the Heterogeneous Attention Encoder from HAM [82] to the NCO models for encoding pickup

---

[20]We find that simple learning rate scheduling with `MultiStepLinear` can improve performance i.e., compared to the original AM implementation.

[21]https://github.com/yd-kwon/POMO

Table 10: In-distribution benchmark results for routing problems with 50 nodes. We report the gaps to the best-known solutions of classical heuristics solvers.

| Method | TSP | | | CVRP | | | OP | | | PCTSP | | | PDP | | |
|---|---|---|---|---|---|---|---|---|---|---|---|---|---|---|---|
| | Cost ↓ | Gap | Time | Cost ↓ | Gap | Time | Prize ↑ | Gap | Time | Cost ↓ | Gap | Time | Cost ↓ | Gap | Time |
| *Classical Solvers* | | | | | | | | | | | | | | | |
| *Gurobi* | 5.70 | 0.00% | 2m | – | – | – | – | – | – | – | – | – | – | – | – |
| *Concorde* | 5.70 | 0.00% | 2m | – | – | – | – | – | – | – | – | – | – | – | – |
| *HGS* | – | – | – | 10.37 | 0.00% | 10h | – | – | – | – | – | – | – | – | – |
| *Compass* | – | – | – | – | – | – | 16.17 | 0.00% | 5m | – | – | – | – | – | – |
| *LKH3* | 5.70 | 0.00% | 5m | 10.38 | 0.10% | 12h | – | – | – | – | – | – | 6.86 | 0.00% | 1h30m |
| *OR Tools* | 5.80 | 1.83% | 5m | – | – | – | – | – | – | 4.48 | 0.00% | 5h | 7.36 | 7.29% | 2h |
| *Greedy One Shot Evaluation* | | | | | | | | | | | | | | | |
| A2C | 5.83 | 2.22% | (<1s) | 11.16 | 7.09% | (<1s) | 14.77 | 8.64% | (<1s) | 5.15 | 14.96% | (<1s) | 7.52 | 9.90% | (<1s) |
| AM | 5.78 | 1.41% | (<1s) | 10.95 | 5.30% | (<1s) | 15.46 | 4.40% | (<1s) | 4.59 | 2.46% | (<1s) | 7.51 | 9.88% | (<1s) |
| POMO | 5.75 | 0.89% | (<1s) | 10.80 | 3.99% | (<1s) | 13.86 | 14.26% | (<1s) | 5.00 | 11.61% | (<1s) | 7.59 | 10.64% | (<1s) |
| Sym-NCO | 5.72 | 0.47% | (<1s) | 10.87 | 4.61% | (<1s) | 15.67 | 3.09% | (<1s) | 4.52 | 2.12% | (<1s) | 7.39 | 7.73% | (<1s) |
| AM-XL | 5.73 | 0.54% | (<1s) | 10.84 | 4.31% | (<1s) | 15.69 | 2.98% | (<1s) | 4.53 | 2.44% | (<1s) | 7.31 | 6.56% | (<1s) |
| AM-PPO | 5.76 | 0.92% | (<1s) | 10.87 | 4.60% | (<1s) | 15.67 | 3.05% | (<1s) | 4.55 | 2.45% | (<1s) | 7.43 | 8.31% | (<1s) |
| PolyNet | 5.72 | 0.68% | 2s | 10.81 | 4.24% | 2s | 15.70 | 2.93% | 2s | 4.54 | 2.45% | 2s | 8.26 | 3.46% | 2s |
| *Sampling with width $M = 1280$* | | | | | | | | | | | | | | | |
| A2C | 5.74 | 0.72% | 40s | 10.70 | 3.07% | 1m24s | 15.14 | 6.37% | 57s | 4.96 | 10.71% | 57s | 7.32 | 6.70% | 1m15s |
| AM | 5.72 | 0.40% | 40s | 10.60 | 2.22% | 1m24s | 15.90 | 1.68% | 48s | 4.52 | 0.99% | 57s | 7.25 | 5.69% | 1m15s |
| POMO | 5.71 | 0.18% | 1m | 10.54 | 1.64% | 2m30s | 14.62 | 9.56% | 1m10s | 4.82 | 7.59% | 1m23s | 7.31 | 6.56% | 1m50s |
| Sym-NCO | 5.70 | 0.14% | 1m | 10.58 | 2.03% | 2m30s | 16.02 | 0.93% | 1m10s | 4.52 | 0.82% | 1m23s | 7.17 | 4.52% | 1m50s |
| AM-XL | 5.71 | 0.17% | 1m | 10.57 | 1.91% | 2m30s | 15.97 | 1.25% | 1m10s | 4.52 | 0.88% | 1m23s | 7.15 | 4.23% | 1m50s |
| AM-PPO | 5.70 | 0.15% | 40s | 10.52 | 1.52% | 1m24s | 16.04 | 0.78% | 48s | 4.48 | 0.18% | 57s | 7.17 | 4.52% | 1m15s |
| PolyNet | 5.70 | 0.15% | 1m20s | 10.42 | 0.53% | 2m40s | 16.08 | 0.52% | 1m15s | 4.47 | 0.13% | 2m15s | 6.93 | 0.81% | 2m10s |
| *Greedy Multistart ($N$)* | | | | | | | | | | | | | | | |
| A2C | 5.80 | 1.81% | 2s | 10.90 | 4.86% | 6s | 14.61 | 9.65% | 4s | 5.12 | 14.29% | 5s | 7.54 | 9.85% | 4s |
| AM | 5.77 | 1.21% | 2s | 10.73 | 3.39% | 6s | 15.71 | 2.84% | 4s | 4.56 | 1.89% | 5s | 7.46 | 8.75% | 4s |
| POMO | 5.72 | 0.29% | 3s | 10.58 | 2.04% | 8s | 13.95 | 13.71% | 7s | 4.98 | 11.16% | 7s | 7.46 | 8.75% | 6s |
| Sym-NCO | 5.72 | 0.36% | 3s | 10.71 | 3.17% | 8s | 15.88 | 1.79% | 7s | 4.55 | 1.59% | 7s | 7.38 | 7.58% | 6s |
| AM-XL | 5.72 | 0.42% | 3s | 10.68 | 2.88% | 8s | 15.85 | 1.95% | 7s | 4.56 | 1.79% | 7s | 7.25 | 5.69% | 6s |
| AM-PPO | 5.74 | 0.61% | 2s | 10.67 | 2.72% | 6s | 15.98 | 1.21% | 4s | 4.53 | 1.18% | 5s | 7.23 | 5.39% | 4s |
| PolyNet | 5.70 | 0.25% | 3s | 10.52 | 1.42% | 18s | 16.05 | 0.71% | 3s | 4.54 | 1.31% | 10s | 7.18 | 4.65% | 5s |
| *Greedy with Augmentation (1280)* | | | | | | | | | | | | | | | |
| A2C | 5.71 | 0.18% | 40s | 10.63 | 2.49% | 1m24s | 14.89 | 7.91% | 48s | 5.15 | 14.96% | 1m | 7.03 | 2.46% | 1m15s |
| AM | 5.70 | 0.07% | 40s | 10.53 | 1.56% | 1m24s | 15.88 | 1.79% | 48s | 4.59 | 2.46% | 1m | 7.14 | 4.08% | 1m15s |
| POMO | 5.70 | 0.06% | 1m | 10.55 | 1.72% | 2m30s | 14.23 | 11.97% | 1m15m | 5.09 | 13.61% | 1m42s | 7.15 | 4.23% | 1m45s |
| Sym-NCO | 5.70 | 0.01% | 1m | 10.53 | 1.54% | 2m30s | 15.94 | 1.41% | 1m15m | 4.58 | 2.17% | 1m42s | 7.03 | 2.48% | 1m45s |
| AM-XL | 5.70 | 0.01% | 1m | 10.52 | 1.47% | 2m30s | 15.90 | 1.66% | 1m15m | 4.59 | 2.54% | 1m42s | 6.98 | 1.75% | 1m45s |
| AM-PPO | 5.70 | 0.15% | 40s | 10.52 | 1.52% | 1m24s | 16.01 | 0.84% | 48s | 4.48 | 0.18% | 1m | 7.00 | 2.04% | 1m15s |
| PolyNet | 5.70 | 0.17% | 1m30s | 10.47 | 0.92% | 3m | 16.05 | 0.72% | 2m | 4.47 | 0.10% | 2m10s | 6.94 | 1.20% | 2m15s |
| *Greedy Multistart with Augmentation ($N \times 16$)* | | | | | | | | | | | | | | | |
| A2C | 5.72 | 0.41% | 32s | 10.67 | 2.81% | 1m | 15.22 | 5.88% | 30s | 5.06 | 12.94% | 35s | 7.10 | 3.51% | 50s |
| AM | 5.71 | 0.21% | 1m | 10.55 | 1.73% | 1m | 16.05 | 0.76% | 30s | 4.54 | 1.28% | 35s | 7.10 | 3.50% | 50s |
| POMO | 5.70 | 0.05% | 48s | 10.48 | 1.11% | 2m | 15.05 | 6.94% | 1m | 4.92 | 9.81% | 1m10s | 7.12 | 3.79% | 1m25s |
| Sym-NCO | 5.70 | 0.03% | 48s | 10.54 | 1.63% | 2m | 16.02 | 0.51% | 1m | 4.53 | 1.17% | 1m10s | 7.01 | 2.19% | 1m25s |
| AM-XL | 5.70 | 0.04% | 48s | 10.53 | 1.50% | 2m | 16.08 | 0.57% | 1m | 4.54 | 1.25% | 1m10s | 7.00 | 2.04% | 1m25s |
| AM-PPO | 5.70 | 0.03% | 32s | 10.51 | 1.45% | 1m | 16.09 | 0.49% | 30s | 4.49 | 0.89% | 35s | 6.98 | 1.75% | 50s |
| PolyNet | 5.70 | 0.15% | 1m | 10.41 | 0.36% | 2m16s | 16.11 | 0.37% | 1m24s | 4.49 | 0.24% | 1m35s | 7.02 | 2.33% | 1m50s |

and delivery pairs, further emphasizing RL4CO's flexibility. We observe that AM-XL, which employs the same RL algorithm as AM but features the encoder architecture of POMO and is trained with an equivalent number of samples, yields performance comparable to NCO solvers using more sophisticated baselines. This suggests that careful controls on architecture and the number of training samples are required when evaluating NCO solvers. We also re-implemented PointerNetworks [139, 8], but we excluded them from the main table due to their poor performance, i.e., more than 4% optimality gap in TSP50.

Table 10 and Table 11 show detailed results for 50 and 20 nodes, respectively.

### E.1.2 Decoding Schemes Comparison

During inference, investing more computational resources (i.e., sampling more), the trained NCO solver can discover improved solutions. We examine the performance gains achieved with varying numbers of samples. As shown in Fig. 18, the *Augmentation* decoding scheme achieves the Pareto front with limited samples and, notably, generally outperforms other decoding schemes. We note that while sampling with a light decoder can be more efficient in terms of speed than sampling, this may not be true for heavy-decoder [93] or decoder-only models [37, 94, 112], where decoding via greedy augmentations may help improve performance.

Table 11: In-distribution results for models trained on 20 nodes.

| Method | TSP | | | CVRP | | | OP | | | PCTSP | | | PDP | | |
|---|---|---|---|---|---|---|---|---|---|---|---|---|---|---|---|
| | Cost ↓ | Gap | Time | Cost ↓ | Gap | Time | Prize ↑ | Gap | Time | Cost ↓ | Gap | Time | Cost ↓ | Gap | Time |
| *Classical Solvers* | | | | | | | | | | | | | | | |
| *Gurobi*[†] | 3.84 | 0.00% | 7s | – | – | – | – | – | – | – | – | – | – | – | – |
| *Concorde* | 3.84 | 0.00% | 1m | – | – | – | 5.39 | 0.00% | 16m | 3.13 | 0.00% | 2m | – | – | – |
| *HGS* | – | – | – | 6.13 | 0.00% | 4h | – | – | – | – | – | – | – | – | – |
| *Compass* | – | – | – | – | – | – | – | – | – | – | – | – | – | – | – |
| *LKH3* | 3.84 | 0.00% | 15s | 6.14 | 0.16% | 5h | – | – | – | – | – | – | – | – | – |
| *OR Tools* | 3.85 | 0.37% | 1m | – | – | – | – | – | – | 3.13 | 0.00% | 5h | 4.70 | 3.16% | 1h |
| *CPLEX* | – | – | – | – | – | – | – | – | – | – | – | – | 4.56 | 0.00% | 7m23s |
| *Greedy One Shot Evaluation* | | | | | | | | | | | | | | | |
| A2C | 3.86 | 0.64% | (<1s) | 6.46 | 5.00% | (<1s) | 5.01 | 6.70% | (<1s) | 3.36 | 7.35% | (<1s) | 4.71 | 3.31% | (<1s) |
| AM | 3.84 | 0.19% | (<1s) | 6.39 | 3.92% | (<1s) | 5.20 | 3.17% | (<1s) | 3.17 | 1.28% | (<1s) | 4.82 | 5.70% | (<1s) |
| POMO | 3.84 | 0.18% | (<1s) | 6.33 | 3.00% | (<1s) | 4.69 | 12.69% | (<1s) | 3.41 | 8.95% | (<1s) | 4.85 | 6.36% | (<1s) |
| Sym-NCO | 3.84 | 0.05% | (<1s) | 6.30 | 2.58% | (<1s) | 5.30 | 1.37% | (<1s) | 3.15 | 0.64% | (<1s) | 4.70 | 3.07% | (<1s) |
| AM-XL | 3.84 | 0.07% | (<1s) | 6.31 | 2.81% | (<1s) | 5.25 | 2.23% | (<1s) | 3.17 | 1.26% | (<1s) | 4.71 | 3.29% | (<1s) |
| PolyNet | 3.84 | 0.10% | (<1s) | 6.40 | 4.44% | (<1s) | 5.26 | 2.28% | (<1s) | 3.18 | 1.98% | (<1s) | 4.69 | 2.92% | (<1s) |
| *Sampling with width $M = 1280$* | | | | | | | | | | | | | | | |
| A2C | 3.84 | 0.15% | 20s | 6.26 | 2.08% | 24s | 5.12 | 4.66% | 22s | 3.28 | 4.79% | 23s | 4.64 | 1.76% | 23s |
| AM | 3.84 | 0.04% | 20s | 6.24 | 1.78% | 24s | 5.30 | 1.30% | 22s | 3.15 | 0.78% | 23s | 4.66 | 2.19% | 23s |
| POMO | 3.84 | 0.02% | 36s | 6.20 | 1.06% | 40s | 4.90 | 8.83% | 37s | 3.33 | 6.39% | 39s | 4.68 | 2.63% | 39s |
| Sym-NCO | 3.84 | 0.01% | 36s | 6.22 | 1.44% | 40s | 5.34 | 0.59% | 37s | 3.14 | 0.35% | 39s | 4.64 | 1.75% | 39s |
| AM-XL | 3.84 | 0.02% | 36s | 6.22 | 1.46% | 40s | 5.32 | 0.93% | 37s | 3.15 | 0.56% | 39s | 4.64 | 1.75% | 39s |
| PolyNet | 3.84 | 0.00% | 47s | 6.14 | 0.23% | 1m15s | 5.35 | 0.52% | 37s | 3.13 | 0.15% | 1m15s | 4.59 | 0.57% | 1m36s |
| *Greedy Multistart ($N$)* | | | | | | | | | | | | | | | |
| A2C | 3.85 | 0.36% | (<1s) | 6.33 | 3.04% | 3s | 5.06 | 5.77% | 2s | 3.30 | 5.18% | 2s | 4.85 | 6.42% | 2s |
| AM | 3.84 | 0.12% | (<1s) | 6.28 | 2.27% | 3s | 5.24 | 2.42% | 2s | 3.16 | 0.95% | 2s | 4.67 | 2.41% | 2s |
| POMO | 3.84 | 0.05% | (<1s) | 6.21 | 1.27% | 4s | 4.76 | 11.32% | 3s | 3.35 | 7.03% | 4s | 4.66 | 2.19% | 4s |
| Sym-NCO | 3.84 | 0.03% | (<1s) | 6.22 | 1.48% | 4s | 5.32 | 0.87% | 3s | 3.15 | 0.62% | 4s | 4.69 | 2.85% | 4s |
| AM-XL | 3.84 | 0.05% | (<1s) | 6.22 | 1.38% | 4s | 5.29 | 1.49% | 3s | 3.15 | 0.64% | 4s | 4.65 | 1.97% | 4s |
| PolyNet | 3.84 | 0.01% | 1s | 6.17 | 0.71% | 5s | 5.34 | 0.58% | 1s | 3.15 | 0.76% | 5s | 4.81 | 5.43% | 5s |
| *Greedy with Augmentation (1280)* | | | | | | | | | | | | | | | |
| A2C | 3.84 | 0.01% | 20s | 6.22 | 1.35% | 24s | 5.04 | 6.10% | 22s | 3.33 | 6.39% | 23s | 4.61 | 1.11% | 23s |
| AM | 3.84 | 0.00% | 20s | 6.20 | 1.07% | 24s | 5.25 | 2.25% | 22s | 3.16 | 0.96% | 23s | 4.63 | 1.54% | 23s |
| POMO | 3.84 | 0.00% | 36s | 6.18 | 0.84% | 45s | 4.85 | 9.76% | 38s | 3.37 | 7.55% | 42s | 4.62 | 1.32% | 42s |
| Sym-NCO | 3.84 | 0.00% | 36s | 6.17 | 0.71% | 45s | 5.33 | 0.77% | 38s | 3.15 | 0.63% | 42s | 4.61 | 0.95% | 42s |
| AM-XL | 3.84 | 0.00% | 36s | 6.17 | 0.68% | 45s | 5.30 | 1.30% | 38s | 3.15 | 0.68% | 42s | 4.61 | 0.96% | 42s |
| PolyNet | 3.84 | 0.00% | 55s | 6.16 | 0.48% | 1m10s | 5.35 | 0.50% | 57s | 3.13 | 0.16% | 1m2s | 4.59 | 0.58% | 1m10s |
| *Greedy Multistart with Augmentation ($N \times 16$)* | | | | | | | | | | | | | | | |
| A2C | 3.84 | 0.01% | 9s | 6.20 | 1.12% | 48s | 5.20 | 3.17% | 32s | 3.28 | 4.95% | 25s | 4.75 | 4.06% | 23s |
| AM | 3.84 | 0.00% | 9s | 6.18 | 0.78% | 48s | 5.34 | 0.56% | 32s | 3.14 | 0.32% | 25s | 4.63 | 1.52% | 23s |
| POMO | 3.84 | 0.00% | 13s | 6.16 | 0.50% | 1m | 5.09 | 5.29% | 45s | 3.35 | 6.95% | 38s | 4.61 | 1.10% | 42s |
| Sym-NCO | 3.84 | 0.00% | 13s | 6.17 | 0.61% | 1m | 5.35 | 0.39% | 45s | 3.14 | 0.24% | 38s | 4.60 | 0.89% | 42s |
| AM-XL | 3.84 | 0.00% | 13s | 6.16 | 0.44% | 1m | 5.35 | 0.46% | 45s | 3.14 | 0.28% | 38s | 4.60 | 0.87% | 42s |
| PolyNet | 3.84 | 0.00% | 18s | 6.14 | 0.16% | 1m20s | 5.37 | 0.31% | 1m | 3.13 | 0.12% | 58s | 4.61 | 1.03% | 55s |

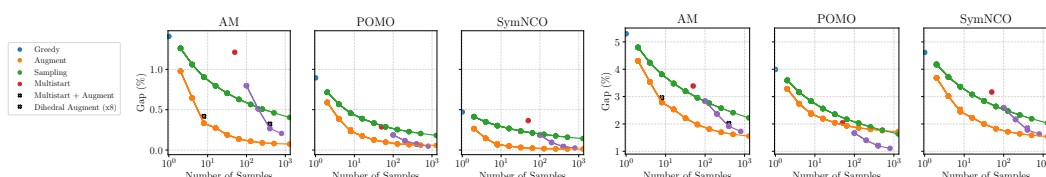

Figure 18: Pareto front of decoding schemes by the number of samples. Left: TSP50; right: CVRP50.

### E.1.3 Sample Efficiency

We additionally evaluate the NCO solvers based on the number of training samples (i.e., the number of reward evaluations). As shown in Fig. 19, we found that actor-critic methods (e.g., A2C and PPO) can exhibit efficacy in scenarios with limited training samples, as demonstrated by the TSP50/100 results in Fig. 19. This observation suggests that NCO solvers with control over the number of samples may exhibit a different trend in sample efficiency: if reward function evaluation is expensive, REINFORCE baselines that include additional reward function evaluations such as Greedy Rollout, POMO, and SymNCO may be sample-inefficient. While this is not the case for most CO problems (for instance: in routing, it is inexpensive to calculate routes), in other areas as Electronic Design Automation, where reward evaluation is resource-intensive due to the necessity of electrical simulations, in which sample efficiency can become even more crucial.

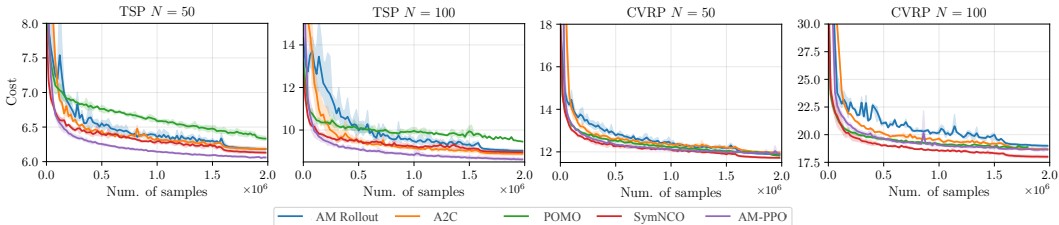

Figure 19: Validation cost curves and number of training samples consumed. Models with greater performance after full training may show worse convergence properties when the number of training samples is limited.

### E.1.4 Out-of-distribution

In this section, we evaluate the out-of-distribution performance of the NCO solvers by measuring the gap compared to the best-known solutions (BKS). The evaluation results are visualized in Fig. 20. Contrary to the in-distribution results, we find that NCO solvers with sophisticated baselines (i.e., POMO and Sym-NCO) tend to exhibit worse generalization when the problem size changes, either for solving smaller or larger instances. This can be seen as an indication of "overfitting" to the training sizes. On the other hand, variants of AM show relatively better generalization results overall.

Besides, we also evaluate the model by sampling decoding strategy with different temperatures as shown in Fig. 21, $k$ values for Top-$k$ as shown in Fig. 22, and $p$ values for Top-$p$ as shown in Fig. 23. A higher temperature or a lower $p$ value with Top-$p$ sampling can improve the generalization ability on large-scale problems, while Top-$k$ sampling has limited contribution to generalization cross problem sizes.

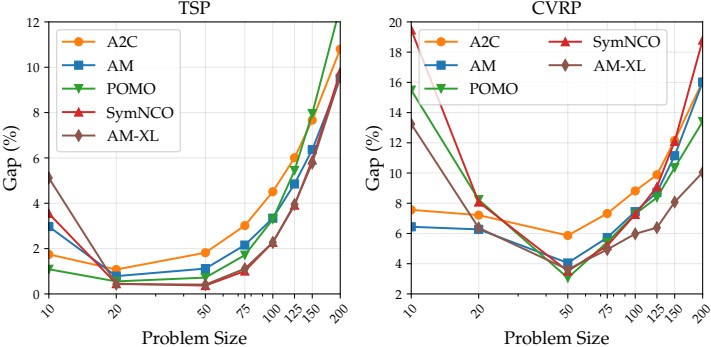

Figure 20: Out-of-distribution generalization by greedy decoding for models with different reinforce baselines trained on 50 nodes. Stronger performance in distribution does not always translate to out-of-distribution.

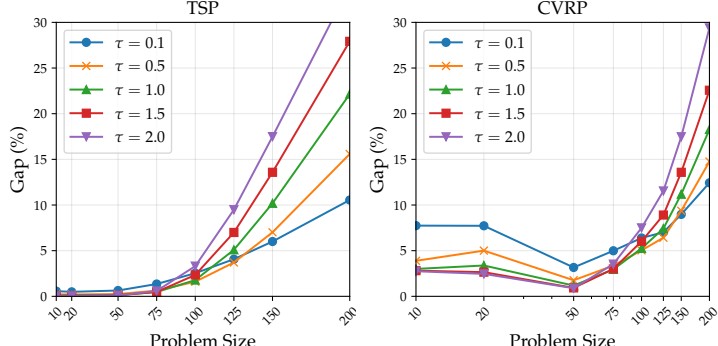

Figure 21: Out-of-distribution generalization by sampling with different temperatures $\tau$ for POMO trained on 50 nodes.

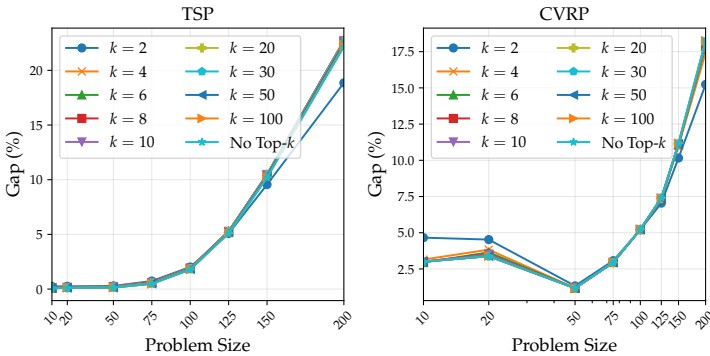

Figure 22: Out-of-distribution generalization by sampling with different Top-$k$ for POMO trained on 50 nodes.

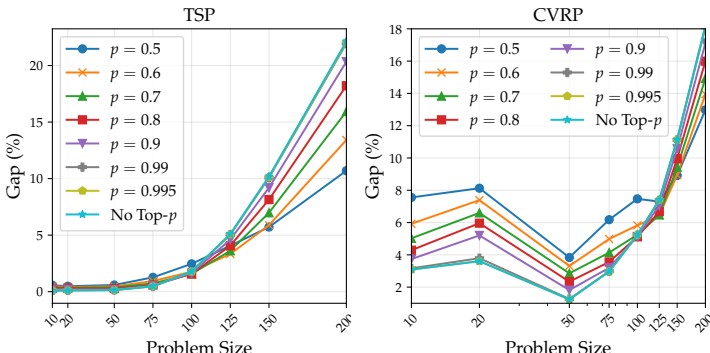

Figure 23: Out-of-distribution generalization by sampling with different Top-$p$ for POMO trained on 50 nodes.

### E.1.5 Search Methods

Table 12: Search Methods results of models pre-trained on 50 nodes. *Classic* refers to Concorde [35] for TSP and HGS [138, 144] for CVRP. OOM is "Out of Memory".

| Type | Metric | TSP | | | | | | CVRP | | | | | |
|---|---|---|---|---|---|---|---|---|---|---|---|---|---|
| | | POMO | | | Sym-NCO | | | POMO | | | Sym-NCO | | |
| | | 200 | 500 | 1000 | 200 | 500 | 1000 | 200 | 500 | 1000 | 200 | 500 | 1000 |
| *Classic* | Cost | 10.17 | 16.54 | 23.13 | 10.72 | 16.54 | 23.13 | 27.95 | 63.45 | 120.47 | 27.95 | 63.45 | 120.47 |
| *Zero-shot* | Cost | 13.15 | 29.96 | 58.01 | 13.30 | 29.42 | 56.47 | 29.16 | 92.30 | 141.76 | 32.75 | 86.82 | 190.69 |
| | Gap[%] | 29.30 | 81.14 | 150.80 | 24.07 | 77.87 | 144.14 | 4.33 | 45.47 | 17.67 | 17.17 | 36.83 | 58.29 |
| | Time[s] | 2.52 | 11.87 | 96.30 | 2.70 | 13.19 | 104.91 | 1.94 | 15.03 | 250.71 | 2.93 | 15.86 | 150.69 |
| *AS* | Cost | 11.16 | 20.03 | OOM | 11.92 | 22.41 | OOM | 28.12 | 63.98 | OOM | 28.51 | 66.49 | OOM |
| | Gap[%] | 4.13 | 21.12 | OOM | 11.21 | 35.48 | OOM | 0.60 | 0.83 | OOM | 2.00 | 4.79 | OOM |
| | Time[s] | 7504 | 10070 | OOM | 7917 | 10020 | OOM | 8860 | 21305 | OOM | 9679 | 24087 | OOM |
| *EAS* | Cost | 11.10 | 20.94 | 35.36 | 11.65 | 22.80 | 38.77 | 28.10 | 64.74 | 125.54 | 29.25 | 70.15 | 140.97 |
| | Gap[%] | 3.55 | 26.64 | 52.89 | 8.68 | 37.86 | 67.63 | 0.52 | 2.04 | 4.21 | 4.66 | 10.57 | 17.02 |
| | Time[s] | 348 | 1562 | 13661 | 376 | 1589 | 14532 | 432 | 1972 | 20650 | 460 | 2051 | 17640 |

A way to adapt to distribution changes is using *transductive RL*, commonly known as (active) search methods, which involve training (a part of) a pre-trained NCO solver to adapt to CO instances of interest. We evaluate 1) *Active Search (AS)* [8] which finetunes a pre-trained model on the searched instances by adapting all the policy parameters and 2) *Efficient Active Search (EAS)*: from [50] which finetunes a subset of parameters (i.e., embeddings or new layers) and adds an imitation learning loss to improve convergence.

We apply AS and EAS to POMO and Sym-NCO pre-trained on TSP and CVRP with 50 nodes to solve larger instances having $N \in [200, 500, 1000]$ nodes. As shown in Table 12, solvers with search methods improve the solution quality. However, POMO generally shows better improvements over

Sym-NCO. This suggests once more that the "overfitting" of sophisticated baselines can perform better in training distributions but eventually worse in different downstream tasks.

### E.1.6 Additional Large-scale Results

We also show in Table 13 additional large-scale results with $10k+$ nodes obtained with the hybrid AR/NAR GLOP model [152]. Fig. 24 demonstrates a solution obtained through our implementation of GLOP for CVRP35K. It represents the maximum scale of CVRP that RL4CO is capable of solving within 24GB of graphics memory while preserving the performance.

Table 13: Performance on large-scale CVRP instances with ten thousands of nodes.

|  | CVRP10K | | CVRP20K | |
| --- | --- | --- | --- | --- |
|  | Obj. | Time | Obj. | Time |
| HGS [138] | 108.1 | 4.01h | 182.7 | 6.03h |
| Random Insertion | 187.9 | 0.16s | 330.4 | 0.61s |
| GLOP-G (Insertion) | 127.0 | **2.42s** | 208.3 | **10.9s** |
| GLOP-G (AM) | 119.6 | 4.68s | 199.6 | 14.8s |
| GLOP-G (LKH) | **111.4** | 5.06s | **191.4** | 17.9s |

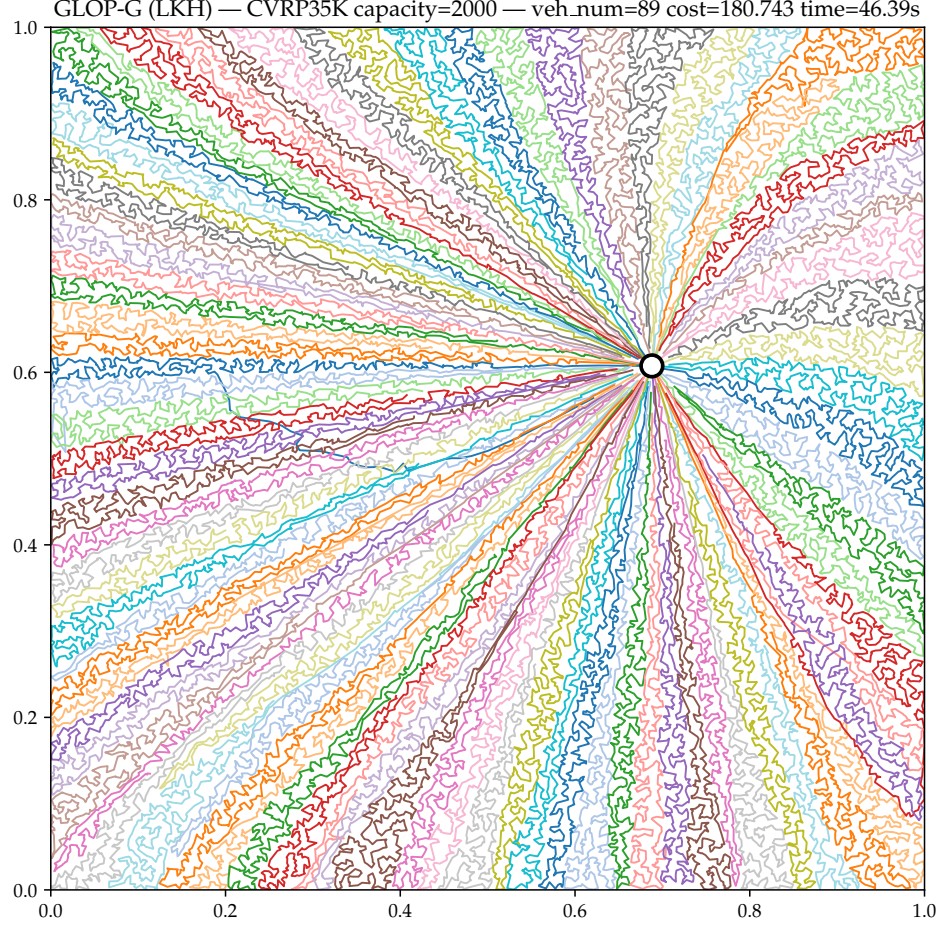

GLOP-G (LKH) — CVRP35K capacity=2000 — veh_num=89 cost=180.743 time=46.39s

Figure 24: A visualization of the solution generated by GLOP on CVRP35K.

Table 14: Benchmarking results of ACO method in TSP with 200, 500, 1000 nodes. The reported values are obtained by averaging over 128 test instances. The time is the average computation time for solving a single instance.

| Method | TSP200 | | | TSP500 | | | TSP1000 | | |
|---|---|---|---|---|---|---|---|---|---|
| | Cost | Gap(%) | Time(s) | Cost | Gap(%) | Time(s) | Cost | Gap(%) | Time(s) |
| *Concorde* [35] | 10.72 | 0.00 | 0.9 | 16.55 | 0.00 | 10.7 | 23.12 | 0.00 | 108.3 |
| *ACO* | 10.88 | 1.52 | 1.0 | 17.23 | 4.11 | 4.0 | 24.42 | 5.65 | 19.8 |
| *DeepACO* | 10.80 | 0.79 | 1.0 | 16.87 | 1.95 | 4.3 | 23.82 | 3.03 | 20.7 |
| *GFACS* | 10.75 | 0.32 | 1.0 | 16.80 | 1.56 | 4.3 | 23.78 | 2.87 | 20.7 |

Table 15: Benchmarking results of ACO methods with different $\tau$ values in TSP with 500 nodes. The reported values are the average cost of 128 test instances.

| Method | $\tau = 0.05$ | $\tau = 0.1$ | $\tau = 0.25$ | $\tau = 0.5$ | $\tau = 0.75$ | $\tau = 1.0$ | $\tau = 1.5$ | $\tau = 2.0$ |
|---|---|---|---|---|---|---|---|---|
| *ACO* | 17.05 | 16.95 | 17.03 | 17.11 | 17.19 | 17.23 | 17.26 | 17.26 |
| *DeepACO* | 17.00 | 16.97 | 16.92 | 16.84 | 16.85 | 16.87 | 16.88 | 16.89 |
| *GFACS* | 16.92 | 16.90 | 16.86 | 16.80 | 16.80 | 16.80 | 16.81 | 16.82 |

## E.2 Learning Heuristics for Ant Colony Optimization

### E.2.1 Experiment Settings

We adhered to the hyperparameters specified in the original papers for DeepACO [150] and GFACS [70] for GFlowNets training. We conducted two distinct benchmarks for ACO methods. The first benchmark evaluated the ability to solve the Traveling Salesman Problem (TSP) at different scales: 200, 500, and 1000. We use the test instances provided by DeepACO[22]. The second benchmark assessed inference capability at various temperature values of $\tau$ in TSP with 500 nodes. The temperature $\tau$ is a hyperparameter for the heatmap distribution of the heuristic matrix in ACO, where a low $\tau$ emphasizes exploitation and a high $\tau$ emphasizes exploration. For both experiments, the optimality gaps are calculated with respect to the average cost of solutions obtained using Concorde [35].

### E.2.2 Results

**TSP Benchmark**   Table 14 shows the results for the first benchmark. In this benchmark, we observed that GFACS outperforms other baselines, and DeepACO surpasses ACO. These results are consistent with their respective claims [150, 70], providing evidence that our benchmark is sufficiently valid. Notably, our algorithm also performed slightly faster than the original implementation, likely due to the batchified environment of RL4CO.

**Performance Comparison for Different Heatmap Temperatures ($\tau$)**   Table 15 shows the results for the second benchmark. This benchmark compared inference performance across different heatmap temperatures ($\tau$). We observed notable performance variation with changes in $\tau$. This highlights the importance of inference and sampling strategies even after deep network training is completed. Additionally, GFACS produced more consistent results with different $\tau$ values. This provides empirical evidence of the robustness of GFACS, which is due to its ability to model a sampler capable of generating diverse and high-reward solutions. The modularization of RL4CO allows for a focused study on inference capabilities, enabling future researchers to contribute to this aspect using the RL4CO pipeline.

---

[22] https://github.com/henry-yeh/DeepACO

### E.3 Learning to Schedule

Compared to routing problems, scheduling problems have not been extensively studied by the NCO community. On the one hand side, NCO methods for scheduling are harder to benchmark due to the absence of well-performing heuristics like the LKH algorithm for the TSP. On the other hand, scheduling problems involve more complex graph representations like disjunctive graphs [153], bipartite graphs [77], or heterogeneous graphs [125], making it harder to encode the problem. With RL4CO, we aim to mitigate these entry barriers for NCO researchers by providing established solution methods along with the environments. Further, by being modular by design, RL4CO allows for quick evaluation of different learning algorithms and network architectures, which can already lead to substantial improvements of the solution quality, as demonstrated in the example of the FJSSP in Table 2. Lastly, by providing benchmark instances like Taillard [129] and easy ways of initializing the environments with external benchmark files, we facilitate the comparison of models with existing methods. The following chapter describes established DRL models for scheduling problems as well as their performance on synthetic and benchmark datasets.

### E.3.1 JSSP

**Models**  To solve the JSSP using DRL methods, we implement the L2D model described in Appendix C.2.7 in RL4CO. To train the encoder-decoder policy, we use the same Proximal Policy Optimization (PPO) algorithm as Zhang et al. [153]. In contrast to most other work in the NCO domain, L2D uses a (dense) stepwise reward function rather than a sparse episodic reward, which is observed only after a complete solution is obtained. This reward determines the change in the lower bound of the makespan given the partial schedule. Due to the dense nature of the reward, the PPO algorithm for the scheduling problems evaluates actions on a stepwise basis, whereas environments with an episodic reward are evaluated based on a full rollout. We compare these methods and discuss the different implementations in Appendix E.3.4.

Further, we demonstrate RL4CO's ability to effortlessly implement a state-of-the-art solver for JSSP instances by exchanging the GCN encoder used by Zhang et al. [153] with the MatNet encoder [77] described in Appendix C.2.11. Furthermore, the greedy decoding scheme of Zhang et al. [153] is replaced by $N = 100$ random samples, of which the best is selected.

**Reproduction and Improvement of Original Results**  We demonstrate RL4CO's capability of learning dispatching rules for the JSSP by training and validating the L2D model of Zhang et al. [153] and our version of L2D with the MatNet encoder on synthetic data. We report the performance achieved with RL4CO together with the baselines the authors of the original papers used, as well as the solutions obtained via the CP-Sat solver Google OR-Tools. The baselines are a set of selected PDRs that have a high practical relevance, namely Most Work Remaining (MWKR) and Most Operations Remaining (MOR).

Table 16: Comparison of RL4CO with L2D [153] and other baselines on the JSSP. For OR-Tools, the fraction of instances solved optimally is reported in parentheses.

| Size | Metric | OR-Tools | PDRs | | L2D | RL4CO | |
|---|---|---|---|---|---|---|---|
| | | | MWKR | MOR | [153] | GCN | MatNet ($\times$128) |
| $6 \times 6$ | Obj. | 487.75 (100%) | 656.96 | 630.19 | 574.09 | 569.53 | 515.11 |
| | Gap | - | 34.6% | 29.2% | 17.7% | 16.8% | 5.6% |
| $10 \times 10$ | Obj. | 808.32 (100%) | 1151.41 | 1101.08 | 988.58 | 972.35 | 865.78 |
| | Gap | - | 42.6% | 36.5% | 22.3% | 20.3% | 7.1% |
| $15 \times 15$ | Obj. | 1187.06 (99%) | 1812.13 | 1693.33 | 1504.79 | 1492.94 | 1318.25 |
| | Gap | - | 52.6% | 42.6% | 26.7% | 25.7% | 11.0% |
| $20 \times 20$ | Obj. | 1555.79 (4%) | 2469.19 | 2263.68 | 2007.76 | 1992.36 | 1847.33 |
| | Gap | - | 58.6% | 45.5% | 29.0% | 28.1% | 18.7% |

The results are listed in Table 16. RL4CO's implementation of L2D manages to outperform the original implementation on all instance types, even when using the same model architecture, learning algorithm, and hyperparameters. The reason is that RL4CO uses an improved implementation of the environment. In the implementation of Zhang et al. [153] the state of the environment does not contain a time dimension. Instead, the environment schedules the selected operation at the earliest feasible start time, given the current schedule. Here, we use the environment proposed by Tassel et al. [132], where the environment transitions through distinct time steps $t = 0, 1, ...T$. In this case, the start time of a selected operation is set to the time step at which it was selected, leading to a more natural form of credit assignment.

Using the MatNet encoder instead of the GCN and employing a decoding scheme based on multiple random rollouts further reduces the makespan by a large margin. One instances of size $6 \times 6$, the gap to the optimal solutions was reduced by 11 percentage points to 5.6%, which corresponds to a third of the gap realized with the GCN encoder.

**Taillard Benchmark and out-of-distribution performance**   With RL4CO, we also provide the possibility to test models against established benchmarks. For the JSSP, a well-recognized benchmark is that of Taillard [129], which is also used by Zhang et al. [153] to validate their model. In Table 17, we report the results of RL4CO on these instances along with the results obtained by Zhang et al. [153] as well as the MOR and MWKR heuristics. We trained our MatNet models on JSSP instances up to size $20 \times 20$. For larger Taillard instances, we report the out-of-distribution performance to demonstrate the model's generalization ability. Similar to the synthetic test instances, our RL4CO implementation paired with the MatNet encoder manages to outperform the original L2D by large margins on all instances of the Taillard benchmark dataset, even when evaluating it on out-of-distribution instances.

Table 17: Results on the Taillard [129] benchmark instances. BKS refers to the best known solutions and % opt. specifies the rate of instances with optimal solutions. Values marked with a $^\dagger$ indicate out-of-distribution performance of the model trained on $20 \times 20$.

| Size | Metric | BKS | PDRs | | L2D | RL4CO |
| --- | --- | --- | --- | --- | --- | --- |
| | | | MWKR | MOR | [153] | MatNet ($\times 128$) |
| $15 \times 15$ | Obj. | 1230.06 (100%) | 1927.5 | 1782.3 | 1547.50 | 1404.30 |
| | Gap | - | 56.7% | 45.0% | 26.0% | 14.2% |
| $20 \times 15$ | Obj. | 1363.22 (90%) | 2190.7 | 2015.8 | 1774.7 | 1570.70 |
| | Gap | - | 60.7% | 47.7% | 30.0% | 15.2% |
| $20 \times 20$ | Obj. | 1617.60 (30%) | 2518.6 | 2309.9 | 2128.1 | 1842.90 |
| | Gap | - | 55.7% | 42.8% | 31.6% | 13.9% |
| $30 \times 15$ | Obj. | 1787.68 (70%) | 2728.0 | 2601.3 | 2378.8 | 2121.19$^\dagger$ |
| | Gap | - | 52.6% | 45.6% | 33.0% | 18.6% |
| $30 \times 20$ | Obj. | 1948.32 (0%) | 3193.3 | 2888.1 | 2603.9 | 2357.90$^\dagger$ |
| | Gap | - | 63.9% | 48.2% | 33.6% | 21.0% |

### E.3.2   FJSSP

**Model**   To solve the FJSSP using DRL methods, we implement the HGNN model described in Appendix C.2.10 in RL4CO and train it with the same PPO algorithm as L2D. Besides HGNN we also implement a second model which exchanges the encoder of HGNN with the MatNet encoder.

**Reproduction and Improvement of Original Results**   We compare the results obtained via RL4CO with those reported by Song et al. [125] and the baseline used by them. Also, Song et al. [125] use MWKR and MOR to benchmark their model as well as the OR-Tools solver. The results, which are obtained on a test set comprising of 100 randomly generated instances, are listed below in Table 18.

Similar to the JSSP, the HGNN implemented in RL4CO achieves better results than the original implementation, although both implementations use the same definition of the environment. However, in RL4CO, we use instance normalization [135] on the input variables as well as between consecutive HGNN layers, which we found to drastically stabilize the training process.

Again, we were able to enhance the quality of the solution further by simply exchanging the encoder with MatNet. Especially on the larger instances, the increased model complexity translates into much better model performance, with the solutions even surpassing OR-Tools on $20 \times 10$ instances.

Table 18: Comparison of RL4CO and HGNN [125] on the FJSSP. For OR-Tools, the fraction of instances solved optimally is reported in parentheses. Both RL4CO and [125] make use of random-rollouts for decoding.

| Size | Metric | OR-Tools | PDRs | | HGNN | RL4CO ($\times$128) | |
| | | | MWKR | MOR | [125] ($\times$128) | HGNN | MatNet |
|---|---|---|---|---|---|---|---|
| $10 \times 5$ | Obj. | 96.59 (15%) | 115.29 | 116.69 | 105.61 | 102.49 | 99.02 |
| | Gap | - | 19.4% | 20.9% | 9.4% | 6.1% | 2.5% |
| $20 \times 5$ | Obj. | 188.45 (0%) | 216.98 | 217.17 | 207.50 | 199.47 | 192.05 |
| | Gap | - | 15.2% | 15.3% | 10.1% | 5.8% | 1.9% |
| $15 \times 10$ | Obj. | 145.42 (5%) | 169.18 | 173.40 | 160.36 | 155.34 | 151.93 |
| | Gap | - | 16.3% | 19.3% | 10.3% | 6.8% | 4.5% |
| $20 \times 10$ | Obj. | 197.24 (0%) | 220.85 | 221.86 | 214.87 | 207.52 | 192.00 |
| | Gap | - | 11.9% | 12.53% | 9.0% | 5.2% | -2.7% |

**Out-of-distribution** In this section, we evaluate the out-of-distribution performance of the DRL models trained with RL4CO on FJSSP $20 \times 10$ instances, by evaluating them on smaller ($20 \times 5$ & $15 \times 10$) and larger ($30 \times 10$ & $40 \times 10$) instances. The results in Table 19 indicate that both HGNN and MatNet manage to generalize well to problems of different sizes. Despite being trained on smaller instances, the HGNN manages to close the performance gap when evaluated on larger instances, with gaps being as small as 3.7% for FJSSP $40 \times 10$ instances. And on FJSSP $20 \times 5$ instances, the average makespan increases by only 1.56 (0.8%) when using the model trained on FJSSP $20 \times 10$ instead of $20 \times 5$ instances. Again, the MatNet model shows superior performance compared to the other baselines and surpasses even the results obtained by OR-Tools on the larger instances. The within-distribution performance of MatNet, therefore, also translates to out-of-distribution instances, indicating that the complexity of the model results in a better generalization ability.

Table 19: Generalization performance of a policy trained on a $20 \times 10$ FJSSP instances on smaller and larger instances. We use 100 test instances per instance size. Gaps are reported with respect to the results of OR-Tools

| Size | Metric | OR-Tools | PDRs | | HGNN | RL4CO ($\times$128) | |
| | | | MWKR | MOR | [125] ($\times$128) | HGNN | MatNet |
|---|---|---|---|---|---|---|---|
| $20 \times 5$ | Obj. | 188.45 (0%) | 216.98 | 217.17 | 207.50 | 201.03 | 193.61 |
| | Gap | - | 15.2% | 15.3% | 10.1% | 6.7% | 2.7% |
| $15 \times 10$ | Obj. | 145.42 (5%) | 169.18 | 173.40 | 160.36 | 162.41 | 150.59 |
| | Gap | - | 16.3% | 19.3% | 10.3% | 11.7% | 3.5% |
| $30 \times 10$ | Obj. | 294.10 (0%) | 319.89 | 320.18 | 312.20 | 309.10 | 286.16 |
| | Gap | - | 8.8% | 8.9% | 6.1% | 5.1% | -2.7% |
| $40 \times 10$ | Obj. | 397.36 (0%) | 425.70 | 425.19 | 415.14 | 412.05 | 381.19 |
| | Gap | - | 7.1% | 7.0% | 4.4% | 3.7% | -4.1% |

### E.3.3 FFSP

**MatNet** To solve the FFSP using DRL, RL4CO implements the policy network described by Kwon et al. [77]. It uses separate policy networks for each stage of the FFSP. Each of the stage

networks employs the MatNet encoder described in Appendix C.2.11, which generates embeddings for jobs and machines using the processing times of the job-machine pairs of the respective stage. The decoder of the attention model [74] then utilizes the machine embeddings of the respective stage as query and the job embeddings as keys and values to compute the probability distribution over jobs.

**Results**  We use the same three instance types described by Kwon et al. [77] to evaluate our implementations of the FFSP environment and the policy network. The instances only differ in the number of jobs, which are set to 20, 50, and 100. We assume that there are $S = 3$ stages, and each stage has $M = 4$ machines. In the $k$th stage, the processing time of the job $j$ on the machine $m$ is given by $p_{jmk}$. Therefore, an instance of the problem is defined by three matrices ($P_1$, $P_2$, and $P_3$), specifying the processing time for each job-machine combination in that stage. We report the results obtained by RL4CO and compare them to those obtained by Kwon et al. [77] in Table 20. Other benchmarks used are the exact solver CPLEX (for which results can only be obtained for FFSP20 instances), the Shortest Job First (SJF) dispatching rule, as well as the evolutionary algorithms Particle Swarm Optimization (PSO), and Genetic Algorithm (GA). One can see that, using RL4CO, we are able to reproduce the results from the original paper.

Table 20: Comparison of RL4CO with the results reported in [77]. Gaps are reported with respect to the best known results.

| Instance | Matric | CPLEX (600s) | SJF | GA | PSO | [77] | RL4CO |
|----------|--------|--------------|------|------|------|------|-------|
| FFSP20 | Obj. | 36.6 | 31.3 | 30.6 | 29.1 | 27.3 | 27.2 |
|  | Gap | 34.5% | 15.0% | 12.5% | 6.9% | 0.3% | 0.0% |
| FFSP50 | Obj. | - | 57.0 | 56.4 | 55.1 | 51.5 | 51.6 |
|  | Gap | - | 10.7% | 9.5% | 7.0% | 0.0% | 0.2% |
| FFSP100 | Obj. | - | 99.3 | 98.7 | 97.3 | 91.5 | 91.3 |
|  | Gap | - | 8.8% | 8.1% | 6.6% | 0.2% | 0.0% |

### E.3.4   Dense and Episodic Rewards

We additionally compare dense and episodic rewards for the TSP and FJSSP environments, with similar training settings as in other experiments, except for the different reward functions.

Here, we compare the performance of the HGNN [125] in solving the FJSSP and AM [74] in solving the TSP when trained using a stepwise vs. an episodic reward. The results in Table 21 show that evaluating the FJSSP in a stepwise manner and stepwise re-encoding the current state significantly outperforms a policy based on a single, episodic reward. This is reasonable since the state of the FJSSP has many dynamic elements, and a policy that relies on a single encoder step may not fully grasp the problem dynamics. On the other hand, stepwise rewards for the TSP (AM model trained with POMO with the settings as Kwon et al. [76]) do not work well, and interestingly, performance approaches roughly that of the nearest insertion algorithms. Different CO problems react to the same learning setup, which again underpins the importance of a unified framework where different algorithms are implemented and are easily exchangeable.

Table 21: Comparison of dense (i.e. stepwise) and episodic rewards for the TSP and the FJSSP

| Reward | TSP | | | FJSSP | | |
|--------|-----|-----|------|---------------|---------------|-----------------|
|  | 20 | 50 | 100 | $10 \times 5$ | $20 \times 5$ | $15 \times 10$ |
| Dense | 4.51 | 7.05 | 9.80 | **102.49** | **199.47** | **155.34** |
| Episodic | **3.83** | **5.81** | **7.82** | 110.65 | 204.88 | 182.90 |

## E.4   Electronic Design Automation: Learning to Place Decaps

**Setup**   In this section, we benchmark models on the mDPP from Appendix B.3.2. We benchmark 3 variants of online DevFormer (DF), namely DF(PG,Critic): REINFORCE (where PG stands for Policy Gradients, an "alias" of the REINFORCE algorithm) with Critic baseline, DF(PG,Rollout): REINFORCE with Rollout baseline as well as PPO. All experiments are run with the same hyperparameters as the other experiments except for the batch size set to $64$, the maximum number of samples set to $10,000$, and a total of only $10$ epochs due to the nature of the benchmark sample efficiency.

### E.4.1   Main Results

Table 22 shows the main numerical results for the task when RS, GA, and DF models are trained for placing 20 decaps. While RS and GA need to take online shots to solve the problems (we restricted the number to $100$), DF models can successfully predict in a zero-shot manner and outperform the classical approaches. Interestingly, the vanilla critic-based method performed the worst, while our implementation of PPO almost matched the rollout policy gradients (PG) baseline; since extensive hyperparameter tuning was not performed, we expect PPO could outperform the rollout baseline given it requires fewer samples. Fig. 25 shows example renderings of the solved environment.

Table 22: Performance of different methods on the mDPP benchmark

| Method | # Shots | Score ↑ | |
| --- | --- | --- | --- |
| | | maxsum | maxmin |
| *Online Test Time Search* | | | |
| Random Search | 100 | 11.55 | 10.63 |
| Genetic Algorithm | 100 | 11.93 | 11.07 |
| *RL Pretraining & Zero Shot Inference* | | | |
| DF-(PG,Critic) | 0 | $10.89 \pm 0.63$ | $9.51 \pm 0.68$ |
| DF-(PPO) | 0 | $12.16 \pm 0.03$ | $11.17 \pm 0.11$ |
| DF-(PG,Rollout) | 0 | $12.21 \pm 0.01$ | $11.26 \pm 0.03$ |

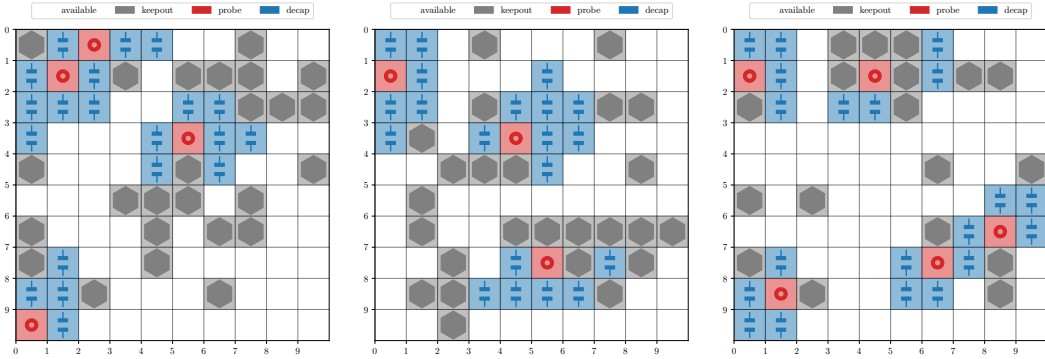

Figure 25: Renders of the environment with *maxmin* objective solved by DF-(PG,Rollout). The model successfully learned one main heuristic for DPP problems, which is that the optimal placement of decaps (blue) is generally close to probing ports (red).

### E.4.2   Generalization to Different Number of Components

In hardware design, the number of components is one major contribution to cost; ideally, one would want to use the least number of components possible with the best performance. In the DPP, increasing the number of decaps *generally* improves the performance at a greater cost, hence Pareto-efficient models are essential to identify. Fig. 26 shows the performance of DF models trained on

1743 20 decaps against the baselines. DF models PPO and PG-rollout can successfully generalize and are
1744 also Pareto-efficient with fewer decaps, important in practice for cost and material saving.

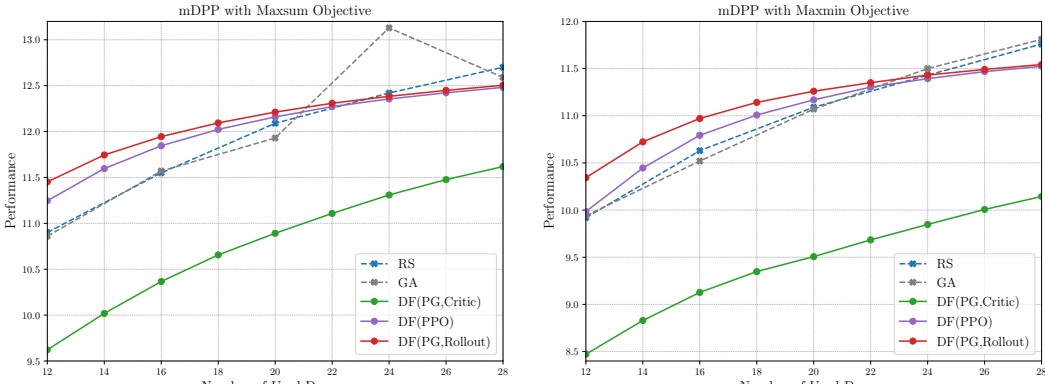

Figure 26: Performance vs number of used decaps for mDPP with *maxsum* objective [Left] and *maxmin* objective [Right].

## E.5 Learning to Improve

1746 In this section, we first show the efficiency of RL4CO when reproducing the improvement methods
1747 on the TSP and PDP with 50 nodes and discuss the potential collaboration of constructive methods
1748 with improvement methods for better inference performance.

### E.5.1 Main results

1750 As shown in Table 23, refactoring and implementing the three improvement methods—DACT [96]
1751 (TSP50), N2S [97] (PDP50), and NeuOpt [98] (PDP50)—using RL4CO consistently results in better
1752 efficiency compared to the original implementations. Specifically, training and testing times ($T =
1753 1,000$) are faster, and peak memory usage is lower. This advancement can be attributed to RL4CO's
1754 streamlined design, which uses a single tensor dictionary variable to store all state information, and
1755 the incorporation of efficient libraries like PyTorch Lightning and TorchRL. These enhancements
1756 demonstrate RL4CO's superior efficiency and ease of implementation.

Table 23: Comparison of time and memory usage for DACT [96] (TSP50), N2S [97] (PDP50), and NeuOpt [98] (PDP50) between the original implementation and the RL4CO implementation.

| | T_train (one epoch) | T_test (1k,1k) | Memory |
|---|---|---|---|
| DACT-Origin | 16m | 38s | 8069MB |
| DACT-RL4CO | **10m** | **26s** | **7135MB** |
| N2S-Origin | 26m | 41s | 13453MB |
| N2S-RL4CO | **17m** | **33s** | **12489MB** |
| NeuOpt-Origin | 14m | 37s | 7273MB |
| NeuOpt-RL4CO | **10m** | **31s** | **6313MB** |

### E.5.2 Discussion

1758 As shown in Fig. 27, bootstrapping improvement with constructive methods can greatly improve
1759 the performance, especially in terms of the Primal Integral (PI, Appendix D.1.2). While in TSP
1760 bootstrapping is consistently better than simply improving with default solutions (i.e. lower final gap
1761 to BKS as well as PI), we note that in PDP with N2S, improving starting from a random initialization
1762 can yield better performance in terms of gap. However, the PI reveals that while N2S from random

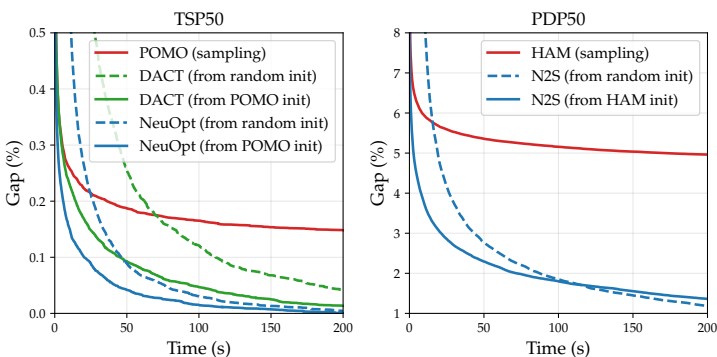

Figure 27: Bootstrapping improvement with constructive methods for TSP50 and PDP50.

init achieves a value of 5.580, N2S from HAM construction initialization achieves a much better 2.234, indicating a much better early convergence speed and Pareto front.

We additionally offer some clues on how to improve such performance. Firstly, we simply initialized from a greedy solution, while more complex inference strategies may offer a significant boost. Furthermore, the trained model as per the setting in Appendix D.3.3 could be further trained and obtain better performance. Importantly, we believe that *end-to-end construction & improvement*, in which both a constructive and improvement method are trained together, could ultimately outperform a separate training and achieve the best of both worlds.

### E.6 Graph Problems: Facility Location Problem (FLP) and Maximum Coverage Problem (MCP)

Here, we present the experimental results and the corresponding discussions on the two CO problems on graphs: the Facility Location Problem (FLP; see Appendix B.4.1) and the Maximum Coverage Problem (MCP; see Appendix B.4.2).

#### E.6.1 Experimental settings

**Baseline methods** We consider two simple baselines: uniform random (UR) and deterministic greedy (DG), where UR chooses $k$ locations uniformly at random and DG chooses $k$ locations one by one in a greedy manner. We also apply two MIP solvers, Gurobi [44] and SCIP [14], to obtain the optimal solutions.

**Benchmark methods** We benchmark with the attention model (AM) with different embedding models (i.e., encoders) and different RL baselines. For FLP, the considered embedding models are: the multilayer perception (MLP), the graph convolutional network (GCN) [72], and the graph attention network [137, 24]. For MCP, since the problem instances are formulated on bipartite graphs, the considered embedding models are: the multilayer perception (MLP), the GraphSAGE model [45] (in short "SAGE"), and the generalized GCN model [81] (in short "GEN"). The considered RL baselines are: Rollout, Mean, Exponential, and Critic. All the models are trained in 100 epochs. The learning rate is $1e-5$ for FLP and $1e-4$ for MCP. In each epoch, $100,000$ training data are used with batch size $1,000$. For the decoding strategies, we consider sampling (with 64 independent samples) and greedy. For sampling (and UR), we report both the "best" performance among the 64 independent samples and the "mean" (i.e., average) performance over the 64 independent samples.

**Test-time active search** We apply three variants of active search at test time: the original active search (AS) proposed by Bello et al. [8], efficient active search (EAS) proposed by Hottung et al. [50] with two variants: EAS-Emb that finetunes embeddings and EAS-Lay that finetunes new layers. We run all the active search variants for 100 iterations.

Table 24: Performance of different methods on the facility location problem (FLP) benchmark. For the performance, the smaller the better.

| Encoder | RL Baseline | Sample (Best) | Sample (Mean) | Greedy | AS | Active Search EAS-Emb | EAS-Lay |
|---|---|---|---|---|---|---|---|
| MLP | Rollout | 10.4895 | 11.0056 | 10.9980 | 10.3004 | 10.2997 | 10.2997 |
| | (Gap) | (2.19%) | (7.23%) | (7.16%) | (0.35%) | (0.34%) | (0.34%) |
| | Mean | 10.5635 | 11.1614 | 10.9350 | 10.2995 | 10.3008 | 10.3008 |
| | (Gap) | (2.91%) | (8.75%) | (6.54%) | (0.34%) | (0.35%) | (0.35%) |
| | Exponential | 10.5726 | 11.1848 | 10.9589 | 10.3054 | 10.3051 | 10.3051 |
| | (Gap) | (3.00%) | (8.98%) | (6.78%) | (0.40%) | (0.39%) | (0.39%) |
| | Critic | 10.5617 | 11.1401 | 10.9439 | 10.2987 | 10.2994 | 10.2994 |
| | (Gap) | (2.90%) | (8.55%) | (6.63%) | (0.33%) | (0.34%) | (0.34%) |
| GCN | Rollout | 10.4232 | 10.6404 | 10.6094 | 10.2955 | 10.2956 | 10.2958 |
| | (Gap) | (1.54%) | (3.66%) | (3.36%) | (0.30%) | (0.30%) | (0.30%) |
| | Mean | 10.4321 | 10.8095 | 10.6076 | 10.2807 | 10.2830 | 10.2830 |
| | (Gap) | (1.63%) | (5.31%) | (3.34%) | (0.15%) | (0.18%) | (0.18%) |
| | Exponential | 10.4729 | 10.9573 | 10.7257 | 10.2837 | 10.2859 | 10.2859 |
| | (Gap) | (2.02%) | (6.75%) | (4.49%) | (0.18%) | (0.20%) | (0.20%) |
| | Critic | 10.7086 | 11.4549 | 11.0139 | 10.2859 | 10.2891 | 10.2891 |
| | (Gap) | (3.82%) | (0.54%) | (6.01%) | (0.20%) | (0.23%) | (0.23%) |
| GAT | Rollout | 10.4685 | 10.9202 | 10.8916 | 10.2956 | 10.2956 | 10.2957 |
| | (Gap) | (1.99%) | (6.40%) | (6.12%) | (0.30%) | (0.30%) | (0.30%) |
| | Mean | 10.6641 | 11.3499 | 11.0133 | 10.2865 | 10.2899 | 10.2898 |
| | (Gap) | (3.90%) | (0.59%) | (7.31%) | (0.21%) | (0.24%) | (0.24%) |
| | Exponential | 10.6487 | 11.3504 | 10.9869 | 10.2864 | 10.2881 | 10.2880 |
| | (Gap) | (3.75%) | (0.60%) | (7.05%) | (0.21%) | (0.22%) | (0.22%) |
| | Critic | 10.6566 | 11.3440 | 10.8813 | 10.2859 | 10.2888 | 10.2888 |
| | (Gap) | (4.33%) | (1.62%) | (7.31%) | (0.20%) | (0.23%) | (0.23%) |
| Uniform Random (Best) | | | | | 12.4788 | | |
| (Gap) | | | | | (21.62%) | | |
| Uniform Random (Mean) | | | | | 15.6327 | | |
| (Gap) | | | | | (52.40%) | | |
| Deterministic Greedy | | | | | 10.9831 | | |
| (Gap) | | | | | (7.02%) | | |
| GUROBI/SCIP (Optimum) | | | | | 10.2650 | | |
| (Gap) | | | | | (0.00%) | | |

## E.6.2 Benchmark Results

**Main benchmark** Table 24 shows the main numerical results when the methods are trained and tested to choose $k = 10$ locations on instances with $n = 100$ locations. Table 25 shows the main numerical results when the methods are trained and tested to choose $k = 10$ sets on instances with $n = 100$ sets and $m = 200$ items in total. Each item has a random weight between 1 and 10, and the number of items in each set is randomly sampled between 5 and 15. The reported results are averaged over 1,000 randomly generated test instances. We also report the average gap between the performance for each setting and the optimum by solvers as described in Appendix D.1.1.

Here we use absolute values since we *minimize* the total distance for FLP while *maximizing* the total weights for MCP. When using absolute values, it is consistent that smaller gaps correspond to better performance. The performance of RL methods with sampling is consistently better than the two baselines, uniform random (UR) and deterministic greedy (DG), showing their effectiveness on those two problems.

**Effect of the encoder** Overall, the performance of different encoders is similar. For FLP, we can observe GCN's marginal superiority (except when we use Critic as the RL baseline). For MCP, the best encoders for different RL baselines are different, but MLP's performance is the overall best.

Table 25: Performance of different methods on the maximum coverage problem (MCP) benchmark. For the performance, the larger the better.

| Encoder | RL Baseline | Sample (Best) | Sample (Mean) | Greedy | AS | Active Search EAS-Emb | EAS-Lay |
|---|---|---|---|---|---|---|---|
| MLP | Rollout | 682.4741 | 662.4359 | 665.1740 | 689.6200 | 689.6070 | 689.6070 |
| | (Gap) | (0.96%) | (3.31%) | (3.05%) | (0.09%) | (0.09%) | (0.09%) |
| | Mean | 682.4011 | 664.7105 | 668.7470 | 682.0610 | 689.5900 | 689.5900 |
| | (Gap) | (1.06%) | (3.96%) | (3.56%) | (1.18%) | (0.09%) | (0.09%) |
| | Exponential | 683.0300 | 665.1467 | 666.6640 | 671.3130 | 689.5870 | 689.5870 |
| | (Gap) | (1.09%) | (3.99%) | (3.64%) | (9.68%) | (0.09%) | (0.09%) |
| | Critic | 683.1511 | 666.9047 | 668.6411 | 687.8240 | 689.3510 | 689.3510 |
| | (Gap) | (1.43%) | (5.40%) | (4.92%) | (0.35%) | (0.13%) | (0.13%) |
| SAGE | Rollout | 681.8690 | 664.1233 | 665.9901 | 689.4810 | 689.5020 | 689.4930 |
| | (Gap) | (1.14%) | (3.71%) | (3.44%) | (0.11%) | (0.11%) | (0.11%) |
| | Mean | 682.1360 | 669.2791 | 670.4091 | 666.0360 | 689.5990 | 689.5890 |
| | (Gap) | (1.06%) | (3.63%) | (3.05%) | (10.44%) | (0.09%) | (0.09%) |
| | Exponential | 680.3970 | 653.0383 | 656.3170 | 675.2220 | 689.5990 | 689.5980 |
| | (Gap) | (1.06%) | (3.95%) | (3.46%) | (2.18%) | (0.09%) | (0.09%) |
| | Critic | 676.9190 | 645.9108 | 649.6940 | 647.9050 | 688.4500 | 688.4650 |
| | (Gap) | (1.94%) | (6.43%) | (5.89%) | (6.12%) | (0.26%) | (0.26%) |
| GEN | Rollout | 680.2640 | 648.2318 | 656.3710 | 689.4430 | 689.4660 | 689.4660 |
| | (Gap) | (1.10%) | (2.96%) | (2.80%) | (0.12%) | (0.11%) | (0.11%) |
| | Mean | 682.1960 | 662.1896 | 664.6721 | 681.3950 | 689.5670 | 689.5670 |
| | (Gap) | (0.97%) | (3.56%) | (3.34%) | (1.28%) | (0.10%) | (0.10%) |
| | Exponential | 682.4290 | 662.5012 | 665.8010 | 689.4060 | 689.5650 | 689.5650 |
| | (Gap) | (1.07%) | (3.70%) | (3.18%) | (0.12%) | (0.10%) | (0.10%) |
| | Critic | 682.3510 | 664.1604 | 667.7340 | 689.6170 | 689.3940 | 689.3940 |
| | (Gap) | (1.45%) | (6.08%) | (4.91%) | (0.09%) | (0.12%) | (0.12%) |
| Uniform Random (Best) | | | | | 527.9360 | | |
| (Gap) | | | | | (-23.50%) | | |
| Uniform Random (Mean) | | | | | 432.7287 | | |
| (Gap) | | | | | (-37.30%) | | |
| Deterministic Greedy | | | | | 680.2050 | | |
| (Gap) | | | | | (-1.46%) | | |
| GUROBI/SCIP (Optimum) | | | | | 690.2350 | | |
| (Gap) | | | | | (0.00%) | | |

**Effect of the RL baseline** For FLP, for the four considered RL baselines (Rollout, Mean, Exponential, Critic), Rollout is consistently better than the other three. For MCP, the differences in the performance of different RL baselines are not significant.

**Effect of active search** Active search significantly improves performance in almost all cases. For FLP, interestingly, Rollout achieves the best overall performance without active search, but Rollout underforms in many cases with test-time active search. Notably, the performance of the original active search (AS) is less stable than the two variants of efficient active search (EAS), especially for MCP. In our understanding, AS was originally designed for routing problems and uses multi-start sampling with distinct initial action (i.e., the first location/set to choose). Such a strategy is useful for routing problems due to symmetry but is less useful for problems without symmetry, such as FLP and MCP.

**Test-time sampling techniques** We also consider other test-time sampling techniques: top-$p$ sampling [48] and different sampling temperatures. Top-$p$ sampling discards actions with low probabilities, and top-$p$ sampling with lower $p$ values discards more low-probability actions. For sampling temperatures, higher temperatures give more uniform sampling. The considered $p$ values are: 0.5, 0.6, 0.7, 0.8, 0.9, 0.95, 0.99, 1.0. The sampling temperatures considered are 0.01, 0.03, 0.1, 0.3, 0.5, 0.7, 0.8, 0.9, 1.0, 1.1, 1.2, 1.5, and 2.0. Fig. 28 show the heatmaps for each combination of encoder and RL baseline, for FLP and MCP. In each subplot, the $x$-axis represents the value of $p$ in top-$p$ sampling, and the $y$-axis represents the sampling temperature. For each combination, the

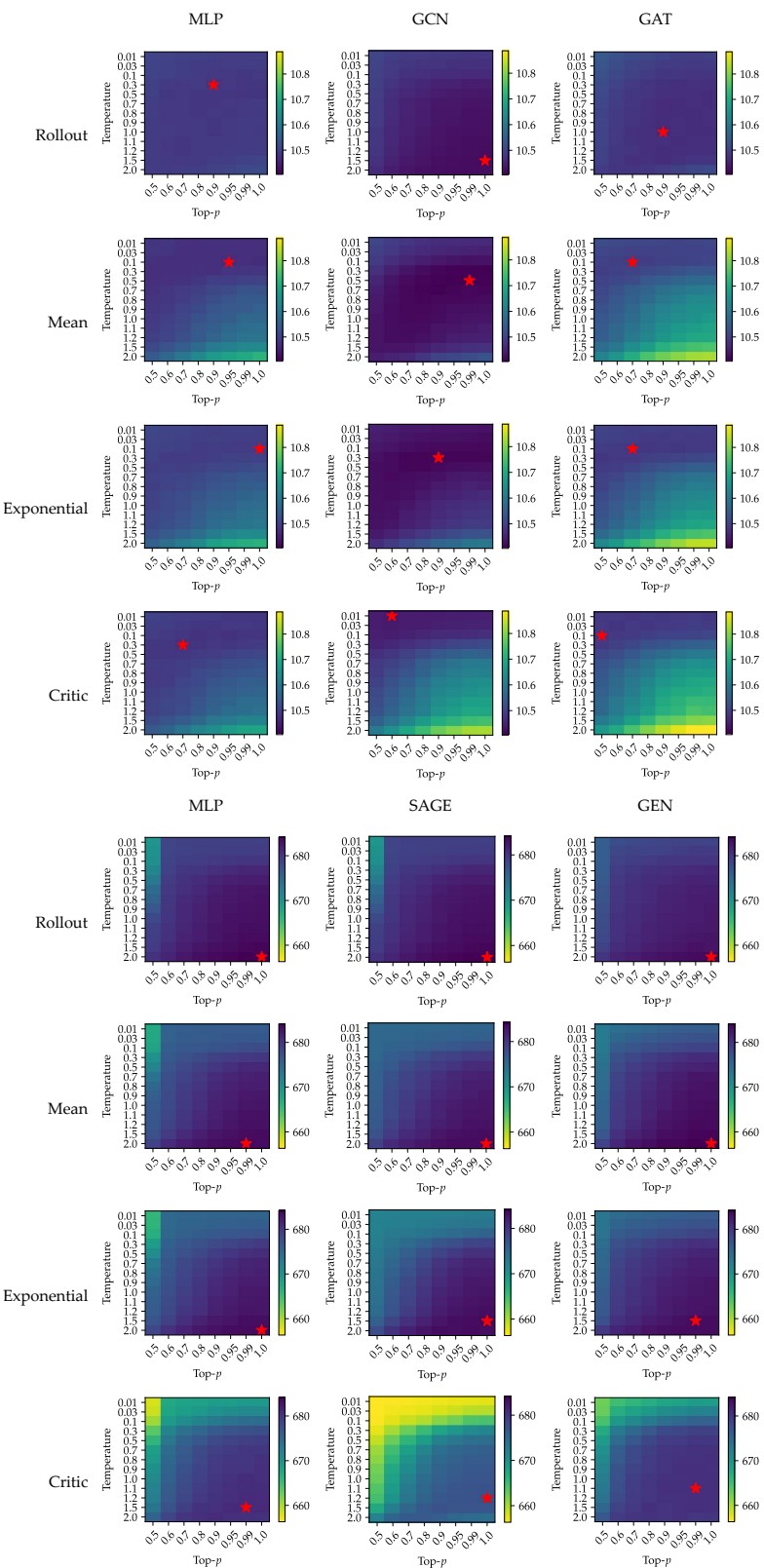

Figure 28: Performance of sampling with different $p$ values for top-$p$ sampling and different sampling temperatures. Top: FLP; Bottom: MCP. For each combination of encoder and RL baseline, the best performance is marked with a star.

Table 26: Performance of different methods on the facility location problem (FLP) out-of-distribution instances. For the performance, the smaller the better.

| Encoder | RL Baseline | Sample (Best) | Sample (Mean) | Greedy | AS | Active Search EAS-Emb | EAS-Lay |
|---|---|---|---|---|---|---|---|
| MLP | Rollout | 14.7612 | 15.2979 | 15.2709 | 14.4160 | 14.4181 | 14.4181 |
| | (Gap) | (3.85%) | (7.63%) | (7.44%) | (1.42%) | (1.43%) | (1.43%) |
| | Mean | 15.0045 | 15.7343 | 15.3075 | 14.5315 | 14.5331 | 14.5331 |
| | (Gap) | (5.56%) | (10.70%) | (7.70%) | (2.23%) | (2.24%) | (2.24%) |
| | Exponential | 15.0022 | 15.7144 | 15.3131 | 14.5274 | 14.5266 | 14.5266 |
| | (Gap) | (5.54%) | (10.56%) | (7.74%) | (2.20%) | (2.19%) | (2.19%) |
| | Critic | 14.9670 | 15.6631 | 15.2781 | 14.5147 | 14.5132 | 14.5132 |
| | (Gap) | (5.30%) | (10.20%) | (7.49%) | (2.11%) | (2.10%) | (2.10%) |
| GCN | Rollout | 14.9564 | 15.4230 | 15.3610 | 14.6254 | 14.6239 | 14.6248 |
| | (Gap) | (5.22%) | (8.51%) | (8.07%) | (2.89%) | (2.88%) | (2.89%) |
| | Mean | 15.1380 | 15.8310 | 15.3713 | 14.6554 | 14.6572 | 14.6574 |
| | (Gap) | (6.50%) | (11.38%) | (8.14%) | (3.10%) | (3.11%) | (3.12%) |
| | Exponential | 15.2197 | 15.9598 | 15.4441 | 14.6961 | 14.6963 | 14.6973 |
| | (Gap) | (7.08%) | (12.29%) | (8.66%) | (3.39%) | (3.39%) | (3.40%) |
| | Critic | 15.1754 | 15.9835 | 15.2815 | 14.6579 | 14.6634 | 14.6642 |
| | (Gap) | (6.53%) | (12.00%) | (8.23%) | (3.12%) | (3.16%) | (3.16%) |
| GAT | Rollout | 14.7503 | 15.2808 | 15.2593 | 14.4142 | 14.4150 | 14.4143 |
| | (Gap) | (3.77%) | (7.51%) | (7.36%) | (1.40%) | (1.41%) | (1.40%) |
| | Mean | 15.1147 | 15.9092 | 15.2895 | 14.5944 | 14.5986 | 14.5946 |
| | (Gap) | (6.34%) | (11.93%) | (7.57%) | (2.67%) | (2.70%) | (2.67%) |
| | Exponential | 15.1639 | 15.9886 | 15.2945 | 14.5991 | 14.6004 | 14.6011 |
| | (Gap) | (6.68%) | (12.49%) | (7.60%) | (2.70%) | (2.71%) | (2.72%) |
| | Critic | 15.1428 | 15.9191 | 15.3835 | 14.6053 | 14.6111 | 14.6111 |
| | (Gap) | (6.76%) | (12.46%) | (7.51%) | (2.75%) | (2.79%) | (2.79%) |
| Uniform Random (Best) | | | | 18.3215 | | | |
| (Gap) | | | | (28.92%) | | | |
| Uniform Random (Mean) | | | | 21.7044 | | | |
| (Gap) | | | | (52.74%) | | | |
| Deterministic Greedy | | | | 15.3090 | | | |
| (Gap) | | | | (7.71%) | | | |
| GUROBI/SCIP (Optimum) | | | | 14.2148 | | | |
| (Gap) | | | | (0.00%) | | | |

best performance is marked with a red star. For FLP, the best performance is usually achieved with a proper (i.e., neither too high nor too low) level of randomness. As the $p$ value of top-$p$ sampling increases, the best sampling temperature decreases. Recall that both increasing the $p$ value and increasing the sampling temperature would increase the randomness in sampling. Overall, compared to other RL baselines, Rollout needs a higher level of randomness to perform best. For MCP, the best performance is usually achieved without top-$p$ sampling and with a high sampling temperature, i.e., without high randomness in the sampling space.

### E.6.3 Out-of-distribution

**Results on out-of-distribution instances** Table 26 shows the main numerical results when the methods are trained to choose $k = 10$ locations on instances with $n = 100$ locations, but tested to choose $k' = 20$ locations on instances with $n' = 200$ locations. Table 27 shows the main numerical results when the methods are trained to choose $k = 10$ sets on instances with $n = 100$ sets and $m = 200$ items in total and tested to choose $k' = 20$ sets on instances with $n' = 200$ sets and $m' = 400$ items in total. Each item has a random weight between 1 and 10, and the number of items in each set is randomly sampled between 5 and 15. The reported results are averaged over $1,000$ randomly generated test instances. We also report the average gap for each setting. Overall, the performance of RL methods generalizes well to out-of-distribution instances, being significantly higher than both Uniform Random and Deterministic Greedy with enough sampling.

Table 27: Performance of different methods on the maximum coverage problem (MCP) out-of-distribution instances. For the performance, the larger the better.

| Encoder | RL Baseline | Sample (Best) | Sample (Mean) | Greedy | AS | Active Search EAS-Emb | EAS-Lay |
|---|---|---|---|---|---|---|---|
| MLP | Rollout | 1356.8970 | 1299.8690 | 1307.5250 | 1385.3340 | 1385.3280 | 1385.3280 |
| | (Gap) | (-1.83%) | (-5.48%) | (-5.03%) | (-0.32%) | (-0.33%) | (-0.33%) |
| | Mean | 1360.7710 | 1306.4015 | 1312.6290 | 1319.8180 | 1383.3580 | 1383.3580 |
| | (Gap) | (-2.34%) | (-6.45%) | (-5.89%) | (-5.04%) | (-0.47%) | (-0.47%) |
| | Exponential | 1360.7830 | 1306.3337 | 1312.7070 | 1088.0180 | 1383.9670 | 1383.9670 |
| | (Gap) | (-2.49%) | (-6.64%) | (-6.23%) | (-21.71%) | (-0.42%) | (-0.42%) |
| | Critic | 1363.9190 | 1313.2830 | 1319.5280 | 1353.9080 | 1377.3780 | 1377.3780 |
| | (Gap) | (-3.29%) | (-7.83%) | (-7.33%) | (-2.59%) | (-0.90%) | (-0.90%) |
| SAGE | Rollout | 1353.9790 | 1297.5763 | 1303.7120 | 1382.2220 | 1382.1140 | 1382.1140 |
| | (Gap) | (-2.55%) | (-6.61%) | (-6.16%) | (-0.55%) | (-0.56%) | (-0.56%) |
| | Mean | 1366.0050 | 1320.5641 | 1325.5570 | 1121.7650 | 1384.3780 | 1384.3650 |
| | (Gap) | (-2.06%) | (-5.98%) | (-5.53%) | (-19.30%) | (-0.39%) | (-0.40%) |
| | Exponential | 1344.1420 | 1281.0377 | 1288.0360 | 1288.2830 | 1383.6030 | 1383.5500 |
| | (Gap) | (-2.30%) | (-6.38%) | (-5.73%) | (-7.31%) | (-0.45%) | (-0.45%) |
| | Critic | 1331.1100 | 1266.6130 | 1276.0670 | 1092.0550 | 1367.4660 | 1367.4690 |
| | (Gap) | (-4.23%) | (-8.87%) | (-8.19%) | (-21.42%) | (-1.61%) | (-1.61%) |
| GEN | Rollout | 1334.2700 | 1269.0966 | 1284.4550 | 1385.6540 | 1385.5750 | 1385.5750 |
| | (Gap) | (-1.68%) | (-4.96%) | (-4.60%) | (-0.30%) | (-0.31%) | (-0.31%) |
| | Mean | 1354.8450 | 1297.2153 | 1302.8560 | 1305.4070 | 1384.3080 | 1384.2980 |
| | (Gap) | (-2.06%) | (-5.98%) | (-5.52%) | (-6.08%) | (-0.40%) | (-0.40%) |
| | Exponential | 1357.4750 | 1300.7056 | 1309.8040 | 1376.1300 | 1384.3780 | 1384.3900 |
| | (Gap) | (-2.11%) | (-6.18%) | (-5.45%) | (-0.99%) | (-0.39%) | (-0.39%) |
| | Critic | 1360.0420 | 1303.4360 | 1313.6640 | 1366.2960 | 1374.8630 | 1374.8370 |
| | (Gap) | (-4.00%) | (-8.68%) | (-7.58%) | (-1.69%) | (-1.08%) | (-1.08%) |
| Uniform Random (Best) | | | | | 1003.3390 | | |
| (Gap) | | | | | (-27.80%) | | |
| Uniform Random (Mean) | | | | | 866.3536 | | |
| (Gap) | | | | | (-37.66%) | | |
| Deterministic Greedy | | | | | 1367.2240 | | |
| (Gap) | | | | | (-1.63%) | | |
| GUROBI/SCIP (Optimum) | | | | | 1389.8450 | | |
| (Gap) | | | | | (0.00%) | | |

**Effect of the encoder** For FLP, unlike the main benchmark, the superiority of GCN no longer exists for out-of-distribution instances. For MCP, the best encoders for different RL baselines are still different, and the performance of MLP is the best.

**Effect of the RL baseline** For FLP, again, Rollout is overall better than the other three. For MCP, the best RL baselines for different encoders are different, and Mean and Critic are overall good choices.

**Effect of active search** Again, active search clearly improves performance in almost all cases. For FLP, unlike the main benchmark, for out-of-distribution instances, Rollout overall performs best with and without active search. Still, the performance of the original active search (AS) is less stable than the two variants of efficient active search (EAS). With active search (especially EAS), the performance of RL methods is consistently better than that of Deterministic Greedy and is close to the optimum.

**Test-time sampling techniques** For out-of-distribution instances, we also consider top-$p$ sampling and different sampling temperatures as the main benchmark. The considered $p$ values are: 0.5, 0.6, 0.7, 0.8, 0.9, 0.95, 0.99, 1.0. The sampling temperatures considered are 0.01, 0.03, 0.1, 0.3, 0.5, 0.7, 0.8, 0.9, 1.0, 1.1, 1.2, 1.5, and 2.0. Fig. 29 show the heatmaps for each combination of encoder and RL baseline, for FLP and MCP. In each subplot, the $x$-axis represents the value of $p$ in top-$p$ sampling, and the $y$-axis represents the sampling temperature. For each combination, the best performance is marked with a red star. For both FLP and MCP, the best performance is usually

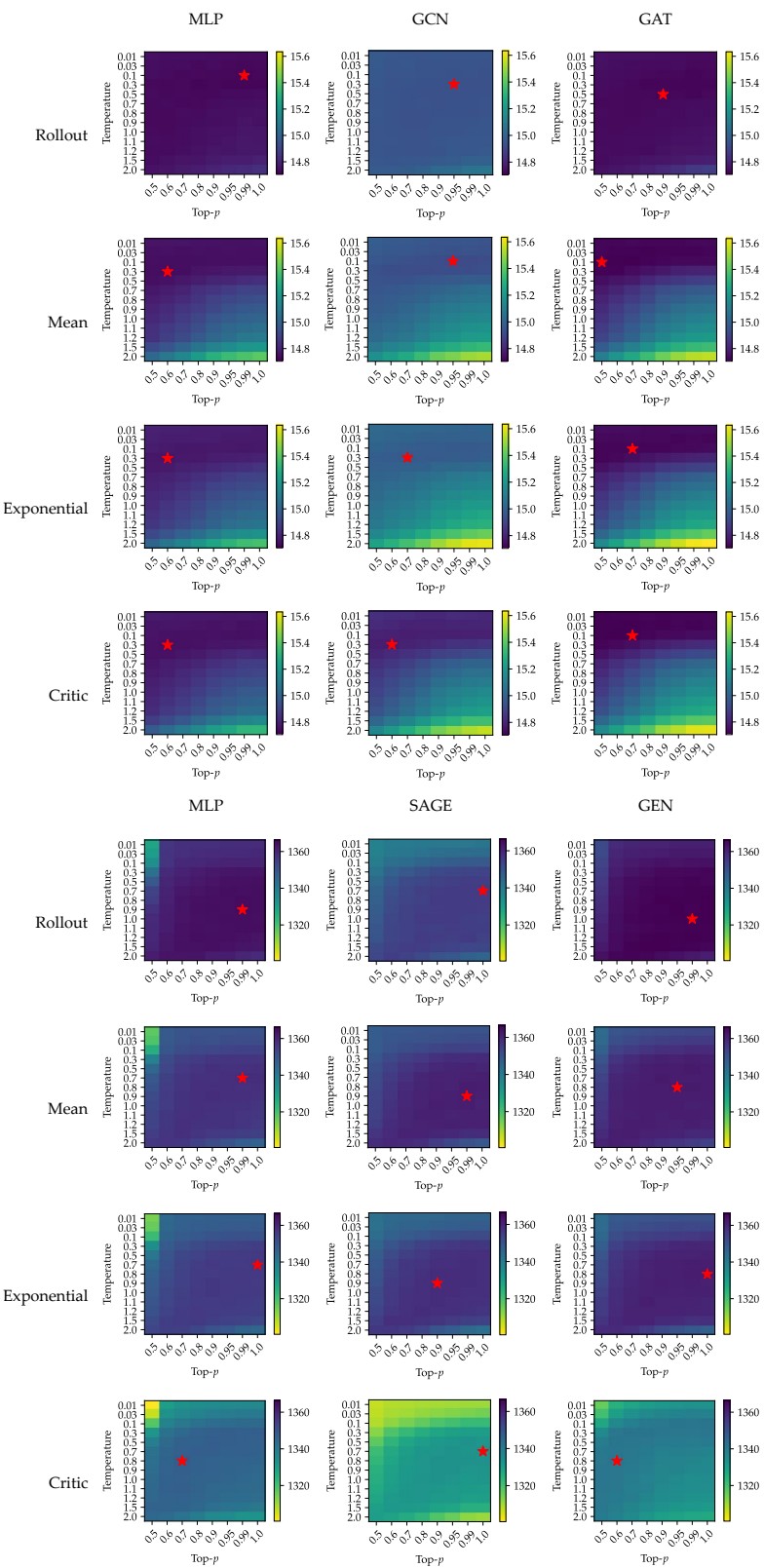

Figure 29: Performance of sampling on out-of-distribution instances with different $p$ values for top-$p$ sampling and different sampling temperatures. Top: FLP; Bottom: MCP. For each combination of encoder and RL baseline, the best performance is marked with a star.

achieved with a proper (i.e., neither too high nor too low) level of randomness. As the $p$ value of top-$p$ sampling increases, the best sampling temperature decreases. Recall that both increasing the $p$ value and increasing the sampling temperature would increase the randomness in sampling.

## E.7 Efficient Software Routines

### E.7.1 Mixed-Precision Training

RL4CO supports multiple device types as well as floating point precisions by leveraging PyTorch Lightning [39].

Table 28: Running time and memory usage of the AM model trained using FP32 and FP16 mixed precision (FP16-mix), evaluated over 5 epochs with a training size of 10,000 in the CVPR20, CVPR50, and CVPR100.

| Problem | Precision | Running time [s] | Memory usage [GiB] |
|---|---|---|---|
| CVRP20 | FP32 | $6.33 \pm 0.26$ | $1.41 \pm 0.04$ |
| | FP16-mix | $5.89 \pm 0.07$ | $0.84 \pm 0.01$ |
| CVRP50 | FP32 | $13.58 \pm 0.12$ | $4.79 \pm 0.40$ |
| | FP16-mix | $11.68 \pm 0.30$ | $2.30 \pm 0.25$ |
| CVRP100 | FP32 | $35.09 \pm 0.71$ | $13.47 \pm 0.63$ |
| | FP16-mix | $25.11 \pm 0.66$ | $8.14 \pm 0.82$ |

As Table 28 shows mixed-precision training can successfully reduce computational costs both in terms of runtime and especially with memory usage.

### E.7.2 FlashAttention

Given that the Attention operator is used on several occasions, especially in autoregressive models, there is a need to support fast and efficient software routines that can compute this ubiquitous operation. In RL4CO, we natively support FlashAttention [34, 33] from both PyTorch 2.0+ and the original FlashAttention repository [23], to which we also made some minor contributions when we found bugs.

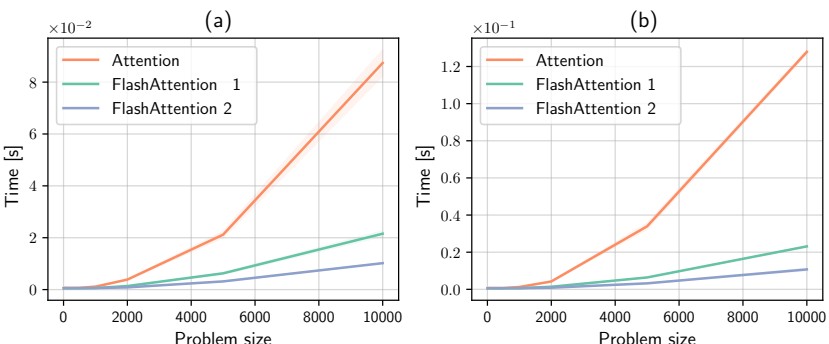

Figure 30: Running time of the graph attention encoder from the Attention Model, equipped with a standard attention layer, FlashAttention1, and FlashAttention2, across different problem sizes for both (a) the TSP and (b) the CVRP environments.

As shown in Fig. 30, different implementations can make a difference, especially with large problem sizes. It should be noted that while more scalable, FlashAttention at the moment is restricted to no or causal masks only. Therefore, usage in the masked attention decoding scheme is not possible at the moment, although it could be even more impactful due to the auto-regressive nature of our encoder-decoder scheme. Recent works as Pagliardini et al. [107] may be useful in extending FlashAttention

---

[23]Available at https://github.com/Dao-AILab/flash-attention.

to other masking patterns. We note that masking should, in principle, be even faster than un-masked attention, given that operations can be skipped in a per-block manner.

### E.7.3   Efficient Memory Handling in Environments

When dealing with RL problems, there is usually a tradeoff between memory and speed. This happens because environments are parallelized using multiple processes or threads, the policy network is replicated to each environment, or observations incoming from each environment need to be gathered, sent to the policy network, and then the output action scattered back to the representative environment. In the first case, network duplication causes large memory consumption; in the second case, communication between processes slows down. In RL4CO, we solve the problem by using batched environments, i.e., every environment is responsible not for a single instance of a problem but a batch of instances at the same time. By doing so, the policy can live in the same process of the environment, in the same device, and receive and send batched data without any communication overhead or additional memory consumption.

To further improve performances, we rewrite a core component of the TorchRL environment, namely the `step` method of the TorchRL base environment. The original `step` method performs some checks that, while useful for generic environments, can be omitted for RL4CO ones. It also duplicates the information in the output `TensorDict` by returning both the previous and the new state. In RL4CO, the previous state is always redundant, hence our `step` method does not keep it, reducing the memory consumption. We can see

Table 29: Comparison of training time in seconds for one epoch with RL4CO and TorchRL step method.

| Configuration | | Step method | |
| --- | --- | --- | --- |
| Environment | Nodes | RL4CO | TorchRL |
| TSP | 50 | 46.3 | 49.6 |
| | 100 | 102.9 | 108.6 |
| | 200 | 284.9 | 302.2 |
| CVRP | 50 | 72.9 | 73.4 |
| | 100 | 147.3 | 154.3 |
| | 200 | 371.7 | 406.4 |

in Table 29 that using RL4CO step method has a great benefit in terms of speed, especially for high-dimensional environments. The results are collected for the TSP and CVRP environment during one epoch of training for a dataset of size 100000. The table shows the difference in training time and peak allocated memory for the training when the environment uses the TorchRL step method and the RL4CO step method. The peak allocated memory is computed using the `torch.cuda.max_memory_allocated` method from PyTorch, and experiments are run on a Tesla V100 DGX 32GB.

### E.8   Towards Foundation Models

**Motivation**   Although learning to solve VRPs has gained significant attention, previous methods are only structured and trained independently on a specific problem, making them less generic and practical. Inspired by the recent success of foundation models in the language and vision domains, some works started to build foundation models for VRPs [89, 157, 13], aiming to solve a wide spectrum of problem variants using a single model. The main idea is to train a (large) model on diverse VRPs, which can be represented by a unified template. Typically, VRPs share several common attributes. For example, CVRP and VRPTW share the capacity attribute while only differing in the time window attribute. Therefore, a simple template could be a union set of attributes that exist in all VRP variants. By training on diverse VRP variants leveraging this unified representation, the foundation VRP model has the potential to efficiently and effectively solve any variant, making it a favorable choice versus traditional solvers (e.g., OR-Tools [111]) in the future.

### E.8.1   Experimental Setting

For traditional solvers, we use HGS-PyVRP [144], an open-source VRP solver based on the state-of-the-art HGS-CVRP [138], and Google's OR-Tools [111], an open-source solver based on constraint programming for complex optimization problems, to solve all VRP variants considered in this study. Both baseline methods solve each instance on a single CPU core with a time limit of 10 and 20

Table 30: Performance on 1,000 test instances. * represents 0.000%, with which the gaps are computed.

| | Method | N = 50 Obj. | Gap | Time | N = 100 Obj. | Gap | Time | | Method | N = 50 Obj. | Gap | Time | N = 100 Obj. | Gap | Time |
|---|---|---|---|---|---|---|---|---|---|---|---|---|---|---|---|
| CVRP | HGS-PyVRP | 10.287 | * | 4.6m | 15.543 | * | 9.2m | VRPTW | HGS-PyVRP | 16.032 | * | 4.6m | 25.433 | * | 9.2m |
| | OR-Tools | 10.523 | 2.294% | 4.6m | 16.361 | 5.263% | 9.2m | | OR-Tools | 16.124 | 0.574% | 4.6m | 25.923 | 1.927% | 9.2m |
| | MTPOMO | 10.408 | 1.176% | 2s | 15.809 | 1.711% | 10s | | MTPOMO | 16.396 | 2.270% | 2s | 26.391 | 3.767% | 11s |
| | MVMoE | 10.397 | 1.069% | 3s | 15.782 | 1.538% | 13s | | MVMoE | 16.394 | 2.258% | 3s | 26.357 | 3.633% | 14s |
| | MVMoE-L | 10.404 | 1.137% | 3s | 15.790 | 1.589% | 12s | | MVMoE-L | 16.393 | 2.252% | 3s | 26.359 | 3.641% | 13s |
| OVRP | HGS-PyVRP | 6.494 | * | 4.6m | 9.730 | * | 9.2m | VRPL | HGS-PyVRP | 10.328 | * | 4.6m | 15.637 | * | 9.2m |
| | OR-Tools | 6.555 | 0.939% | 4.6m | 10.081 | 3.607% | 9.2m | | OR-Tools | 10.570 | 2.343% | 4.6m | 16.466 | 5.302% | 9.2m |
| | MTPOMO | 6.712 | 3.357% | 2s | 10.241 | 5.252% | 10s | | MTPOMO | 10.454 | 1.220% | 2s | 15.921 | 1.816% | 12s |
| | MVMoE | 6.696 | 3.111% | 3s | 10.213 | 4.964% | 13s | | MVMoE | 10.442 | 1.104% | 3s | 15.886 | 1.592% | 13s |
| | MVMoE-L | 6.704 | 3.234% | 2s | 10.215 | 4.985% | 12s | | MVMoE-L | 10.450 | 1.181% | 2s | 15.898 | 1.669% | 10s |
| VRPB | HGS-PyVRP | 9.688 | * | 4.6m | 14.386 | * | 9.2m | OVRPTW | HGS-PyVRP | 10.485 | * | 4.6m | 16.900 | * | 9.2m |
| | OR-Tools | 9.829 | 1.455% | 4.6m | 15.010 | 4.338% | 9.2m | | OR-Tools | 10.497 | 0.114% | 4.6m | 17.023 | 0.728% | 9.2m |
| | MTPOMO | 9.975 | 2.962% | 2s | 15.014 | 4.365% | 10s | | MTPOMO | 10.664 | 1.707% | 2s | 17.426 | 3.112% | 11s |
| | MVMoE | 9.954 | 2.746% | 3s | 14.962 | 4.004% | 13s | | MVMoE | 10.665 | 1.717% | 3s | 17.421 | 3.083% | 15s |
| | MVMoE-L | 9.963 | 2.839% | 2s | 14.976 | 4.101% | 11s | | MVMoE-L | 10.665 | 1.717% | 2s | 17.411 | 3.024% | 14s |
| OVRPB | HGS-PyVRP | 6.897 | * | 4.6m | 10.304 | * | 9.2m | OVRPBL | HGS-PyVRP | 6.904 | * | 4.6m | 10.310 | * | 9.2m |
| | OR-Tools | 6.940 | 0.623% | 4.6m | 10.611 | 2.979% | 9.2m | | OR-Tools | 6.949 | 0.652% | 4.6m | 10.613 | 2.939% | 9.2m |
| | MTPOMO | 7.392 | 7.177% | 2s | 11.787 | 14.392% | 10s | | MTPOMO | 7.400 | 7.184% | 2s | 11.786 | 14.316% | 10s |
| | MVMoE | 7.566 | 9.700% | 3s | 11.873 | 15.227% | 13s | | MVMoE | 7.577 | 9.748% | 3s | 11.875 | 15.179% | 13s |
| | MVMoE-L | 7.388 | 7.119% | 2s | 11.806 | 14.577% | 12s | | MVMoE-L | 7.391 | 7.054% | 2s | 11.814 | 14.588% | 12s |
| OVRPBLTW | HGS-PyVRP | 11.597 | * | 4.6m | 19.005 | * | 9.2m | OVRPBTW | HGS-PyVRP | 11.590 | * | 4.6m | 19.167 | * | 9.2m |
| | OR-Tools | 11.612 | 0.129% | 4.6m | 19.198 | 1.016% | 9.2m | | OR-Tools | 11.610 | 0.173% | 4.6m | 19.314 | 0.767% | 9.2m |
| | MTPOMO | 11.986 | 3.354% | 2s | 20.048 | 5.488% | 11s | | MTPOMO | 11.980 | 3.365% | 2s | 20.209 | 5.436% | 11s |
| | MVMoE | 11.949 | 3.305% | 3s | 20.092 | 5.720% | 15s | | MVMoE | 11.957 | 3.167% | 3s | 20.254 | 5.671% | 15s |
| | MVMoE-L | 11.961 | 3.139% | 3s | 20.033 | 5.409% | 14s | | MVMoE-L | 11.951 | 3.115% | 2s | 20.173 | 5.249% | 14s |
| OVRPL | HGS-PyVRP | 6.510 | * | 4.6m | 9.709 | * | 9.2m | OVRPLTW | HGS-PyVRP | 10.455 | * | 4.6m | 16.962 | * | 9.2m |
| | OR-Tools | 6.571 | 0.937% | 4.6m | 10.047 | 3.481% | 9.2m | | OR-Tools | 10.465 | 0.096% | 4.6m | 17.100 | 0.814% | 9.2m |
| | MTPOMO | 6.732 | 3.410% | 2s | 10.216 | 5.222% | 10s | | MTPOMO | 10.625 | 1.626% | 2s | 17.486 | 3.089% | 11s |
| | MVMoE | 6.713 | 3.118% | 3s | 10.187 | 4.923% | 13s | | MVMoE | 10.631 | 1.683% | 3s | 17.483 | 3.072% | 15s |
| | MVMoE-L | 6.725 | 3.303% | 2s | 10.185 | 4.903% | 12s | | MVMoE-L | 10.635 | 1.722% | 3s | 17.474 | 3.019% | 14s |
| VRPBL | HGS-PyVRP | 9.688 | * | 4.6m | 14.373 | * | 9.2m | VRPBLTW | HGS-PyVRP | 18.361 | * | 4.6m | 29.026 | * | 9.2m |
| | OR-Tools | 9.820 | 1.363% | 4.6m | 15.084 | 4.947% | 9.2m | | OR-Tools | 18.422 | 0.332% | 4.6m | 29.830 | 2.770% | 9.2m |
| | MTPOMO | 9.994 | 3.159% | 2s | 15.033 | 4.592% | 10s | | MTPOMO | 19.028 | 3.633% | 2s | 31.062 | 7.014% | 11s |
| | MVMoE | 9.971 | 2.921% | 3s | 14.979 | 4.286% | 13s | | MVMoE | 18.967 | 3.300% | 3s | 31.114 | 7.194% | 15s |
| | MVMoE-L | 9.977 | 2.983% | 2s | 14.990 | 4.293% | 11s | | MVMoE-L | 18.998 | 3.469% | 3s | 31.032 | 6.911% | 13s |
| VRPBTW | HGS-PyVRP | 18.167 | * | 4.6m | 29.000 | * | 9.2m | VRPLTW | HGS-PyVRP | 15.951 | * | 4.6m | 25.678 | * | 9.2m |
| | OR-Tools | 18.374 | 1.139% | 4.6m | 29.964 | 3.324% | 9.2m | | OR-Tools | 16.036 | 0.533% | 4.6m | 26.156 | 1.862% | 9.2m |
| | MTPOMO | 18.995 | 4.558% | 2s | 31.184 | 7.531% | 11s | | MTPOMO | 16.310 | 2.251% | 2s | 26.650 | 3.785% | 11s |
| | MVMoE | 18.934 | 4.222% | 3s | 31.223 | 7.666% | 15s | | MVMoE | 16.315 | 2.282% | 3s | 26.635 | 3.727% | 14s |
| | MVMoE-L | 18.970 | 4.420% | 2s | 31.138 | 7.372% | 14s | | MVMoE-L | 16.311 | 2.257% | 3s | 26.637 | 3.735% | 13s |

seconds for instances with 50 and 100 nodes, respectively. We parallelize traditional solvers across 16 CPU cores as in [74]. For neural solvers, we mostly follow the training setups from previous works [89, 157, 13]. In specific, the model is trained over 300 epochs, with each epoch containing 100,000 instances generated on the fly. The Adam optimizer is used with a learning rate of $3e - 4$, a weight decay of $1e - 6$, and a batch size of 256. The learning rate decays by 10 at 270 and 295 epochs. Note that different from Liu et al. [89], Zhou et al. [157], we allow various problem variants to be trained in each batch training following Berto et al. [13]. We consider 16 VRP variants as shown in Table 7, including the constraints of capacity, time window, backhaul, open route, and duration limit. The training variants include CVRP, OVRP, VRPL, VRPB, VRPTW, and OVRPTW. During inference, we use greedy rollout with x8 instance augmentation following Kwon et al. [76]. We report the average results (i.e., objective values and gaps) over the test dataset that contains 1,000 instances, and the total time to solve the entire test dataset. The gaps are computed with respect to the results of HGS-PyVRP. All neural solvers are implemented using RL4CO.

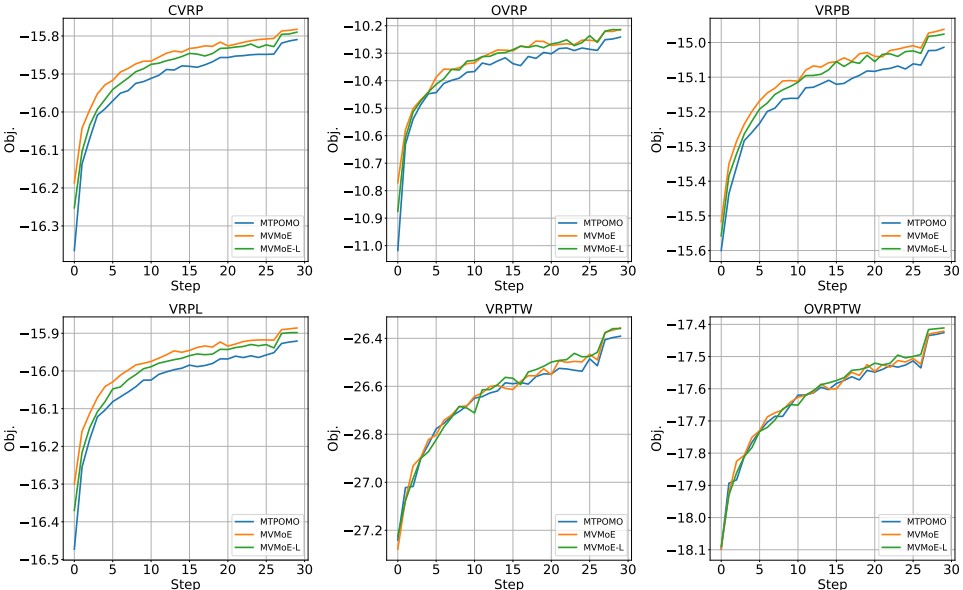

Figure 31: The validation curves of foundation models on $N = 100$.

### E.8.2 Empirical Results

We show the comprehensive evaluation results and validation curves in Table 30 and Fig. 31, respectively. The conclusions are consistent with previous studies [89, 157, 13] that 1) the foundation VRP solvers exhibit remarkable zero-shot generalization performance, even only trained on several VRPs with simple constraints; 2) conditional computation (e.g., mixture-of-experts [57, 121]) can greatly enhance the model capacity without a proportional increase in computation. In Table 31, we further show the performance on CVRPLIB [86], which is a real-world benchmark dataset including instances with diverse distributions. We empirically observe that training on multiple VRPs can significantly improve the out-of-distribution generalization performance of neural VRP solvers, demonstrating the great promise of developing foundation models in VRPs.

### E.8.3 Discussion

Foundation models, a class of large-scale deep learning models pre-trained on extensive datasets of diverse tasks, have recently revolutionized the fields of language and vision domains. They can generate text, translate languages, summarize content, and more, all without task-specific training. This versatility makes them incredibly useful across various applications, from chatbots to academic research. Aiming for a more powerful and general solver, recent studies explore the possibility of pretraining a large model on a huge amount of optimization tasks. The long-term goal is to develop a foundation model for VRPs (or more broadly COPs), which can efficiently solve any problem variant, comparably or better to the conventional solvers with respect to the solution quality and inference speed. Despite the recent advancements of foundation VRP models [89, 157, 13], there are many challenges that need to be addressed by the NCO community, including but not limited to: 1) *scaling:* current autoregressive-based models are challenging to scale to the parameter levels of large language models (e.g., billions of parameters) due to the expensive training cost. RL-based training is data inefficient and converges slowly, whereas SL-based training requires a significant amount of optimal solutions, which are non-trivial to obtain for NP-hard problems. They also fail to be efficiently trained on large-scale instances; 2) *performance:* the empirical results are still far short of traditional solvers (e.g., OR-Tools). They may also suffer from generalization and robustness issues; 3) *generality:* the current problem formulation or template cannot solve novel problem variants in a zero-shot manner; 4) *interpretability:* the decision-making of foundation models is hard to explain.

Table 31: Results on CVRPLib datasets with diverse distributions and sizes. All models are only trained on the uniformly distributed data with the size $N = 100$.

| Benchmark | Size $N$ | Ins. Num. | POMO-CVRP | | MTPOMO | | MVMoE | | MVMoE-L | |
|-----------|----------|-----------|-----------|------|--------|------|-------|------|---------|------|
| | | | Obj. | Gap | Obj. | Gap | Obj. | Gap | Obj. | Gap |
| Set A | 31-79 | 27 | 1088.5 | 4.9% | 1084.2 | 4.3% | 1081.0 | 3.8% | 1085.4 | 4.4% |
| Set B | 30-77 | 23 | 1013.9 | 5.5% | 1010.3 | 5.0% | 1003.5 | 4.0% | 1001.2 | 4.0% |
| Set F | 44-134 | 3 | 796.0 | 12.7% | 812.7 | 16.3% | 819.0 | 13.8% | 799.0 | 14.1% |
| Set M | 100-199 | 5 | 1157.4 | 6.3% | 1179.4 | 8.6% | 1181.8 | 8.8% | 1151.4 | 6.0% |
| Set P | 15-100 | 23 | 643.9 | 14.7% | 621.8 | 8.4% | 616.1 | 5.9% | 619.8 | 6.9% |
| Set X | 100-1000 | 100 | 77199.6 | 21.1% | 71153.8 | 11.7% | 72798.7 | 15.0% | 72446.1 | 13.9% |

Moreover, there is another line of research leveraging the existing large language models (LLMs) to generate solutions [149, 91, 55] or algorithms [117, 90, 151], yielding impressive results when integrated with problem-specific heuristics or general meta-heuristics. Some studies employ LLMs to investigate the interpretability of solvers [66], automate problem formulation or simplify the use of domain-specific tools [146, 2, 142] through text prompts. However, their performance is highly dependent on the utilized LLMs, and their outputs may be extremely sensitive to the designed prompts.

We view both as promising directions towards foundation models in combinatorial optimization. We call the attention from both the machine learning (ML) and operations research (OR) communities to advance the development of impactful foundational models and learning methods that are scalable, robust, generalizable, and interpretable across various optimization tasks in future work.

### E.9 Generalization of Training on Multiple Distributions and Multiple Tasks

Recent neural methods mostly train and test neural networks on the same task with instances of the same distribution and size, and hence suffer from inferior generalization performance. Some attempts have been made to alleviate the generalization issue, focusing on either distribution [16, 59, 147] or size [122]. More aligned to the diverse distribution and size settings in the benchmark dataset TSPLib and CVRPLib, Manchanda et al. [100] and Zhou et al. [156] consider generalization across both distribution and size in VRPs.

However, these generalization methods adopt extra model architectures and training paradigms, resulting in additional computational burdens. As a more efficient alternative, we observe that diversified training datasets significantly improve generalization performance. Specifically, as indicated in the prior works, training on mixed distributions [16] and mixed VRP variants [89, 157, 13] boosts the generalization capability. RL4CO, detailed in Appendix B.1.6, supports multiple VRP variants and the generation of diverse coordinate distributions, enabling straightforward experimental setups. The implementation specifics are outlined in Appendix D.3.4. Evaluation results on the CVRPLib [86], summarized in Table 4 and fully detailed in Table 32, demonstrate that training across multiple distributions (i.e., MDPOMO) achieves better generalization on datasets of similar size to the training set, whereas training across multiple VRP tasks (i.e., MTPOMO) exhibits superior generalization across larger and more diverse distributions. This indicates that different VRP variants share foundational knowledge, and learning from this diversity enhances generalization beyond conventional training on a single distribution, size, and task. These key findings highlight the necessity of developing foundational models across diverse combinatorial optimization domains.

Table 32: Full Results on CVRPLIB instances with models trained on $N = 50$. Greedy multi-start decoding is used.

| Instance | BKS | POMO Obj. | POMO Gap | MTPOMO Obj. | MTPOMO Gap | MDPOMO Obj. | MDPOMO Gap | Instance | BKS | POMO Obj. | POMO Gap | MTPOMO Obj. | MTPOMO Gap | MDPOMO Obj. | MDPOMO Gap |
|---|---|---|---|---|---|---|---|---|---|---|---|---|---|---|---|
| A-n32-k5 | 784 | 821 | 4.72% | 831 | 5.99% | 817 | 4.21% | X-n125-k30 | 55539 | 58759 | 5.80% | 58560 | 5.44% | 59924 | 7.90% |
| A-n33-k5 | 661 | 683 | 3.33% | 689 | 4.24% | 685 | 3.63% | X-n129-k18 | 28940 | 30611 | 5.77% | 30437 | 5.17% | 30516 | 5.45% |
| A-n33-k6 | 742 | 759 | 2.29% | 745 | 0.40% | 750 | 1.08% | X-n134-k13 | 10916 | 11805 | 8.14% | 12043 | 10.32% | 11771 | 7.83% |
| A-n34-k5 | 778 | 791 | 1.67% | 791 | 1.67% | 791 | 1.67% | X-n139-k10 | 13590 | 14562 | 7.15% | 14993 | 10.32% | 15328 | 12.79% |
| A-n36-k5 | 799 | 831 | 4.01% | 803 | 0.50% | 812 | 1.63% | X-n143-k7 | 15700 | 17293 | 10.15% | 17337 | 10.43% | 17062 | 8.68% |
| A-n37-k5 | 669 | 712 | 6.43% | 699 | 4.48% | 673 | 0.60% | X-n148-k46 | 43448 | 47711 | 9.81% | 46442 | 6.89% | 49444 | 13.80% |
| A-n37-k6 | 949 | 995 | 4.85% | 998 | 5.16% | 999 | 5.27% | X-n153-k22 | 21220 | 24506 | 15.49% | 23928 | 12.76% | 24562 | 15.75% |
| A-n38-k5 | 730 | 753 | 3.15% | 749 | 2.60% | 774 | 6.03% | X-n157-k13 | 16876 | 18702 | 10.82% | 18201 | 7.85% | 18560 | 9.98% |
| A-n39-k5 | 822 | 835 | 1.58% | 842 | 2.43% | 842 | 2.43% | X-n162-k11 | 14138 | 15678 | 10.89% | 15615 | 10.45% | 16257 | 14.99% |
| A-n39-k6 | 831 | 838 | 0.84% | 844 | 1.56% | 842 | 1.32% | X-n167-k10 | 20557 | 22331 | 8.63% | 23083 | 12.29% | 22839 | 11.10% |
| A-n44-k6 | 937 | 962 | 2.67% | 959 | 2.35% | 958 | 2.24% | X-n172-k51 | 45607 | 50471 | 10.67% | 48799 | 7.00% | 50689 | 11.14% |
| A-n45-k6 | 944 | 984 | 4.24% | 981 | 3.92% | 965 | 2.22% | X-n176-k26 | 47812 | 54316 | 13.60% | 53773 | 12.47% | 53197 | 11.26% |
| A-n45-k7 | 1146 | 1166 | 1.75% | 1163 | 1.48% | 1162 | 1.40% | X-n181-k23 | 25569 | 27331 | 6.89% | 27571 | 7.83% | 27572 | 7.83% |
| A-n46-k7 | 914 | 924 | 1.09% | 945 | 3.39% | 938 | 2.63% | X-n186-k15 | 24145 | 26981 | 11.75% | 27157 | 12.47% | 27011 | 11.87% |
| A-n48-k7 | 1073 | 1108 | 3.26% | 1121 | 4.47% | 1102 | 2.70% | X-n190-k8 | 16980 | 19414 | 14.33% | 19955 | 17.52% | 18355 | 8.10% |
| A-n53-k7 | 1010 | 1040 | 2.97% | 1080 | 6.93% | 1047 | 3.66% | X-n195-k51 | 44225 | 50357 | 13.87% | 47675 | 7.80% | 49878 | 12.78% |
| A-n54-k7 | 1167 | 1192 | 2.14% | 1191 | 2.06% | 1181 | 1.20% | X-n200-k36 | 58578 | 66149 | 12.92% | 62862 | 7.31% | 62466 | 6.64% |
| A-n55-k9 | 1073 | 1095 | 2.05% | 1124 | 4.75% | 1123 | 4.66% | X-n204-k19 | 19565 | 22013 | 12.51% | 22297 | 13.96% | 23018 | 17.65% |
| A-n60-k9 | 1354 | 1388 | 2.51% | 1398 | 3.25% | 1389 | 2.58% | X-n209-k16 | 30656 | 33810 | 10.29% | 33745 | 10.08% | 34060 | 11.10% |
| A-n61-k9 | 1034 | 1059 | 2.42% | 1090 | 5.42% | 1051 | 1.64% | X-n214-k11 | 10856 | 13108 | 20.74% | 13005 | 19.80% | 12586 | 15.94% |
| A-n62-k8 | 1288 | 1343 | 4.27% | 1329 | 3.18% | 1364 | 5.90% | X-n219-k73 | 117595 | 133173 | 13.25% | 125415 | 6.65% | 126942 | 7.95% |
| A-n63-k9 | 1616 | 1660 | 2.72% | 1660 | 2.72% | 1654 | 2.35% | X-n223-k34 | 40437 | 44173 | 9.24% | 44066 | 8.97% | 44609 | 10.32% |
| A-n63-k10 | 1314 | 1349 | 2.66% | 1342 | 2.13% | 1347 | 2.51% | X-n228-k23 | 25742 | 30685 | 19.20% | 29896 | 16.14% | 29593 | 14.96% |
| A-n64-k9 | 1401 | 1432 | 2.21% | 1438 | 2.64% | 1441 | 2.86% | X-n233-k16 | 19230 | 22082 | 14.83% | 22602 | 17.54% | 23553 | 22.48% |
| A-n65-k9 | 1174 | 1231 | 4.86% | 1234 | 5.11% | 1239 | 5.54% | X-n237-k14 | 27042 | 31000 | 14.64% | 31880 | 17.89% | 31617 | 16.92% |
| A-n69-k9 | 1159 | 1224 | 5.61% | 1207 | 4.14% | 1205 | 3.97% | X-n242-k48 | 82751 | 89900 | 8.64% | 87933 | 6.26% | 90125 | 8.91% |
| A-n80-k10 | 1763 | 1839 | 4.31% | 1825 | 3.52% | 1840 | 4.37% | X-n247-k50 | 37274 | 41688 | 11.84% | 42340 | 13.59% | 43318 | 16.22% |
| B-n31-k5 | 672 | 688 | 2.38% | 705 | 4.91% | 694 | 3.27% | X-n251-k28 | 38684 | 43430 | 12.27% | 42379 | 9.55% | 42721 | 10.44% |
| B-n34-k5 | 788 | 798 | 1.27% | 802 | 1.78% | 803 | 1.90% | X-n256-k16 | 18839 | 23449 | 24.47% | 21559 | 14.44% | 25704 | 36.44% |
| B-n35-k5 | 955 | 979 | 2.51% | 975 | 2.09% | 976 | 2.20% | X-n261-k13 | 26558 | 30384 | 14.41% | 31345 | 18.02% | 30630 | 15.33% |
| B-n38-k6 | 805 | 830 | 3.11% | 817 | 1.49% | 834 | 3.60% | X-n266-k58 | 75478 | 83838 | 11.08% | 83806 | 11.03% | 91188 | 20.81% |
| B-n39-k5 | 549 | 561 | 2.19% | 561 | 2.19% | 557 | 1.46% | X-n270-k35 | 35291 | 40274 | 14.12% | 39378 | 11.58% | 41661 | 18.05% |
| B-n41-k6 | 829 | 849 | 2.41% | 850 | 2.53% | 848 | 2.29% | X-n275-k28 | 21245 | 25909 | 21.95% | 25718 | 21.05% | 26474 | 24.61% |
| B-n43-k6 | 742 | 762 | 2.70% | 756 | 1.89% | 770 | 3.77% | X-n280-k17 | 33503 | 37659 | 12.40% | 39309 | 17.33% | 38119 | 13.78% |
| B-n44-k7 | 909 | 942 | 3.63% | 940 | 3.41% | 934 | 2.75% | X-n284-k15 | 20226 | 25024 | 23.72% | 24791 | 22.57% | 23504 | 16.21% |
| B-n45-k5 | 751 | 772 | 2.80% | 775 | 3.20% | 771 | 2.66% | X-n289-k60 | 95151 | 106073 | 11.48% | 104253 | 9.57% | 107238 | 12.70% |
| B-n45-k6 | 678 | 736 | 8.55% | 745 | 9.88% | 736 | 8.55% | X-n294-k50 | 47161 | 54318 | 15.18% | 53458 | 13.35% | 54899 | 16.41% |
| B-n50-k7 | 741 | 767 | 3.51% | 765 | 3.24% | 753 | 1.62% | X-n298-k31 | 34231 | 40064 | 17.04% | 39609 | 15.71% | 41296 | 20.64% |
| B-n50-k8 | 1312 | 1347 | 2.67% | 1330 | 1.37% | 1328 | 1.22% | X-n303-k21 | 21736 | 26078 | 19.98% | 25228 | 16.07% | 25380 | 16.76% |
| B-n52-k7 | 747 | 762 | 2.01% | 762 | 2.01% | 763 | 2.14% | X-n308-k13 | 25859 | 30557 | 18.17% | 31927 | 23.47% | 31625 | 22.30% |
| B-n56-k7 | 707 | 740 | 4.67% | 744 | 5.23% | 734 | 3.82% | X-n313-k71 | 94043 | 106936 | 13.71% | 101767 | 8.21% | 116306 | 23.67% |
| B-n57-k7 | 1153 | 1153 | 0.00% | 1175 | 1.91% | 1162 | 0.78% | X-n317-k53 | 78355 | 96382 | 23.01% | 84483 | 7.82% | 106138 | 35.46% |
| B-n57-k9 | 1598 | 1651 | 3.32% | 1645 | 2.94% | 1644 | 2.88% | X-n322-k28 | 29834 | 35987 | 20.62% | 35503 | 19.00% | 37562 | 25.90% |
| B-n63-k10 | 1496 | 1537 | 2.74% | 1589 | 6.22% | 1572 | 5.08% | X-n327-k20 | 27532 | 33039 | 20.00% | 33478 | 21.60% | 34083 | 23.79% |
| B-n64-k9 | 861 | 937 | 8.83% | 931 | 8.13% | 923 | 7.20% | X-n331-k15 | 31102 | 36123 | 16.14% | 37292 | 19.90% | 37114 | 19.33% |
| B-n66-k9 | 1316 | 1353 | 2.81% | 1374 | 4.41% | 1350 | 2.58% | X-n336-k84 | 139111 | 153850 | 10.60% | 150341 | 8.07% | 158211 | 13.73% |
| B-n67-k10 | 1032 | 1070 | 3.68% | 1115 | 8.04% | 1065 | 3.20% | X-n344-k43 | 42050 | 48339 | 14.96% | 48035 | 14.23% | 49217 | 17.04% |
| B-n68-k9 | 1272 | 1337 | 5.11% | 1339 | 5.27% | 1343 | 5.58% | X-n351-k40 | 25896 | 30923 | 19.41% | 30498 | 17.77% | 30965 | 19.57% |
| B-n78-k10 | 1221 | 1306 | 6.96% | 1311 | 7.37% | 1307 | 7.04% | X-n359-k29 | 51505 | 58300 | 13.19% | 59810 | 16.12% | 59431 | 15.39% |
| E-n22-k4 | 375 | 421 | 12.27% | 427 | 13.87% | 433 | 15.47% | X-n367-k17 | 22814 | 30083 | 31.86% | 28335 | 24.20% | 27747 | 21.62% |
| E-n23-k3 | 569 | 621 | 9.14% | 574 | 0.88% | 578 | 1.58% | X-n376-k94 | 147713 | 162451 | 9.98% | 160107 | 8.39% | 173422 | 17.40% |
| E-n33-k4 | 835 | 844 | 1.08% | 845 | 1.20% | 858 | 2.75% | X-n384-k52 | 65928 | 76341 | 15.79% | 76040 | 15.34% | 77891 | 18.15% |
| E-n51-k5 | 521 | 534 | 2.50% | 555 | 6.53% | 546 | 4.80% | X-n393-k38 | 38260 | 45226 | 18.21% | 44953 | 17.49% | 47317 | 23.67% |
| E-n76-k7 | 682 | 708 | 3.81% | 721 | 5.72% | 721 | 5.72% | X-n401-k29 | 66154 | 73618 | 11.28% | 76247 | 15.26% | 73121 | 10.53% |
| E-n76-k8 | 735 | 775 | 5.44% | 770 | 4.76% | 777 | 5.71% | X-n411-k19 | 19712 | 26432 | 34.09% | 25671 | 30.23% | 25525 | 29.49% |
| E-n76-k10 | 830 | 876 | 5.54% | 863 | 3.98% | 868 | 4.58% | X-n420-k130 | 107798 | 123789 | 14.83% | 119818 | 11.15% | 128982 | 19.65% |
| E-n76-k14 | 1021 | 1051 | 2.94% | 1070 | 4.80% | 1058 | 3.62% | X-n429-k61 | 65449 | 75236 | 14.95% | 76115 | 16.30% | 78711 | 20.26% |
| E-n101-k8 | 815 | 876 | 7.48% | 879 | 7.85% | 887 | 8.83% | X-n439-k37 | 36391 | 44326 | 21.80% | 43772 | 20.28% | 47436 | 30.35% |
| E-n101-k14 | 1067 | 1137 | 6.56% | 1150 | 7.78% | 1138 | 6.65% | X-n449-k29 | 55233 | 63887 | 15.67% | 67416 | 22.06% | 66168 | 19.80% |
| F-n45-k4 | 724 | 753 | 4.01% | 747 | 3.18% | 729 | 0.69% | X-n459-k26 | 24139 | 32530 | 34.76% | 31774 | 31.63% | 31437 | 30.23% |
| F-n72-k4 | 237 | 272 | 14.77% | 270 | 13.92% | 268 | 13.08% | X-n469-k138 | 221824 | 267934 | 20.79% | 248139 | 11.86% | 260902 | 17.62% |
| F-n135-k7 | 1162 | 1415 | 21.77% | 1385 | 19.19% | 1478 | 27.19% | X-n480-k70 | 89449 | 100833 | 12.73% | 103101 | 15.26% | 103785 | 16.03% |
| M-n101-k10 | 820 | 974 | 18.78% | 908 | 10.73% | 905 | 10.37% | X-n491-k59 | 66483 | 78531 | 18.12% | 78999 | 18.83% | 80703 | 21.39% |
| M-n121-k7 | 1034 | 1242 | 20.12% | 1181 | 14.22% | 1204 | 16.44% | X-n502-k39 | 69226 | 79183 | 14.38% | 77585 | 12.07% | 78419 | 13.28% |
| M-n151-k12 | 1015 | 1143 | 12.61% | 1116 | 9.95% | 1164 | 14.68% | X-n513-k21 | 24201 | 34479 | 42.47% | 32744 | 35.30% | 39592 | 63.60% |
| M-n200-k16 | 1274 | 1468 | 15.23% | 1464 | 14.91% | 1521 | 19.39% | X-n524-k153 | 154593 | 179926 | 16.39% | 174390 | 12.81% | 193416 | 25.11% |
| M-n200-k17 | 1275 | 1468 | 15.14% | 1473 | 15.53% | 1521 | 19.29% | X-n536-k96 | 94846 | 112396 | 18.50% | 111393 | 17.45% | 111191 | 17.23% |
| P-n16-k8 | 450 | 536 | 19.11% | 455 | 1.11% | 452 | 0.44% | X-n548-k50 | 86700 | 106722 | 23.09% | 109595 | 26.41% | 114193 | 31.71% |
| P-n19-k2 | 212 | 238 | 12.26% | 221 | 4.25% | 221 | 4.25% | X-n561-k42 | 42717 | 53160 | 24.45% | 54559 | 27.72% | 64356 | 50.66% |
| P-n20-k2 | 216 | 244 | 12.96% | 221 | 2.31% | 221 | 2.31% | X-n573-k30 | 50673 | 63498 | 25.31% | 61820 | 22.00% | 57024 | 12.53% |
| P-n21-k2 | 211 | 241 | 14.22% | 231 | 9.48% | 242 | 14.69% | X-n586-k159 | 190316 | 222036 | 16.67% | 214162 | 12.53% | 236527 | 24.28% |
| P-n22-k2 | 216 | 227 | 5.09% | 219 | 1.39% | 248 | 14.81% | X-n599-k92 | 108451 | 127051 | 17.15% | 131764 | 21.50% | 132380 | 22.06% |
| P-n22-k8 | 603 | 767 | 27.20% | 597 | -1.00% | 671 | 11.28% | X-n613-k62 | 59535 | 74314 | 24.82% | 76519 | 28.53% | 82989 | 39.40% |
| P-n23-k8 | 529 | 550 | 3.97% | 545 | 3.02% | 543 | 2.65% | X-n627-k43 | 62164 | 74305 | 19.53% | 76288 | 22.72% | 77838 | 25.21% |
| P-n40-k5 | 458 | 469 | 2.40% | 463 | 1.09% | 474 | 3.49% | X-n641-k35 | 63682 | 75524 | 18.60% | 79364 | 24.63% | 78067 | 22.59% |
| P-n45-k5 | 510 | 518 | 1.57% | 525 | 2.94% | 519 | 1.76% | X-n655-k131 | 106780 | 121331 | 13.63% | 123635 | 15.78% | 286775 | 168.53% |
| P-n50-k7 | 554 | 577 | 4.15% | 576 | 3.97% | 563 | 1.62% | X-n670-k130 | 146332 | 178277 | 21.83% | 175430 | 19.88% | 197324 | 34.85% |
| P-n50-k8 | 631 | 648 | 2.69% | 651 | 3.17% | 653 | 3.49% | X-n685-k75 | 68205 | 85840 | 25.86% | 86689 | 27.10% | 92401 | 35.48% |
| P-n50-k10 | 696 | 729 | 4.74% | 726 | 4.31% | 725 | 4.17% | X-n701-k44 | 81923 | 96856 | 18.23% | 101554 | 23.96% | 99307 | 21.22% |
| P-n51-k10 | 741 | 756 | 2.02% | 774 | 4.45% | 771 | 4.05% | X-n716-k35 | 43373 | 54951 | 26.69% | 55906 | 28.90% | 57471 | 32.50% |
| P-n55-k7 | 568 | 586 | 3.17% | 590 | 3.87% | 588 | 3.52% | X-n733-k159 | 136187 | 163853 | 20.31% | 159532 | 17.14% | 202275 | 48.53% |
| P-n55-k10 | 694 | 707 | 1.87% | 714 | 2.88% | 710 | 2.31% | X-n749-k98 | 77269 | 94552 | 22.37% | 92530 | 19.75% | 101096 | 30.84% |
| P-n60-k10 | 744 | 769 | 3.36% | 769 | 3.36% | 762 | 2.42% | X-n766-k71 | 114417 | 136873 | 19.63% | 140820 | 23.08% | 149744 | 30.88% |
| P-n60-k15 | 968 | 991 | 2.38% | 1003 | 3.62% | 1016 | 4.96% | X-n783-k48 | 72386 | 90822 | 25.47% | 94551 | 30.62% | 96054 | 32.70% |
| P-n65-k10 | 792 | 808 | 2.02% | 820 | 3.54% | 812 | 2.53% | X-n801-k40 | 73305 | 91023 | 24.17% | 94591 | 29.04% | 102682 | 40.08% |
| P-n70-k10 | 827 | 866 | 4.72% | 876 | 5.93% | 864 | 4.47% | X-n819-k171 | 158121 | 184644 | 16.77% | 182548 | 15.45% | 417753 | 164.20% |
| P-n76-k4 | 593 | 639 | 7.76% | 645 | 8.77% | 649 | 9.44% | X-n837-k142 | 193737 | 224297 | 15.77% | 231397 | 19.44% | 285547 | 47.39% |
| P-n76-k5 | 627 | 680 | 8.45% | 673 | 7.34% | 662 | 5.58% | X-n856-k95 | 88965 | 106823 | 20.07% | 112092 | 26.00% | 128899 | 44.89% |
| P-n101-k4 | 681 | 751 | 10.28% | 746 | 9.54% | 773 | 13.51% | X-n876-k59 | 99299 | 122331 | 23.19% | 123350 | 24.22% | 117776 | 18.61% |
| X-n101-k25 | 27591 | 29873 | 8.27% | 29574 | 7.19% | 30455 | 10.38% | X-n895-k37 | 53860 | 72775 | 35.12% | 77568 | 44.02% | 86211 | 60.06% |
| X-n106-k14 | 26362 | 27868 | 5.71% | 27583 | 4.63% | 26996 | 2.40% | X-n916-k207 | 329179 | 378802 | 15.07% | 375026 | 13.93% | 429299 | 30.42% |
| X-n110-k13 | 14971 | 15970 | 6.67% | 16196 | 8.18% | 16348 | 9.20% | X-n936-k151 | 132715 | 167857 | 26.48% | 172305 | 29.83% | 175681 | 32.37% |
| X-n115-k10 | 12747 | 14190 | 11.32% | 14323 | 12.36% | 13533 | 6.17% | X-n957-k87 | 85465 | 111777 | 30.79% | 121909 | 42.64% | 116564 | 36.39% |
| X-n120-k6 | 13332 | 14381 | 7.87% | 14078 | 5.60% | 14157 | 6.19% | X-n979-k58 | 118976 | 146052 | 22.76% | 142602 | 19.86% | 145171 | 22.02% |

# F   Supplementary Material References

We repeat the references here for reader convenience since PDF files (main paper / supplementary) are separate upon submission. This section also adds Supplementary-only ones.