# OpenReview forum: "RL4CO: an Extensive Reinforcement Learning for Combinatorial Optimization Benchmark"
_NeurIPS.cc/2024/Datasets_and_Benchmarks_Track — Submitted to NeurIPS 2024 Track Datasets and Benchmarks_

### Official Review · Reviewer_Vcfw · 2024-07-16
**A Benchmark for Neural Combinatorial Optimization Research (Updated)**

**Rating:** 7
**Confidence:** 4

**Review:**

This is a much-needed benchmark for research in neural combinatorial optimization, since it has many different CO problems that can be tested and various techniques for solving each of the problems. This reviewer is not aware of a comparable set of benchmarks for solving this problem and the use of ML techniques for combinatorial optimization problems is a growing research topic in operations research.

The benchmark should be useful to researchers in the field based on the examples that are included. The key contributions could use more description although there are ample examples to show how the benchmark can be used to develop RL models that produce better solutions.

In summary, the benchmark seems to be useful to the CO community and the examples show promising results, but the paper omits key information in describing the benchmark.

**Strengths:**

The benchmark consists of many practical problems for which RL solutions have been proposed. There is the ability to combine these in novel ways to generate solutions that improve on existing results and examples of doing so are shown. The motivation for providing such a benchmark is demonstrated by the user community that has already been attracted to the Slack workspace. This should be very useful for those in the Operations Research community, and related areas of research, who are exploring ML techniques for optimizing these challenging NP-hard problems.

The paper presents the core goals clearly, and adequately addresses the limitations of the existing work.

**Additional Feedback:**

This review has been revised based on feedback received during the rebuttal period.

**Clarity:**

The paper is well-written, but the coverage is not typical for this sort of paper. It does seem to go into a lot of detail that would best be included in a tutorial, when more important information about the benchmarks could be included.

There are a few places where the paper could be improved. Places that reference the "Appendix" should read "the Appendix" not "Appendix".

On line 59,  "Paradigms" is misspelled "Paragiams"
On line 235, it seems that "improve upon them" is included earlier in the sentence.

**Correctness:**

The evaluation methods show that the benchmarks are productive in finding solutions that are closer to optimal than existing solutions. The experiments are described in detail and appear to be performed properly.

**Documentation:**

The documentation is adequate, although it might be helpful in the Readme to provide more clarity about which benchmarks are where. The license, etc. is provided but it's not clear that all the GitHub projects that are linked have or will have the same license. It would be helpful to users to include information on the licenses for contributed projects -- this would allow users to decide whether the restrictions, if any, might impact their use. The maintenance plan relies on contributors to maintain their GitHub repositories.

**Ethics:**

There do not appear to be any ethical concerns. The data is derived from existing problems, presumably none of these have personally identifiable data.

**Limitations:**

The authors correctly point out that the benchmark is useful for reinforcement learning but isn't directly applicable to other ML approaches, and it isn't practical to create a comprehensive benchmark that can be used with all problems. (Note: This comment was an observation and didn't factor into the evaluation. All benchmarks have some limitations. Agree that it is possible to extend the benchmark.)

There don't seem to be an negative societal impacts, so this isn't an issue.

**Opportunities For Improvement:**

The paper gets into too much detail, such as what commands to use with which Python scripts to produce results. The authors should consider whether putting this information in an appendix or linking to a tutorial might be better and providing more details about the benchmarks, more examples or enlarging some figures might be a better use of the space. For instance, it is difficult to find some of the decoding schemes in Figure 4.

**Relation To Prior Work:**

The related work is discussed sufficiently.

**Summary And Contributions:**

This paper presents an extensive benchmark for reinforcement learning for combinatorial optimization based on existing problems and incorporates a number of existing techniques for solving these problems. As explained in the paper, training a RL model to solve multiple problems leads to better overall performance. Some examples of benchmark studies are included to illustrate how the benchmark has been used to find better solutions.

The contents of the paper are atypical of papers that present datasets and benchmarks. For instance, there are detailed examples with Python commands for running the benchmarks, but the figures that show the results are often too small to see some details.

This reviewer still holds the opinion that Python commands etc. belong in the appendix, especially if the paper is struggling to fit in the space allotted. The GitHub repository should provided documentation/examples on running the code.

---

> ### Author Rebuttal · Authors · 2024-08-16
>
> Thanks for your review. We will address your concerns point by point.
>
>
> We will start by replying to the most critical point:
>
> > Basic information about the dataset, such as which methods are included in the dataset, are not included in the paper [...] the paper omits key information in describing the dataset [...]the paper omits more important information about the datasets that should be included.
>
>
> The necessary descriptions are explicitly detailed in our paper. Section 4.5 of our manuscript clearly lists the CO environments and baseline methods. The [supplementary material](https://openreview.net/pdf?id=Becrgm5xAq) provides extensive details, including Appendix B for environment descriptions and Appendix C for baseline descriptions, spanning almost 20 pages. Extensive details are available not only throughout the text but also through [our online documentation](https://rl4.co/). We finally remark that this paper does not introduce new "datasets" but is rather a *benchmark* of RL environments and baselines for combinatorial optimization.
>
>
> > The authors correctly point out that the benchmark is useful for reinforcement learning but isn't directly applicable to other ML approaches, and it isn't practical to create a comprehensive benchmark that can be used with all problems.
>
> While RL4CO was built around RL, it could be easily extended to other ML approaches in the future. For example, Supervised Learning (SL) approaches are very easy to implement, compared to RL, since they simply require input and output data. This can be easily managed in the future by creating a custom `DataModule` with PyTorch Lightning to load such instances.
>
>
> > The contents of the paper are atypical of papers that present datasets and benchmarks. For instance, there are detailed examples with Python commands for running the benchmarks [...] The paper gets into too much detail, such as what commands to use with which Python scripts to produce results [...] but the coverage is not typical for this sort of paper. It does seem to go into a lot of detail that would best be included in a tutorial.
>
> Several papers, particularly the ones introducing RL libraries, include snippets of code, commands and details on how to run the code. One reason for this is that there is no single "one-fits-all" RL library - although we believe TorchRL seems to be a good candidate - due to several different advantages and disadvantages of each implementation, and as such, providing clear examples in the main text is helpful for readers. For example, [1r] shows at page 2 configuration files, implementation and logs alongside bash command to reproduce the results. [2r] on page 4 shows a snippet of code that is very similar to ours in RL4CO's Section 4.4. [3r] shows not only a snippet similar to ours at page 4, but also another snippet at page 6 showing how to utilize such code. Note that these example papers were published in NeurIPS Datasets and Benchmark Track 2023.
>
>
> > The figures that show the results are often too small to see some details. [...] For instance, it is difficult to find some of the decoding schemes in Figure 4.
>
> Thanks for this feedback, we agree with you. We will ensure that the figures will be more easily readable in the camera-ready version.
>
> > The license, etc. is provided but it's not clear that all the GitHub projects that are linked have or will have the same license.
>
> If you mean what license new projects based on RL4CO will have, this is totally free and up to new projects! We believe in open and _libre_ software and another project can have any other license. The only condition of our license, the MIT one, is the following: "*The above copyright notice and this permission notice shall be included in all copies or substantial portions of the Software*".
>
>
> > There are a few places where the paper could be improved. Places that reference the "Appendix" should read "the Appendix" not "Appendix". On line 59, "Paradigms" is misspelled "Paragiams" On line 235, it seems that "improve upon them" is included earlier in the sentence.
>
> Thanks for spotting these typos! We will fix them in the camera-ready version.
>
>
> ---
>
> ### References
>
> [1r] Tarasov, Denis, et al. "CORL: Research-oriented deep offline reinforcement learning library." Advances in Neural Information Processing Systems 36 (2024).
>
> [2r] Ji, Jiaming, et al. "Safety gymnasium: A unified safe reinforcement learning benchmark." Advances in Neural Information Processing Systems 36 (2024).
>
> [3r] Kurenkov, Vladislav, et al. "Katakomba: tools and benchmarks for data-driven NetHack." Advances in Neural Information Processing Systems 36 (2024).

---

> > ### Author Response · Authors · 2024-08-20
> > **Thanks for your update!**
> >
> > We deeply appreciate your review update and the increase in score!
> >
> > We will provide additional answers to the updated parts here:
> >
> > > This reviewer still holds the opinion that Python commands etc. belong in the appendix, especially if the paper is struggling to fit in the space allotted.
> >
> > Following up on our previous response: we will try to limit the use of example commands in the main text, as you suggested, in the camera-ready version.
> >
> > > The GitHub repository should provided documentation/examples on running the code.
> >
> > We totally agree that a project like ours should be provided with comprehensive and clear documentation for the community to grow! We have provided extensive documentation on the [RL4CO website](https://rl4.co/) as well as several tutorials and examples [here](https://rl4.co/examples/1-quickstart/) that can be easily run on consumer hardware.
> >
> > > The license, etc. is provided but it's not clear that all the GitHub projects that are linked have or will have the same license. It would be helpful to users to include information on the licenses for contributed projects -- this would allow users to decide whether the restrictions, if any, might impact their use.
> >
> > We agree with you that it may not have been clear exactly how the license would affect future projects based on RL4CO. We have just added a new "Licensing" section under the "About" tab in documentation ([link](https://rl4.co/docs/content/general/licensing/)), which we will report here:
> >
> > *Our library is released under the [MIT License](https://opensource.org/license/MIT), which is an open and permissive license. This means:*
> >
> > - *You can use, modify, and distribute the code without any restrictions, even for commercial purposes.*
> > - *Your projects will not inherit any additional limitations from our library, even if you modify or extend it.*
> >
> > *All contributions to the library are covered by the MIT License, ensuring that everything is free to use under the same open terms. A copy of the license is available [here](https://github.com/ai4co/rl4co/blob/main/LICENSE).*
> >
> >
> > We hope our response addresses your remaining concerns, and we sincerely appreciate your continued support!

---

### Official Review · Reviewer_su7y · 2024-07-22
**Good presentation but lacks technique depth**

**Rating:** 4
**Confidence:** 3
**Correctness:** yes
**Clarity:** yes

**Review:**

good presentation

**Strengths:**

authors introduce RL4CO, a unified and extensive benchmark

**Additional Feedback:**

depth

**Documentation:**

yes

**Limitations:**

1,In section 3, I did not see the intrinsic models in RL4CO. I see several eqns which are related to general RL algorithms.
2, After I read this paper, it looks like the summary or report of engineering, and I did not see the depth or intrinsic characteristics in RL4CO. For example “They are based on the RL4COEnvBase class that extends from the EnvBase in TorchRL”, “Hydra additionally allows for automatically parsing parameters (un-)defined in configs - i.e., python run.py190 experiment=routing/pomo env=cvrp env.generator_params.num_loc=50”, which should be put to the readme of github or docs. The codes in Sec 4.4 can be moved to appendix.
3,authors said that “RL4CO enhances the efficiency of environments”, I did not see the implementation details. Authors should list them.
4, In section 4, authors describe the lib in terms of environments, policies, RL algs, utilities, and environments. I suggest to describe the kernel of RL4CO, not several points without focus.
5, In section 5, the implementation details including datasets are not described in detail. The compared methods are not enough. For example, in TSP, the compared methods include conventional methods (such as christofides algorithm, GA, SA, cheapest insertion, etc. ) and ML/RL methods. Therefore, the performance evaluation is weak.
6, there are too many references.
7, authors said that this lib covers 23 state-of-the-art methods and more than 20 CO problems. These should be listed and described.

**Opportunities For Improvement:**

depth

**Relation To Prior Work:**

yes

**Summary And Contributions:**

The field lacks a unified benchmark for easy development and standardized comparison of algorithms across diverse CO problems. To fill this gap, authors introduce RL4CO, a unified and extensive benchmark with library coverage of 23 state-of-the-art methods and more than 20 CO problems. However, these are not listed and described. Authors focus on describing the components of the lib not the methods or kernel of this RL4CO. I think it is not like a paper but a summary of the lib.

---

> ### Author Rebuttal · Authors · 2024-08-16
>
> Thanks for your feedback.
>
> We think there have been several misunderstandings. We will provide answers to your concerns in the following rebuttal.
>
> The most critical issue raised regards some supposedly missing listing and description of the methods and CO problems:
>
> > Authors introduce RL4CO [...] with coverage of 23 state-of-the-art methods and more than 20 CO problems. However, these are not listed and described. [...] I did not see the intrinsic models in RL4CO. [...]  authors said that this lib covers 23 state-of-the-art methods and more than 20 CO problems. These should be listed and described.
>
> The necessary descriptions are explicitly detailed in our paper. Section 4.5 of our manuscript clearly lists the CO environments and baseline methods. The [supplementary material](https://openreview.net/attachment?id=Becrgm5xAq&name=supplementary_material) provides extensive details, including Appendix B for environment descriptions and Appendix C for baseline descriptions, spanning almost 20 pages. Extensive details are available not only throughout the text but also through [our online documentation](https://rl4.co/).
>
> > After I read this paper, it looks like the summary or report of engineering, and I did not see the depth or intrinsic characteristics in RL4CO. For example “They are based on the RL4COEnvBase class that extends from the EnvBase in TorchRL”, “Hydra additionally allows for automatically parsing parameters (un-)defined in configs - i.e., python run.py190 experiment=routing/pomo env=cvrp env.generator_params.num_loc=50”, which should be put to the readme of github or docs. The codes in Sec 4.4 can be moved to appendix. [...] I think it is not like a paper but a summary of the lib.
>
>
> We will cite for context the "scope" section of the [Call for NeurIPS 2024 Datasets and Benchmarks](https://neurips.cc/Conferences/2024/CallForDatasetsBenchmarks#:~:text=SCOPE.%C2%A0This%20track%20welcomes,yielding%20important%20new%20insight.):
>
> "This track welcomes all work on data-centric machine learning research (DMLR) and *open-source libraries and tools* that enable or accelerate ML research, covering ML datasets and *benchmarks* as well as *algorithms, tools, methods, and analyses* for working with ML data. This includes but is not limited to [...] *data generators and reinforcement learning environments* [...] and *benchmarking tools*."
>
> We believe the inclusion of code snippets and description of the library implementation is functional to the audience of the NeurIPS Datasets and Benchmarks track. Several papers, particularly the ones introducing RL libraries, include snippets of code, commands and details on how to run the code. One reason for this is that there is no single "one-fits-all" RL library - although we believe TorchRL seems to be a good candidate - due to several different advantages and disadvantages of each implementation, and as such, providing clear examples in the main text is helpful for readers. For example, [1r] shows at page 2 configuration files, implementation and logs alongside bash command to reproduce the results. [2r] at page 4 shows a snippet of code which is very similar to ours in RL4CO's Section 4.4. [3r] shows not only a snippet similar to ours at page 4, but also another snippet at page 6 showing how to utilize such code. Note that these example papers were published in NeurIPS Datasets and Benchmark Track 2023.
>
> > Authors said that “RL4CO enhances the efficiency of environments”, I did not see the implementation details.
>
> As mentioned throughout the text (for instance, Section 2) we make our environments parallelizable and runnable on hardware accelerator as GPUs, thus overcoming a critical bottleneck of data collection in RL. Section A.2 of the Appendix further provides motivation to our software choices. Section E.7.3 in Appendix provides additional details of our even more efficient implementation of TorchRL environments for CO.
>
> > The compared methods are not enough. For example, in TSP, the compared methods [does not] include conventional methods (such as christofides algorithm, GA, SA, cheapest insertion, etc. ) and ML/RL methods. Therefore, the performance evaluation is weak.
>
> RL4CO is a benchmark of RL methods. We include necessary comparisons with the SotA classical baselines such as Concorde and LKH for TSP. Other algorithms you mentioned are much older and outdated (i.e., the Christofides Algorithm was published in 1978), and we do not believe they provide meaningful insights in this context.
>
> > There are too many references.
>
> We have invested significant time and resources in researching and creating RL4CO based on the abundant literature of NCO. Our benchmark is extensive both in the collection of methods as well as in our 70+ pages long manuscript. Therefore, we believe the number of references (less than 160) is adequate for a paper of our scope.
>
>
> ---
>
> We hope the clarifications above address your concerns and demonstrate the depth and comprehensiveness of our work. We respectfully ask that you reconsider your evaluation, particularly in light of the detailed descriptions, implementation insights, and extensive comparisons we have included in both the main text and supplementary materials. We believe RL4CO aligns well with the scope and expectations of the NeurIPS Datasets and Benchmarks track. Your reevaluation would be greatly appreciated, and we are grateful for your thoughtful consideration.
>
> We remain available for any further clarification and would be happy to provide additional details if needed.
>
>
>
> ---
>
> ### References
>
>
> [1r] Tarasov, Denis, et al. "CORL: Research-oriented deep offline reinforcement learning library." NeurIPS 2023.
>
> [2r] Ji, Jiaming, et al. "Safety gymnasium: A unified safe reinforcement learning benchmark." NeurIPS 2023.
>
> [3r] Kurenkov, Vladislav, et al. "Katakomba: tools and benchmarks for data-driven NetHack." NeurIPS 2023.

---

> > ### Comment · Reviewer_su7y · 2024-08-23
> > **response to rebuttal**
> >
> > thanks for your rebuttal.
> >
> > After reading the paper, I think authors provide a lib for RL4CO, which is contributes to the community. The website is also clear.
> >
> > I suggest to lists the details of how RL algs solve CO problems in your lib in the appendix. For example, the state transition under some action.
> >
> > In addition, how to run the algs, including param setting, should be shown in detail. how to see the results.

---

> > > ### Author Response · Authors · 2024-08-26
> > >
> > > Thank you for your update and new feedback.
> > >
> > > In the following response, we would like to clarify the additional points raised during the rebuttal period. Please note that since the manuscript (both main and appendix) cannot be updated during the rebuttal period, we will provide additional context through our online documentation, which we believe will also enhance the experience for fellow NCO researchers.
> > >
> > >
> > > ### About additional details for state transitions
> > >
> > > While Section B of the Appendix introduces each environment in RL4CO, we agree that a general state transition, as described in Section 3 of the main paper, could be provided.
> > >
> > > We have uploaded a new, updated version of the Environment Overview in our online documentation (link [here](https://rl4.co/docs/content/intro/environments/)), in which we introduce a problem description for each problem class (routing, scheduling, electronic design automation, graph), alongside the state transitions and links to relevant API documentation for each problem.
> > >
> > >
> > > ### About additional experimental details
> > >
> > > The general benchmarking setup is available in Appendix D, in which, specifically, Appendix D.3 describes hyperparameter settings for each experiment in the main text. Hyperparameters for additional experiments are also available in Appendix E. In practice, we note that experimental configurations can be additionally accessed through the `configs` folder (link [here](https://github.com/ai4co/rl4co/tree/main/configs)) through `.yaml` files.
> > >
> > > Reproducing the results can be done easily thanks to Hydra: since we provide the configurations, one can run configurations under the `configs/experiment` folder and then change the hyperparameters. For instance: `python run.py experiment=routing/am env=cvrp env.generator_params.num_loc=50` will run the Attention Model configuration on the CVRP with 50 nodes. We additionally provide a new walkthrough tutorial (link [here](https://rl4.co/docs/content/start/hydra/)) on Hydra and an explanation of how to reproduce and how to run new experiments by changing a configuration.
> > >
> > > We will also clarify the points you raised when we can modify the paper for the camera-ready version.
> > >
> > > ---
> > >
> > > We hope our response addresses your remaining concerns. We sincerely appreciate your feedback and support.

---

> > > > ### Comment · Reviewer_su7y · 2024-08-30
> > > > **Thanks for your rebuttal**
> > > >
> > > > Thanks for your rebuttal.
> > > >
> > > > Although the presentation is good. I still think this paper lacks the technique in depth, which is the important contributions in the AI4CO community. For example, your paper mentioned the efficiency of environments (line 152), and scalable architectures (line 320); however, your work did not try to solve these important problems. In fact, there are several libs such as Jumanji for this community. Therefore, I did not accept it.
> > > >
> > > > Jumanji (ICLR 2024) https://github.com/instadeepai/jumanji

---

> > > > > ### Author Response · Authors · 2024-08-31
> > > > >
> > > > > > I still think this paper lacks the technique in depth
> > > > >
> > > > > We believe we have already explained in great length our contributions throughout the main text, supplementary material, and rebuttals, and importantly evidenced through the recognition of RL4CO on Github. If not, can you please point out what "technique" you are referring to?
> > > > >
> > > > > > Your paper mentioned the efficiency of environments (line 152), and scalable architectures (line 320); however, your work did not try to solve these important problems.
> > > > >
> > > > > We did solve these problems. We improved the efficiency of environments by making them batched and runnable on GPU with TorchRL. Scalable architectures are benchmarked, including in the main text in Section 5.
> > > > >
> > > > > > In fact, there are several libs such as Jumanji for this community. Therefore, I did not accept it.
> > > > >
> > > > > We have already cited all relevant libs to the best of our knowledge, including Jumanji, and provided the differences with them in the Related Works in Section 2. RL4CO offers more environments and baselines than any existing similar lib. Furthermore, our baselines are closely aligned with state-of-the-art methods. E.g., Jumanji has some actor-critic models and interfaces with general RL procedures, but does not integrate powerful RL approaches specifically for CO. The table below further shows the differences:
> > > > >
> > > > > | Library                                             | Environments \# | Baselines \# | Hardware Acceleration | Availability | Modular Baselines | Open Community |
> > > > > |-----------------------------------------------------|-----------------|----------------------|-----------------------|--------------|------------------|----------------|
> > > > > | ORL                      | 3               | 1                    | ✗                     | ✗            | ✗                | ✗              |
> > > > > | OR-Gym                    | 9               | -                    | ✗                     | ✓            | ✗                | ✗              |
> > > > > | Graph-Env            | 2               | -                    | ✗                     | ✓            | ✗                | ✗              |
> > > > > | RLOR                        | 2               | 2                    | ✗                     | ✓            | ✓                | ✗              |
> > > > > | RoutingArena           | 1               | 8                    | ✓                     | ✗            | ✗                | ✗              |
> > > > > | Jumanji                   | 22              | 3                    | ✓                     | ✓            | ✗                | ✗              |
> > > > > | **RL4CO**                                    | **27**| **23**               | **✓**                 | **✓**        | **✓**            | **✓**          |
> > > > >
> > > > > Reporting Section 2 of our paper:
> > > > >
> > > > > *Despite the variety of general-purpose RL software libraries, there is a lack of a unified and extensive benchmark for CO problems. ORL proposes an RL benchmark for Operations Research (OR) with a PPO baseline; OR-Gym and Graph-Env provide a collection of OR environments. RLOR proposes a general-purpose library for OR and tests on the canonical TSP and CVRP environments. However, a major downside of the above libraries is that they cannot be massively parallelized due to their reliance on the OpenAI Gym API, which can only run on CPU, unlike ours, which is based on TorchRL, a recent official PyTorch library for RL that enables hardware-accelerated execution of both environments and algorithms. [...] We also mention Routing Arena, whose scope is different from ours, namely, comparing NCO and classical solvers only for the CVRP. The most related work is Jumanji, which provides a variety of CO environments written in JAX that can be hardware-accelerated alongside an actor-critic baseline. While Jumanji is an RL environment suite, ours is a full-stack library that integrates environments, policies, and RL algorithms under a unified framework.*
> > > > >
> > > > >
> > > > > Overall, compared to previous libraries, we have more environments and more baselines. Also, we make baselines modular under a unified framework and have an active open community.
> > > > >
> > > > > As per the above, we are confident there is enough evidence in favor of the contribution of RL4CO.
> > > > >
> > > > >
> > > > > **We also noticed that the score changed from an initial "3" for the initial review - where we addressed the concerns - then raised to a "5" after the first answer - where we addressed the *new* concerns - and then lowered to a "4" with further *new* concerns - which we have just addressed above.**
> > > > >
> > > > > We are surprised you decreased your score from 5 to 4 after our response.  Can you please elaborate on which new concerns have led to this decision?
> > > > > If you have specific limitations that we did not address, we invite you to provide them. Please keep in mind that the discussion period is ending soon. We will address them as soon as possible.
> > > > >
> > > > > As we have invested significant effort and diligence in preparing this paper, we hope this rebuttal will be satisfactory.
> > > > >
> > > > > Thank you in advance for your cooperation.

---

> > > > > > ### Comment · Reviewer_su7y · 2024-09-01
> > > > > >
> > > > > > Jumanji supports CO problems, e.g., TSP, Knapsack, Job Shop Scheduling, and CVRP, which is a suite of scalable RL lib . I do not quite agree with the differences between RL4CO and Jumanji authors listed, and I still think RL4CO is very similar to Jumanji.
> > > > > > https://github.com/instadeepai/jumanji
> > > > > >
> > > > > > The score changes since finally I hope authors can solve the important problems in this community I just mentioned, not just describe them in literature.

---

> > > > > > > ### Author Response · Authors · 2024-09-01
> > > > > > >
> > > > > > > **This review comment came 14 minutes before the rebuttal ended, and we are answering just now, 3 minutes before the deadline.**
> > > > > > >
> > > > > > > >  I do not quite agree with the differences between RL4CO and Jumanji authors listed, and I still think RL4CO is very similar to Jumanji.
> > > > > > >
> > > > > > > We disagree with your opinion. You should clearly write down in what you do not agree with.
> > > > > > >
> > > > > > > > The score changes since finally I hope authors can solve the important problems in this community I just mentioned, not just describe them in literature.
> > > > > > >
> > > > > > > We did not just describe them in the literature, but we have provided them in the code implementation.

---

> > > > > > > > ### Comment · Reviewer_su7y · 2024-09-02
> > > > > > > >
> > > > > > > > I think Jumanji has Modular Baselines, and Open Community, while authors put an X mark in the above table. So I disagree with the differences authors listed.
> > > > > > > >
> > > > > > > > authors said " Jumanji has some actor-critic models and interfaces with general RL procedures, but does not integrate powerful RL approaches specifically for CO". We only focus on if current libs such as Jumanji can solve CO problems well. Jumanji can be used in games and CO problems. Jumanji integrate powerful RL methods, and users easily connect RL frameworks and libraries such as Stable Baselines3, RLlib, OpenAI Gym and DeepMind-Env.

---

> ### Author Response · Authors · 2024-09-02
> **Final Clarifications**
>
> We would like to clarify that the scopes of Jumanji and RL4CO are fundamentally different. Jumanji is described as 'a diverse suite of scalable reinforcement learning environments in JAX' (from their GitHub description), whereas RL4CO is a comprehensive framework specifically designed for CO in PyTorch, with a strong focus on NCO baselines. Moreover, while Jumanji is implemented in JAX, the majority of prevailing NCO papers are based on PyTorch, making RL4CO more aligned with the needs of the NCO community, where we have modularized key baselines using PyTorch in RL4CO.
>
> We recognize Jumanji's merits, and we do believe it is a useful tool to explore some CO problems and games from a general RL perspective, which is why it can be connected to RL frameworks such as Stable Baselines3 as you correctly mention.
>
> We note however, about the modular baselines, that Jumanji does not include powerful problem-specific ones, such as specific architecture for improvement methods to mention one example. You may see that each network for each problem in Jumanji is not modular, i.e., they are separated even in different folders such as `jumanji/training/networks/cvrp/` (link [here](https://github.com/instadeepai/jumanji/tree/main/jumanji/training/networks/cvrp)) and they only use a generic transformer (see at line 29 [here](https://github.com/instadeepai/jumanji/blob/fd511b4523425bdcc69f2fd3c8869cefddd49f1d/jumanji/training/networks/cvrp/actor_critic.py#L29)) for each problem. Note that this is not a criticism of Jumanji: the authors of Jumanji do not claim that Jumanji is a baseline library for CO, rather, as they described, Jumanji is a general-purpose environment library containing CO problems.
>
> For example, it is challenging to train a foundation NCO model capable of solving 16 variants of VRP problems using Jumanji, as seen in the latest NCO papers. However, RL4CO easily accomplishes this, as demonstrated in Table 10 of the appendix. Additionally, RL4CO allows us to bootstrap an improvement solver with the constructive NCO solver effectively as shown in the main paper where Jumanji has no implementations of key baselines and methods in its library.
>
> About the open community: we are referring to the AI4CO community (Slack invitation link available [here](https://bit.ly/ai4co-slack), which has gathered more than 190 diverse researchers and practitioners at the time of writing). We are not aware of such an effort by Jumanji's team, which we note is under InstaDeep AI, which is a for-profit multi-million dollar company. Conversely, AI4CO is a non-profit open research group which welcomes everyone, including people from Jumanji.
>
>
> A side note, we believe that these concerns/questions should ideally have been pointed out at the beginning of the review and rebuttal process. This would have allowed us sufficient time to respond and address them thoroughly. Raising these issues at the end of the rebuttal period is not conducive to a productive and constructive review process in our view.
>
> We will follow the suggestion to expand our discussion and make the comparison with Jumanji more clear in the final paper. We hope you can make an independent and objective assessment of the overall contribution of our RL4CO work. We should remind that the system will not allow us to respond any further since the author rebuttal is about to end. Thank you for your cooperation and support.

---

### Official Review · Reviewer_7ic3 · 2024-07-24
**Excellent work.**

**Rating:** 7
**Confidence:** 3
**Correctness:** Yes
**Clarity:** yes. very  easy to read.

**Review:**

1. Large number of problems and baselines are taken care of varying from routing, scheduling, coverage,
2.  Evaluation results are shown in appendix on a large number of baselines and baseline variants.

Overall, RL4CO seems like a valuable contribution to the field of applying RL to CO problems. The focus on modularity, efficiency, and extensive coverage makes it a promising tool for researchers in this area.

**Strengths:**

See review.
1. Very easy to read and understand. All terms explained very clearly.
2. Large number of problems and baselines.
3. Explanation of different problems, instance generators, different distributions etc.
4. Visualization of different instances/ problems and distributions
5. Sample efficiency results are also added.

**Additional Feedback:**

NA

**Documentation:**

Yes seems correct.

**Limitations:**

Yes

**Opportunities For Improvement:**

1. In TSP only size distribution is considered and not spatial distirbution?

Line 602:
Following the standard protocol of NCO for routing, we randomly sample node coordinates from the 2D unit square (i.e., [0, 1]^2).

See below work which shows different spatial distributions and its impact on different Neural CO solvers.

[A] Manchanda, S., Michel, S., Drakulic, D., & Andreoli, J. M. (2022, September). On the generalization of neural combinatorial optimization heuristics. In Joint European Conference on Machine Learning and Knowledge Discovery in Databases (pp. 426-442). Cham: Springer Nature Switzerland.



2. Similar to point 1 shouldn't the library support other distribution for other problems such as in MCP( sampling weights from different distributions etc.)

**Relation To Prior Work:**

yes

**Summary And Contributions:**

The authors propose RL4CO  a unified benchmark for applying reinforcement learning (RL) to solve combinatorial optimization (CO) problems. It provides a standardized framework for developing and comparing different RL algorithms and CO problem environments.  It has coverage of 23 state-of-the-art methods and more than 20 CO problems.

The contributions are:
Modularity: Easy to combine different components.
Efficiency: Faster training and lower memory usage compared to similar tools.
Comprehensive: Includes 20 CO problems and 23 baseline models.

---

> ### Author Rebuttal · Authors · 2024-08-16
>
> Thank you for your constructive feedback in reviewing our paper and appreciation of our strenghts. We are especially grateful for your acknowledgement of the main strenghts of RL4CO: modularity, efficiency, and comprehensiveness.
>
> We are happy to address the opportunities for improvement you pointed out  in the following.
>
> > 1. In TSP only size distribution is considered and not spatial distribution?
>
> We consider both size and distribution shifts. In the Supplementary Material at Section E.9 (Page 66) we include several generalization studies on CVRP, a practical extension of the TSP, including different spatial distributions as per [1r]. In Table 32, training on multiple spatial distributions (MDPOMO, Multi-Distribution POMO) is shown to improve the generalization performance in set A and B of CVRPLib.
>
> > Line 602: Following the standard protocol of NCO for routing, we randomly sample node coordinates from the 2D unit square (i.e., [0, 1]^2).
>
> Thanks for spotting this inaccuracy. This refers to the main routing experiments only; we will clarify this in the text. We sample by default in the main routing experiments from the uniform distribution and in the unit square for routing ($[0, 1]^2$) since this is a common practice to have the data normalized. However, in other generalization experiments such as the ones mentioned above, we change this sampling to different distributions.
>
>
> > 2. Similar to point 1 shouldn't the library support other distribution for other problems such as in MCP( sampling weights from different distributions etc.)
>
> Thanks for pointing this out! We started out supporting different distributions for routing problems initially, since it is more common for such problems to be studied with different distributions. However, we plan to extend the support across all problems of RL4CO.
>
> We implemented sampling for MCP, as you mentioned as an example. We additionally added a new notebook in the documentation at this [link](https://rl4.co/examples/other/3-data-generator-distributions/) for exploring data generation in RL4CO with different distributions, such as size and different spatial locations - this also includes the MCP example in which weights are sampled from different distributions. We believe this new notebook will be useful for the community to get started with RL4CO's modular data generators.
>
> ---
>
> ### References
>
> [1r] Manchanda, S., Michel, S., Drakulic, D., & Andreoli, J. M. (2022, September). On the generalization of neural combinatorial optimization heuristics. In Joint European Conference on Machine Learning and Knowledge Discovery in Databases (pp. 426-442). Cham: Springer Nature Switzerland.

---

> > ### Comment · Reviewer_7ic3 · 2024-08-26
> > **Thanks**
> >
> > Thanks for the clarification . I maintain my score.

---

> > > ### Author Response · Authors · 2024-08-26
> > >
> > > Thank you for your valuable feedback and recognition of our paper. If you have further questions or concerns, we remain available for further clarification!

---

### Official Review · Reviewer_hVj8 · 2024-08-13
**Important topic with tech contributions not satisfactory**

**Rating:** 6
**Confidence:** 4
**Correctness:** 1. tech contributions are not satisfa…

**Review:**

(sorry for the delayed review coments)

However, the manuscript could be improved by addressing the following concerns:
1. For the presentation/expressions, this paper is a bit too descriptive (sounds bragging), which may result from the advertisement for organizing RL4CO competition.  When wrapping up it into a paper, please try to focus on the tech contributions.
2. It is very great that it includes 23 SOTA methods for over 20 CO problems. However, a more deep contribution may be reproducing the winner teams in each tasks. Considering that the CO field is developing very fast, those winning solutions would contributed more to the academia.
3). What are the technical contributions?   "modularized implementation" and "flexible configurations" are good features; "benchmarking study" should be considered as results and verification.  Then, the technical contributions are not not satisfactory.
4). The Github repo "https://github.com/ai4co/rl4co" is not very active, considering that the CO field has very active research progress.
5). For such a repo, providing several home-grown examples would help the community to grow.
6). Regarding extensibility, after the RL4CO competition, would you please clarify what new methods are added into this project?   Due to the "modularized implementation" and "flexible configurations" features, a natural testing would be some new methods are added.

**Strengths:**

1. The topic of RL4CO is very important. The RL4CO competition provides a good foundation for this project.
2. It is impressive to include 23 methods for over 20 problems.

**Additional Feedback:**

In the rebuttal, please try to address the above 2), 3), 5), and 6).

**Clarity:**

The paper is written in a descriptive way, more tech detailed and contributions are expected.

**Documentation:**

It is clear enough, as the website provides good documentations.

**Ethics:**

This project may not have ethic issues, as it deals with combinatorial optimizations that are math-oriented.

**Limitations:**

The reviewer did not notice such issues.

**Opportunities For Improvement:**

For rebuttal, the above 2), 3), 5), and 6) could help improve it.

**Relation To Prior Work:**

The reviewer believes it is clear enough.

**Summary And Contributions:**

Based on the notable RL4CO competition, the authors wrap up the work into an organized benchmark.  It claims to include 23 methods for over 20 CO problems.

---

> ### Author Rebuttal · Authors · 2024-08-16
>
> Thank you for your review. We will address your concerns point by point in the following rebuttal.
>
> > Based on the notable RL4CO competition, the authors wrap up the work into an organized benchmark [...] this paper is a bit too descriptive (sounds bragging), which may result from the advertisement for organizing RL4CO competition [...] After the RL4CO competition, would you please clarify what new methods are added into this project?
>
> We think there has been a misunderstanding. We did not organize any RL4CO competition, and this paper is not a result of a competition nor a wrap-up of results from teams in a competition. This paper is a new benchmark - with a scope on modularity, efficiency, and comprehensiveness, as mentioned by Reviewer 7ic3 - of several RL environments and methods from the RL for the CO community. We recognize, however, that it would indeed be a great idea to organize an RL for CO competition in the future!
>
> > [...] a more deep contribution may be reproducing the winner teams in each tasks. Considering that the CO field is developing very fast, those winning solutions would contributed more to the academia.
>
> As a benchmark, we include several methods from the literature for a wide range of CO problems to facilitate benchmarking new methods in the future. Moreover, the modular design of RL4CO also allows the adoption of methods originally proposed for one problem to other problems or even the combination of approaches to yield new SotA methods. For example, in the experiments in Table 2 we developed a SotA solver for the FJSSP by using architectures originally proposed for other problems. Note that due to the extensiveness of our benchmark, many results and tables are in our [supplementary material](https://openreview.net/attachment?id=Becrgm5xAq&name=supplementary_material).
>
> > What are the technical contributions? "modularized implementation" and "flexible configurations" are good features; "benchmarking study" should be considered as results and verification. Then, the technical contributions are not not satisfactory.
>
> We will cite for context the "scope" section of the [Call for NeurIPS 2024 Datasets and Benchmarks](https://neurips.cc/Conferences/2024/CallForDatasetsBenchmarks#:~:text=SCOPE.%C2%A0This%20track%20welcomes,yielding%20important%20new%20insight.):
>
> "This track welcomes all work on data-centric machine learning research (DMLR) and *open-source libraries and tools* that enable or accelerate ML research, covering ML datasets and *benchmarks* as well as *algorithms, tools, methods, and analyses* for working with ML data. This includes but is not limited to [...] *data generators and reinforcement learning environments* [...] and *benchmarking tools*."
>
> Our technical contribution is the introduction of the new RL4CO library, a unified and extensive benchmark in Reinforcement Learning for Combinatorial Optimization, filling a gap in the CO field that lacks a unified and efficient benchmark for easy development and standardized comparison of methods. Note that RL4CO is not merely a collection of previous code, rather a novel in-depth unified framework for ultimately helping researchers and practitioners in the increasingly recognized field of CO.
>
> > The Github repo "https://github.com/ai4co/rl4co" is not very active, considering that the CO field has very active research progress.
>
> We politely disagree with the reviewer’s opinion that the RL4CO repository is not very active. Looking at the number of stars and contributions on Github - more than 350 stars, 20+ contributors, and more than 1400 commits - we can confidently say our repository is more active than many others. For example, when compared to the 10 Oral papers in NeurIPS Datasets and Benchmarks Track 2023, RL4CO has more stars than 7 out 10 of them.
>
> > Providing several home-grown examples would help the community to grow
>
> We agree with you that providing examples is really important for the community to grow! That is why we have provided an extensive documentation with a library of several examples [here](https://rl4.co/examples/1-quickstart/) which can be easily run on consumer hardware.
>
> > Regarding extensibility, after the RL4CO competition, would you please clarify what new methods are added into this project? Due to the "modularized implementation" and "flexible configurations" features, a natural testing would be some new methods are added.
>
> Regarding the non-existing (at least not yet) RL4CO competition, please refer to the above response. About new methods added into the project: we would like to note that, since this is a benchmark track, we are not aware it is required to have new methods. Otherwise, it would be more appropriate for us to submit our work at the main NeurIPS Research track.
>
> Nonetheless, thanks to RL4CO's modularity and extensibility, we also tested new methods. Here are some examples, to mention the ones in the main text. In Section 5.1, we show that by changing and tuning components of a policy, thanks to our framework's modularity, we could get new state-of-the-art results in the FJSSP. In Section 5.2, we benchmarked several methods for the first time due to their original implementation not being compatible with certain environments. In Section 5.3, we show that by integrating constructive and improvement methods for the first time, we can achieve better performance than a constructive or improvement method alone.

---

> > ### Comment · Reviewer_hVj8 · 2024-08-25
> > **Could you please point out difference between this work with the ML4CO competition?**
> >
> > Dear authors,
> >
> > In the above comment, sorry that I was confused with the ML4CO competition: https://www.ecole.ai/2021/ml4co-competition/
> >
> > A followup question would be: Could you please point out difference between this work with the ML4CO competition?
> >
> > I believe the reviewers and AC would be happy to know the differences.

---

> > > ### Author Response · Authors · 2024-08-26
> > >
> > > Thanks for your feedback - we will gladly take this opportunity to highlight the differences between the ML4CO competition and our proposed RL4CO benchmark in the following response!
> > >
> > > Firstly, a key difference between ML4CO [1r] and RL4CO is that ML4CO is a competition, and RL4CO is a benchmark. A competition such as the ML4CO one is a time-bound event where participants aim to achieve the best performance on a specific problem within a set timeframe, often focusing on optimizing existing methods. A benchmark like RL4CO serves as a standardized and ongoing framework for evaluating and comparing different approaches to solve a broader class of problems, with the goal of advancing the field through continuous improvement.
> > >
> > > The ML4CO competition is based on the Ecole library [2r] which, as per the description ([link](https://www.ecole.ai/#:~:text=Rather%20than%20trying%20to%20predict%20solutions%20to%20combinatorial%20optimization%20problems%20directly%2C%20the%20philosophy%20behind%20Ecole%20is%20to%20work%20in%20cooperation%20with%20the%20state%2Dof%2Dthe%2Dart%20Mixed%20Integer%20Linear%20Programming%20solver%20SCIP%20that%20acts%20as%20a%20controllable%20algorithm.)), has the following focus:
> > >
> > > > Rather than trying to predict solutions to combinatorial optimization problems directly, the philosophy behind Ecole is to work in cooperation with the state-of-the-art Mixed Integer Linear Programming solver SCIP that acts as a controllable algorithm.
> > >
> > > In contrast, RL4CO's direction is orthogonal, i.e., we do not need a Mixed Integer Linear Programming (MILP) solver to obtain solutions. This is especially important for problems that cannot be easily formulated or solved with MILP.
> > >
> > > Moreover, as per the ML4CO competition description ([link](https://www.ecole.ai/2021/ml4co-competition/#:~:text=The%20competition%27s%20main%20scientific%20question%20is%20the%20following%3A%20is%20machine%20learning%20a%20viable%20option%20for%20improving%20traditional%20combinatorial%20optimization%20solvers%20on%20specific%20problem%20distributions%2C%20when%20historical%20data%20is%20available%20%3F)):
> > >
> > > > The competition's main scientific question is the following: is machine learning a viable option for improving traditional combinatorial optimization solvers on specific problem distributions, when historical data is available?
> > >
> > > This is fundamentally different from the RL4CO benchmark's goal. In RL4CO, our goal is to learn how to solve general CO problems without necessarily leveraging prior (possibly expensive or impossible to collect) labeled datasets. In contrast, the ML4CO competition leverages historical data to help a classical, pre-existing solver in terms of finding good solutions faster by predicting better heuristic rules, such as branching or the best hyperparameters for a specific distribution.
> > >
> > > While integrating machine learning with traditional MILP solvers offers significant benefits, including theoretical guarantees on solution quality (such as primal/dual bounds), it can present challenges in scaling efficiently for certain real-time, large-scale applications like vehicle routing problems. In these cases, domain-expertise-based heuristics have proven highly effective, which motivated our development of RL4CO.
> > >
> > > In summary: the ML4CO competition differs with respect to our RL4CO benchmark in several aspects, including scope, reliance on traditional solvers, and usage of labeled datasets. In this sense, we see the ML4CO competition not in "competition" with the RL4CO benchmark but rather as an orthogonal direction.
> > >
> > > ---
> > >
> > > We hope our response addresses your question, and we will add a discussion section reporting the above in the camera-ready version. If you have further questions or concerns, we remain available for further clarification. We appreciate your continued support.
> > >
> > > ---
> > >
> > >
> > > ### References
> > >
> > > [1r] Gasse, M., Bowly, S., Cappart, Q., Charfreitag, J., Charlin, L., Chételat, D., ... & Kun, M. (2022, July). The machine learning for combinatorial optimization competition (ml4co): Results and insights. In NeurIPS 2021 competitions and demonstrations track (pp. 220-231). PMLR.
> > >
> > > [2r] Prouvost, A., Dumouchelle, J., Gasse, M., Chételat, D., & Lodi, A. (2021). Ecole: A library for learning inside milp solvers. arXiv preprint arXiv:2104.02828.

---

> > > > ### Comment · Reviewer_hVj8 · 2024-08-31
> > > > **Thanks for your clarification.**
> > > >
> > > > Appreciate the added info and clarification.
> > > > I have read all response and other reviewers' comments.  This work has good merits, while the reviewers also raise important concerns about whether it justifies an acceptance. I will actively join the voting process.

---

> > > > > ### Author Response · Authors · 2024-08-31
> > > > >
> > > > > Thank you for acknowledging the value of our work. We believe we have addressed your previous concerns thoroughly. However, we are wondering about the "important concerns" to make you hesitate to vote for a clear acceptance. It would be beneficial if you could explicitly outline the further concerns and we are available to make further clarification until the end of the discussion period. We kindly remind that the discussion period is ending. We will address them as soon as possible.

---

> > > > > > ### Comment · Reviewer_hVj8 · 2024-08-31
> > > > > >
> > > > > > A few questions that need more info from the authors:
> > > > > >
> > > > > > 1. What are your "home-grown examples"?  please note that it is not just "examples"; not just "documentations"
> > > > > >
> > > > > > 2. For the above response: "Nonetheless, thanks to RL4CO's modularity and extensibility, we also tested new methods. Here are some examples, to mention the ones in the main text. In Section 5.1, we show that by changing and tuning components of a policy, thanks to our framework's modularity, we could get new state-of-the-art results in the FJSSP. In Section 5.2, we benchmarked several methods for the first time due to their original implementation not being compatible with certain environments. In Section 5.3, we show that by integrating constructive and improvement methods for the first time, we can achieve better performance than a constructive or improvement method alone."
> > > > > >
> > > > > >      This is descriptive.  Please give one example of implementing a particular existing paper using your framework.  How exactly? You can point out aspects like: modeling, components, policy networks, training, testing, etc.   Show the tech/engineering parts please.
> > > > > >
> > > > > > 3. One feedback that leads to more concerns of this work is:
> > > > > >     After reading all the responses (to all reviewers),  the authors did not expand the tech parts or engineering parts.  For developing such a library, there would be some milestones that the authors know so well. The current responses took a very descriptive approach.  Please explain more here.

---

> > ### Author Response · Authors · 2024-09-01
> >
> > > 1. What are your "home-grown examples"? please note that it is not just "examples"; not just "documentations"
> >
> > We have already replied above to this question. Also: if the examples are not "code", i.e., scripts and Jupyter Notebooks, nor our documentation, what do you mean by home-grown examples? **Please be specific**.
> >
> > > 2. This is descriptive. Please give one example of implementing a particular existing paper using your framework. How exactly? You can point out aspects like: modeling, components, policy networks, training, testing, etc. Show the tech/engineering parts please.
> >
> >
> > All right then, let us take a new example compared to the ones we already gave.
> >
> > Let us take the Heterogeneous Attention Model as an example [1r]. This model implements a new attention mechanism for pickup and delivery problems and additionally proposes a new Initial Node Embedding and a new Context Embedding to encode raw features into the latent space.
> >
> > *Modeling: policy and components*. The policy is based on the Attention Model as seen [here](https://github.com/ai4co/rl4co/blob/main/rl4co/models/zoo/ham/policy.py), so we will have the same decoder, which is very convenient. Then, we use the proposed encoder ([here](https://github.com/ai4co/rl4co/blob/main/rl4co/models/zoo/ham/attention.py)). We then just need to implement the Init ([here](https://github.com/ai4co/rl4co/blob/193d7692155ef00b6a566a870a396c4cc7a93ec9/rl4co/models/nn/env_embeddings/init.py#L293)) and Context Embedding ([here](https://github.com/ai4co/rl4co/blob/193d7692155ef00b6a566a870a396c4cc7a93ec9/rl4co/models/nn/env_embeddings/context.py#L238)) (see Figure 15 in Appendix as an illustration) which is simple since it is already modularized. The environment and data generator if needed can be added as seen [here](https://github.com/ai4co/rl4co/tree/main/rl4co/envs/routing/pdp) and we have already described this process in documentation.
> >
> > *Training and testing*: then, we can train the model by simply creating a new yaml configuration: environment ([here](https://github.com/ai4co/rl4co/tree/main/configs/env)) and model based on the implemented REINFORCE in this case ([here](https://github.com/ai4co/rl4co/tree/main/configs/model)). If the user does not want to train the model with Hydra, then it can be done by a simple Python script as shown [here](https://github.com/ai4co/rl4co/blob/main/examples/2b-train-simple.py). Testing can either be done automatically by loading the dataset names (conveniently reported directly to the logger such as Wandb) or by loading the policy in the evaluation script [here](https://github.com/ai4co/rl4co/tree/main/rl4co/tasks).
> >
> > It is quite easy to adapt to new problems as well as different neural networks without needing to reimplement the wheel.
> >
> >
> > > 3. After reading all the responses (to all reviewers), the authors did not expand the tech parts or engineering parts. For developing such a library, there would be some milestones that the authors know so well. The current responses took a very descriptive approach. Please explain more here.
> >
> > What "tech parts" or "engineering parts" should we develop? What "milestones" are you referring to? We are not aware of comments asking to re-write, re-develop, re-engineer any part of our library. Again, **please be specific**.
> >
> > If anything, mentioning some *criticism* (!) points by Reviewer Vcfw: "*There are detailed examples with Python commands for running the benchmarks [...] The paper gets into too much detail, such as what commands to use with which Python scripts to produce results [...] but the coverage is not typical for this sort of paper. It does seem to go into a lot of detail that would best be included in a tutorial.*" and also Reviewer su7y: "*After I read this paper, it looks like the summary or report of engineering [...]  which should be put to the readme of github or docs [...] I think it is not like a paper but a summary of the lib*". It shows that, contrary to your comment, we included enough details (even too detailed in some reviewers' opinions) in terms of engineering and tech.
> >
> > We are puzzled by your comment asking to expand on something we were asked to shrink down.
> >
> > As we have invested significant effort and diligence in preparing this paper, we hope this rebuttal will be satisfactory.
> >
> > Thank you in advance for your cooperation.
> >
> > ---
> >
> > #### References
> >
> > [1r] Li, J., Xin, L., Cao, Z., Lim, A., Song, W. and Zhang, J., 2021. Heterogeneous attentions for solving pickup and delivery problem via deep reinforcement learning. IEEE Transactions on Intelligent Transportation Systems, 23(3), pp.2306-2315.

---

### Author Rebuttal · Authors · 2024-08-16

We sincerely thank the reviewers for their feedback and for recognizing our work as being extensive in the number of CO problems (Reviewers hVj8, 7ic3, and Vcfw), useful (Reviewer Vcfw), clearly presented (Reviewers su7y and 7ic3), well-documented (Reviewer hVj8 and Vcfw) and with valuable contributions to the field (Reviewer 7ic3).

Since there appear to be some misunderstandings, we would like to address the key points raised to ensure clarity and reinforce the contributions of our paper.

### Clarification on the Scope and Contribution
Our paper introduces RL4CO, a unified and comprehensive benchmark specifically designed for Reinforcement Learning (RL) in Combinatorial Optimization (CO).

The benchmark is not derived from any competition results; rather, it is a novel and independent contribution aimed at filling a critical gap in the RL for CO field. RL4CO provides a modular, efficient, and extensible framework that enables standardized comparisons and facilitates the development of new methods.

We would also like to emphasize that RL4CO is more than just a collection of existing methods: it introduces a new, in-depth framework that not only consolidates existing approaches but also demonstrates the potential to achieve new state-of-the-art results by leveraging its modular design.

To conclude, our work aligns with the objectives of the NeurIPS Datasets and Benchmarks track, which welcomes open-source libraries and tools that accelerate machine learning research.

### About Missing Problems and Methods Descriptions
The necessary descriptions are explicitly detailed in our paper. Section 4.5 of our manuscript clearly lists the CO environments and baseline methods. The [supplementary material](https://openreview.net/attachment?id=Becrgm5xAq&name=supplementary_material) provides extensive details, including Appendix B for environment descriptions and Appendix C for baseline descriptions, spanning almost 20 pages. Extensive details are available not only throughout the text but also through [our online documentation](https://rl4.co/). We finally remark that this paper does not introduce new "datasets" but is rather a *benchmark* of RL environments and baselines for combinatorial optimization.


### Activity and Impact of the RL4CO Repository
The activity level and impact of the RL4CO repository are substantial, as evidenced by its strong community engagement, contributions, and ongoing developments. With over 350 stars, 20+ contributors, and more than 1400 commits on GitHub, RL4CO is more active than many other benchmarks in the community; for instance, RL4CO has more stars than 7 out  of the 10 Oral papers accepted at the NeurIPS Datasets and Benchmarks Track 2023. This highlights the relevance of our work, as well as the interest it has already generated among researchers and practitioners.

### Addressing Concerns on Implementation Details and Clarity
We included detailed implementation examples and Python commands in the paper to ensure that the benchmark is accessible and easy to use. This level of detail is essential for practical adoption and aligns with the expectations for papers introducing new libraries and tools. We have also taken note of the feedback regarding figure clarity and typographical errors and will ensure that these are addressed in the final version.

### Extensibility and Future Directions
We are committed to further expanding the benchmark to support a broader range of methods and problem types, ensuring its continued relevance and utility to the research community. While our initial focus was on RL approaches, the modular nature of RL4CO allows for future extensions to other machine learning paradigms. Our long-term plan is to become the to-go RL for CO benchmark library, and it is our hope that our work will ultimately benefit the NCO field with new ideas and collaborations.

We believe RL4CO represents a significant and timely contribution to the field of combinatorial optimization. It provides a robust, well-documented, and highly active benchmark that will support future research and development. We sincerely hope that the clarifications provided in our rebuttal could address any potential concerns and we would like to clarify that RL4CO is well-positioned to make a meaningful impact on the NCO community.

We politely ask reviewers to take our answers into account when (re)evaluating our paper. We greatly appreciate your consideration and look forward to the opportunity to contribute further to the field.

---

> ### Comment · Reviewer_hVj8 · 2024-08-31
> **Authors provided descriptive responses, and the salience of development in the rebuttal phase is unsatisfactory for a library.**
>
> I would like to raise a bit serious concern here, since I noticed some contradictory observations that are different from open-source projects (I created and lead some popular ones in the past).
>
> Over the rebuttal phase (in the past month), it seems that the authors did not provide more evidences of engineering efforts in improving, polishing or maintaining the proposed RL4CO library.
>
> Several aspects echoing this coment:
> 1). The responses are descriptive, while such a library should be "engineering-heavy". Why not provide more evidences for some questions? say home-grown example, modularity and extensibility, reproduction, or fair comparison?
> 2). I also checked this repo: https://github.com/ai4co/rl4co
>      There were few changes here: https://github.com/ai4co/rl4co/tree/main/rl4co/envs/graph
>      There are 4 issues (one open and 3 closed), and one pull request about python version (https://github.com/ai4co/rl4co/pull/204)
>
> Therefore, I would still say that the community of RL4CO is not active (while CO problems are very interesting); and it looks a bit strange to me if the authors claim to actively maintain it.
>
> I would like to vote a "clear rejection" at this moment, and I believe this RL4CO library would enjoy more impacts given another chance of full reviewing process at a later AI conference.      I am generally happy with authors' responses and clarification.  Such a rejection suggestion came from the above concerns.

---

> > ### Author Response · Authors · 2024-09-01
> > **Dear Reviewer hVj8, we totally disagree with this. Please reconsider your judgement at once. (1/2)**
> >
> > First of all, let us ask a question. We have already replied to the points above in the first rebuttal answer. Then you asked us:
> >
> > > Sorry that I was confused with the ML4CO competition [...] Could you please point out difference between this work with the ML4CO competition?
> >
> > After we *explained the difference between a benchmark and a competition*, you said "This work has good merits, while the reviewers also raise important concerns about whether it justifies an acceptance". Upon us asking you what the "important concerns are", **you answered with requests less than 16 hours before the rebuttal deadline. Why did you not do this before?**
> >
> > ---
> >
> >
> > This said. Let's go through the points you raised once more.
> >
> >
> > > The responses are descriptive, while such a library should be "engineering-heavy". Why not provide more evidences for some questions? say home-grown example, modularity and extensibility, reproduction, or fair comparison?
> >
> > Responses to a rebuttal are meant to be descriptive. **Your opinion also contrasts with criticism (!) by Reviewers Vcfw and su7y**: "*There are detailed examples with Python commands for running the benchmarks [...] The paper gets into too much detail, such as what commands to use with which Python scripts to produce results [...] but the coverage is not typical for this sort of paper. It does seem to go into a lot of detail that would best be included in a tutorial.*"  "*After I read this paper, it looks like the summary or report of engineering [...]  which should be put to the readme of github or docs [...] I think it is not like a paper but a summary of the lib*".  It seems that, contrary to what you are suggesting, our manuscript includes rich (even too rich in some reviewers' opinions) engineering and technical details.
> >
> > Moreover, your points are not specific at all, making it hard to engage in constructive criticism. Note we have already extensively replied in our 70-page long manuscript (main text + supplementary), our library, online documentation, and rebuttals. Note that a longer version of this answer is available on your thread.
> >
> >
> > > Over the rebuttal phase (in the past month), it seems that the authors did not provide more evidences of engineering efforts in improving, polishing or maintaining RL4CO. [...] There were few changes here: https://github.com/ai4co/rl4co/tree/main/rl4co/envs/graph There are 4 issues (one open and 3 closed), and one pull request about python version (https://github.com/ai4co/rl4co/pull/204) [...]  > Therefore, I would still say that the community of RL4CO is not active.
> >
> >
> > Are you evaluating the activeness of our repository solely based on the number of pull requests and issues in the last three weeks when we were focusing on the rebuttal and discussion?
> >
> > First of all, if anything, one should also look at the number of commits, which is more indicative of our work. Secondly, we are not aware of guidelines in the NeurIPS Datasets and Benchmarks Track saying that a work should be judged based on the metrics you mentioned. Finally: if you want to judge RL4CO based on Github statistics, [here](https://docs.google.com/spreadsheets/d/16nX-wmSp6I5R4qrVeMHAIR2gcn_HqK1lYDg_JPiu-v8/edit?usp=sharing) is a spreadsheet of the past year's Oral papers. Note that we do not claim we are necessarily better than any specific paper/library, neither say RL4CO is perfect or does not have ways to improve and further develop. But the numbers are clear and the fact is: **RL4CO has more stars than 7/10 ORAL papers from last year and more active than 8/10. According to your point, then most of the ORAL papers in NeurIPS 2023 Datasets and Benchmarks Track should have been rejected as well.**

---

> > > ### Author Response · Authors · 2024-09-01
> > > **Dear Reviewer hVj8, we totally disagree with this. Please reconsider your judgement at once. (2/2)**
> > >
> > > > [...] it looks a bit strange to me if the authors claim to continue maintaining it.
> > >
> > > **Sorry, this is not only wrong, but we find it even offensive**. You are groundlessly claiming (at least implying) that we want to just dump the repository after we get accepted and move on to another topic. We find this unsubstantiated claim outrageous and unfair.
> > >
> > > RL4CO was initially submitted in 2023. We opened the rebuttal discussion [here](https://openreview.net/forum?id=LJhfKeqZdu) due to the fact that after a successful rebuttal with 4 out of 5 reviewers initially giving acceptance scores, **two reviewers changed the score and modified the content of their review just before the rebuttal was ending - which is similar to what is, unfortunately, happening this year as well, by the way.**
> > >
> > > *But we did not give up.*
> > >
> > >  We could have instead moved on and submitted many incremental salami-sliced works to several conferences, but we did not. On the contrary, we addressed all the constructive criticism from the previous rebuttal. We have collaborated with several researchers who helped us and contributed to this paper. We created the AI4CO community. We expanded our scope from the initial three environments and five baselines to over 20 environments and over 20 baselines for this submission.
> > >
> > > **We worked hard maintaining, greatly improving RL4CO continuously, answering to issues on Github and on Slack and helping out fellow researchers for more than one year. Is this not enough to prove we will maintain it afterward?**
> > >
> > >
> > > Moreover. We have already answered your feedback above. If, quoting verbatim what you *just* said:
> > >
> > > >  I am generally happy with authors' responses and clarification.
> > >
> > > **Then why would you like to vote a "clear rejection" (3) after you initially judged our work as a "weak accept" (6)?**
> > >
> > > To quote your words, we "*noticed some contradictory observations*" here in your comments, which makes us, and hopefully others involved in the review process, question the validity of your criticism.
> > >
> > >
> > > Note that we will opt-in to open our rebuttal if needed.
> > >
> > > ---
> > >
> > > **Please reconsider your judgment at once.**

---

> > ### Comment · Reviewer_hVj8 · 2024-09-01
> > **About the comment "Home-grown example"**
> >
> > Dear authors,
> >     It seems that there were misunderstandings. Let us try to clarify a bit and see how much we can achieve.
> >
> > 1. The meaning of "Home-grown example" is this: "native to or characteristic of a particular area".
> >     This reviewer is not asking for "detailed examples", as you did provide many examples and a lot of details in the appendix. Since there are already a lot in your library and appendix, thus, this reviewer tried to help by asking whether it is possible to pick one or two as home-grown examples for your RL4CO library.
> >     Imagine this: RL4CO has "20 CO problems", "23 methods", "77 pages of appendix", etc., the readers or users will be overwhelmed with info and get panic.   Then, why not help users/readers pick one or two as your "home-grown examples".  This is a kind suggestion and there is no contradiction with other reviewers' comments, actually, it aligned very well with other reviewers' comments.  I believe the author team would accept this suggestion;   even if it is not directly executable at this moment, it is good to bring this suggestion to the author team.
> >     This comment has been provided in the beginning.
> >
> > 2. Please rethink about your above rebuttal:
> >
> > "Responses to a rebuttal are meant to be descriptive. Your opinion also contrasts with criticism (!) by Reviewers Vcfw and su7y: "There are detailed examples with Python commands for running the benchmarks [...] The paper gets into too much detail, such as what commands to use with which Python scripts to produce results [...] but the coverage is not typical for this sort of paper. It does seem to go into a lot of detail that would best be included in a tutorial." "After I read this paper, it looks like the summary or report of engineering [...] which should be put to the readme of github or docs [...] I think it is not like a paper but a summary of the lib". It seems that, contrary to what you are suggesting, our manuscript includes rich (even too rich in some reviewers' opinions) engineering and technical details.
> >
> > Moreover, your points are not specific at all, making it hard to engage in constructive criticism. Note we have already extensively replied in our 70-page long manuscript (main text + supplementary), our library, online documentation, and rebuttals. Note that a longer version of this answer is available on your thread."
> >
> > I believe that other reviewers and the area chair (and possibly the authors) would now agree with this: the reviewers (Reviewers Vcfw, su7y, and hVj8) made the same comment, while this reviewer (Reviewers hVj8) even tried to help the authors to find a good way: "pick one or two as your "home-grown" examples. A sad news is that there was some misunderstandings in the interactions.

---

> > > ### Comment · Reviewer_hVj8 · 2024-09-01
> > > **About "the salience of development in the rebuttal phase is unsatisfactory for a library"**
> > >
> > > Regarding the authors' questions like "Are you evaluating the activeness of our repository solely based on the number of pull requests and issues in the last three weeks when we were focusing on the rebuttal and discussion?"  "we are not aware of guidelines in the NeurIPS Datasets and Benchmarks Track saying that a work should be judged based on the metrics you mentioned."
> > >
> > > On this page of NeurIPS 2024 Datasets and Benchmarks Track: https://neurips.cc/Conferences/2024/CallForDatasetsBenchmarks
> > >
> > > On the above page, there are a few guidelines that require such reviewing metrics:
> > > 1. "how it will be made available and maintained".
> > > 2. "is guaranteed to be maintained for many years".
> > > 3. "how it will be made available and maintained."
> > > Therefore, there is no problem with checking the activity of a library.  In the rebuttal phase, 4 reviewers were providing questions, asking for clarification, and giving suggestions, it is not wrong (kinda natural) to see whether the RL4CO's authors and developers were actively engaging in the RL4CO library.
> > >
> > > As an author (of my own projects), when I suggestions and comments, I would engage in discussions and communications with reviewers, in the meantime being actively improving the project.
> > >
> > > The arguments put by the authors make this reviewer feel unpleasant. In particular, I was trying to give constructive comment. After reading the responses (including to other reviewers), I got confused why there is nothing need to change in terms of adjusting codes, adding additional experimental results, documentation (the page: https://rl4.co/), (or based on those 20 CO problem, pick one or two to illustrate the claimed good features of "unified and extensive, standardized, efficient, in-depth, modularized and flexible, decoupling science from heavy engineering").
> > >
> > > As a reviewer, I got very puzzled as "authors provided descriptive responses", but not showing some more evidence.
> > > 1). if given a CO task (that aligned well with existing examples, there are 20 of them), how to leverage ""unified and extensive, standardized, efficient" RL4CO library?
> > > 2). If given a new CO task (not directly available, but sill align well), how to leverage "modularized and flexible, decoupling science from heavy engineering" RL4CO library?
> > >
> > > The authors are the experts here, would you please educate your external reviewers, since a reviewer is obligated to learn this library during the rebuttal phase.

---

> > > ### Author Response · Authors · 2024-09-01
> > >
> > > We appreciate the reviewer’s follow-up clarification. Regarding the request for "Home-grown examples," we would like to emphasize that several examples were already provided in the original main paper, specifically in Section 5, "Benchmarking Study." To summarize:
> > >
> > > Section 5.1: We demonstrate the flexibility and modularity of our RL4CO framework by modifying policy components in the FJSSP CO problem. This shows that by exploring different modules implemented in RL4CO, one can easily achieve superior performance compared to the state-of-the-art reported in the original paper.
> > >
> > > Section 5.2: We discuss the impact of selecting different baselines during on-policy RL training for various VRP problems. We explore different decoding schemes, generalization performance, and performance on large-scale instances—all of which serve as "Home-grown examples" that verify the utility of our library for NCO research purposes.
> > >
> > > Section 5.3: We introduce a new hybrid approach by combining construction and improvement models to enhance inference performance. This method is easily implemented within RL4CO due to its modular architecture, which allows for the seamless integration of diverse NCO approaches in the future to achieve the effects of "1+1>2".
> > >
> > > Please note that most of the above "Home-grown examples" provided in our original main paper and the appendix go beyond the current literature on NCO. Our library enables new results and insights in the field, which have not been previously reported or discussed. We believe this provides ample evidence of the unique benefits of RL4CO.
> > >
> > > Regarding the lengthy appendix, we acknowledge your suggestion and will strive to reorganize and refine the presentation as per your guidance. If you believe certain examples from the appendix should be moved to the main paper, please let us know, and we would appreciate your constructive feedback. However, we would like to clarify that most "serious" concerns raised by the reviewers are not as crucial as they may seem and are even not a valid issue to our view. We have already highlighted several "Home-grown examples" (summarized above) and key differences compared to "Jumanji" (see section 2 and table 12) in the original main paper. We believe that the experiments and examples provided in our main paper and appendix demonstrate capabilities that cannot be achieved using Jumanji.

---

> > ### Comment · Reviewer_hVj8 · 2024-09-01
> > **Your work RL4CO is very much appreciated at my end.**
> >
> > Dear authors,
> >      Please note that RL4CO is very much appreciated at my end. And the goal of the reviewing process is to provide constructive comments to help it before formally publishing it.
> >
> > I understand that the authors may feel that "Authors provided descriptive responses, and the salience of development in the rebuttal phase is unsatisfactory for a library" is unfair to your over two years' efforts. However, the reviewer's responsibility is to ask questions and request clarification. After this reviewing period, there is voting; and our area chair will make final decision.
> >
> > Your response "You are groundlessly claiming (at least implying) that we want to just dump the repository after we get accepted and move on to another topic. We find this unsubstantiated claim outrageous and unfair" can be understood here. However, as a reviewer, I could point out my observations and raise concerns;  and the authors could write to our area chair about your feelings in another channel; and this is a standard reviewing protocol.
> >
> > There is not need to feel disappointed as you already got a strong work.  The question would be: whether it is ready for publication at this round?  As a reviewer, there is no problem of raising concerns and hoping a bit more polishing. Your clarifications would help address such concerns.
> >
> > In the operation research field and also AI field, a lot NCO algorithms are compared with solvers, say running Gurobi, SCIP for one hour and reporting the gap. Now, I am scared to ask this question: Would it be fair to request such comparison in RL4CO?  Is it possible to get the authors sharing and viewpoints?

---

> > ### Comment · Reviewer_hVj8 · 2024-09-01
> > **70 pages of appendix is lengthy, can this reviewer request clarification of key points/features/claims?**
> >
> > Thanks for your response.
> >
> > "We can reuse most of the modular components while engineering the necessary differences. For instance, if we want to model a PDP with additional constraints like last-in-first-out, we can reuse the constructive module, such as Heter-AM, and modify the feasibility masking and problem-specific features by customizing our provided modules. We have a detailed Jupyter Notebook available here (https://github.com/ai4co/rl4co/blob/main/examples/3-creating-new-env-model.ipynb) that demonstrates how to extend RL4CO to solve new problems from scratch. Moreover, we highlight that many examples in the appendix demonstrate the ability to extend our RL4CO library to various CO problem variants."
> >
> > I would say that this above response shows more willingness to explain and clarify things about RL4CO;  and other responses were not due to overwhelming info/docs.  Before this, I read all responses (to all reviewers), the authors throw docs (or links to Jupyter notebooks), which was  considered "only descriptive responses".  Still, this reviewer expected/expects the authors would be more willing to take an educational role to all reviewers; do not scare reviewers/readers/users by claims like "23 methods, 20 CO problems; 77 pages of appendix".
> >
> > It would be very good to discuss and understand and communicate how this library is "extensive" (in the title), "efficient" (in abstract), "modularized and flexible and extensible" in Section 5.1, 5.2, 5.3 (which is short and lacks explanations due to space limit), with the authors' kind help.  Please do not put a lot info, which would be overwhelming to future readers/users too.

---

> > > ### Author Response · Authors · 2024-09-01
> > > **Hope we have addressed any misunderstanding in the review process**
> > >
> > > We appreciate your feedback and your recognition of our work. We sincerely apologize if our previous responses seemed aggressive; as we feel that our work was potentially treated unfairly.
> > >
> > > We fully acknowledge and appreciate your efforts to provide any new feedback and ask questions. Regarding the new questions and concerns you raised, we believe that these concerns/questions should ideally have been pointed out at the beginning of the review and rebuttal process. This would have allowed us sufficient time to respond and address them thoroughly. Raising these issues at the end of the rebuttal period is not conducive to a productive and constructive review process in our view. Otherwise, it appears to us that the reviewer may be more focused on finding reasons to reject our paper rather than offering constructive suggestions. If we have misunderstood your intentions, we sincerely apologize. We are sorry!!
> > >
> > > For your question, we would like to clarify that our work is not intended to introduce a new benchmark method or approach for comparing RL solvers with OR solvers. Rather, our focus is on providing a comprehensive library and benchmarks of existing RL-hased neural solvers for CO problems. While comparing with OR methods is not our primary contribution and scope, and no single work can cover all benchmarking studies in the field, we do agree that it helps to understand the gaps of neural solvers and we kindly refer the reviewer to the existing results in our original submission. Specifically, in Table 10 (page 43) and Table 14 (page 48) of the appendix, we have compared our implemented algorithms with several OR solvers, including Gurobi, Concorde, HGS, LKH, and OR tools.
> > >
> > > Also, we want to clarify that our intention is not to overwhelm or intimidate reviewers, readers, or users with statements like "23 methods, 20 CO problems; 77 pages of appendix." Instead, we aim to showcase the robustness of our RL4CO framework across a wide range of experiments. We appreciate your feedback and will work to organize the results more effectively in the revision, following your suggestions. We apologize for any confusion caused.
> > >
> > > Lastly, we thank you for the further reply to clarify any misunderstanding here. We hope you can make an independent and objective assessment of the overall contribution of our RL4CO work. Thank you for your cooperation and support!

---

> > ### Comment · Reviewer_hVj8 · 2024-09-01
> > **A reviewer votes; our area chair will make final decision. Please help clarify.**
> >
> > I cannot reject your work directly; I could try to ask good questions.
> > Your patience and kind explanation will earn you good reputation, also in your open-source community.
> >
> > This reviewer ask restricted questions (not very hashing) for clarifications. If you look back, you will find the reviewer acknowledged error in "confused with ML4CO", and constructive suggested the authors to further clarify the differences.  Such an interaction was intentionally trying to help you earn points from all reviewers and our area chair.  At that moment, this reviewer would like to see RL4CO becoming a flag project in the AI4CO community.
> >
> > However, other concerns like the feature claims " how this library is "extensive" (in the title), "efficient" (in abstract), "modularized and flexible and extensible" in Section 5.1, 5.2, 5.3 (which is short and lacks explanations due to space limit)" still do not get good explanations yet.  Those are remaining concerns, possibly also by another Reviewer su7y (who lowered one points); I understood this also caused a bit tension.
> >
> > However, please remain calm and patient. A reviewing process could challenge the authors about such potential issues, your kind responses always help you earn the voting.
> >
> > I did not try to reject it at all (a good library with 90 pages of docs and websites, 2 years' efforts).  I was obligated to ask questions, even hashing ones, so that our area chair would make a good decision.  A reviewer is not supposed to play a "nice" face; could raise critical questions and check authors' responses.
> >
> >
> >
> > To our area chair:
> >
> > Instead of "clear rejection", which the authors do not agree with; this reviewer would like to stick to the original score of "6 weak accept", the main concerns like the feature claims " how this library is "extensive" (in the title), "efficient" (in abstract), "modularized and flexible and extensible" in Section 5.1, 5.2, 5.3 (which is short and lacks explanations due to space limit)" still do not get good explanations yet.   If it was not considered offensive, this reviewer still believe RL4CO would be better if it goes through another whole round of reviewing, in a later AI conference.

---

> ### Author Response · Authors · 2024-09-01
>
> Regarding the library’s activity, we believe we have been consistently active since last year and have provided statistics to compare with other benchmarks. NCO is an emerging and important field that does not yet have a large community like NLP or CV (which have thousands of researchers working on them). Despite this, we have demonstrated our significant contributions. We fully acknowledge the importance of maintaining and actively engaging with the RL4CO library. We are committed to ensuring the library remains active and well-supported. We also believe that for any open-source GitHub repo, it will have a number of active "issues" and "pull requests" awaiting development, and people may focus on how many of those issues are resolved rather than how many remain open in a particular month.
>
>
> To address the two questions you raised:
> 1. Our "Home-grown examples" provided in the main paper and appendix already address these concerns (see the above reply).
> 2. We can reuse most of the modular components while engineering the necessary differences. For instance, if we want to model a PDP with additional constraints like last-in-first-out, we can reuse the constructive module, such as Heter-AM, and modify the feasibility masking and problem-specific features by customizing our provided modules. We have a detailed Jupyter Notebook available here (https://github.com/ai4co/rl4co/blob/main/examples/3-creating-new-env-model.ipynb) that demonstrates how to extend RL4CO to solve new problems from scratch. Moreover, we highlight that many examples in the appendix demonstrate the ability to extend our RL4CO library to various CO problem variants.
>
> On a separate note, we appreciate the reviewers' efforts in the review process, though we regret any unpleasantness that has arisen. We feel that many of the review comments were inaccurate, overly general, and lacked objectivity, making them difficult to address effectively. More importantly, the main "concerns" raised often seem invalid from our perspective, as they pertain to points that we have already addressed and discussed in the main paper, appendix, documentation, and rebuttal process. Additionally, some of the behavior appeared unprofessional and even seemed biased toward rejecting our paper. We have already communicated these concerns to the Area Chair.
>
> Thank you for your suggestions, and we hope our clarification is helpful.

---

> ### Author Response · Authors · 2024-09-01
> **Thank you for your suggestions**
>
> Thank you for your suggestions and reply; we truly appreciate your feedback. Due to the page limitations, we hope the reviewer and area chair understand the challenges in detailing every design and engineering aspect within the main paper, particularly for an extensive benchmark library like ours. We will try our best to refine the paper in the final version.
>
> This situation differs from the main track, where the main paper typically serves as the primary source of information. Instead, we have chosen to include more in-depth explanations and results in the appendix. Additionally, we believe that new users will benefit from our comprehensive documentation (https://rl4.co), which includes examples and detailed descriptions available at our documentation site. This resource offers clear instructions on how to use our library and is regularly updated, providing far more detail than what could be accommodated in the main paper.
>
> We also sincerely appreciate the reconsideration of your score to 6. We are eager to present our paper at this year's NeurIPS and believe that this opportunity will significantly enhance collaboration on our open-source RL4CO projects. We are confident that it will also contribute meaningfully to the AI for CO research community, e.g., in advancing foundation model development for the field of NCO.

---

> > ### Comment · Reviewer_hVj8 · 2024-09-01
> >
> > Thanks for providing this:
> >
> > "we kindly refer the reviewer to the existing results in our original submission. Specifically, in Table 10 (page 43) and Table 14 (page 48) of the appendix, we have compared our implemented algorithms with several OR solvers, including Gurobi, Concorde, HGS, LKH, and OR tools."
> >
> > Yes, the appendix is lengthy; and the reviewer tried to locate it in appendix;  Also, the reviewing guidance allows not reading appendix.
> >
> > In both page 43 and page 48, it is not clarified which results are from OR solvers, including Gurobi, Concorde, HGS, LKH, and OR tools."  Could you please provide say line numbers?  And what is the training dataset and what is the testing dataset (how they are generated?  are they generated from a graph distribution?)  Would you please also give hyperparameters used in the training process (learning rate, batch size; policy network; tensor data structure; sampling speed since it mentioned GPU accelerations using TorchRL and TensorDict, and other parameters, etc.)?
> >
> > Please keep in mind, this reviewer has been extensively working in both RL and AI4CO field and would be very happy to see that RL4CO helps the community grow.  NCO has a good number of publications recent years; it is challenging to build a comprehensive library. However, it does need more clarifications, as the authors would also agree with it when working out a lots of codes and docs.  One of your responsibility is to explain your efforts (also docs) to a wider community.

---

> > > ### Author Response · Authors · 2024-09-02
> > > **Final Clarifications**
> > >
> > > Thank you for the follow-up questions.
> > >
> > > Regarding the results from OR solvers:
> > > On page 43, specifically Table 10, we have OR solvers in the first 6 rows, under the heading "Classical Solvers," separated from the RL-based solvers by a horizontal line.
> > > - On page 48, specifically Table 14, we have Concorde as the first row, separated from the RL-based solvers with a horizontal line. Note that in Table 15, we do not report OR solvers since we only compare different algorithms at different temperatures $\tau$; the TSP 500 result from Concorde can be referred to in Table 14, i.e., 16.55.
> > >
> > >
> > > Regarding other experimental details i.e. for the main routing experiments:
> > > - The training and testing distributions are the same as described in Kool et al., 2019 [74], as described in line 601. We generate the test datasets with [this function](https://github.com/ai4co/rl4co/blob/193d7692155ef00b6a566a870a396c4cc7a93ec9/rl4co/data/generate_data.py#L324), i.e., we generate 10,000 samples for each validation set with seed 4321 and 10,000 samples for each test set with seed 1234, and they are generated from uniform distributions. Note that each problem has different generation details, i.e., locations for TSP and all problems as detailed at line 602, CVRP as per line 633, OP as in line 643, PDP in line 665.
> > > - Hyperparameters for these experiments are described in Section D.3.3 (lines 1364-1374) and also listed in Listing 1 at page 38. I.e., Learning rate of Adam of $10^{-4}$, batch size of 512 with a total of 1,280,000 samples per epoch for 100 epochs, with Attention policy network with 128 as embedding dimension, 512 of feedforward hidden, 8 attention heads, and 3 encoder layers.
> > > - The tensor data structure is based on TensorDict (available on [Github](https://github.com/pytorch/tensordict)). This allows us to do all the operation on GPU natively, and to conveniently reshape the state tensordict (called `td` through the environments) for several operations natively, i.e., batching the state without iterating explicitly through the keys as at line 19 in the [code](https://github.com/ai4co/rl4co/blob/main/rl4co/utils/ops.py#L19). The sampling speed is much faster than CPU-based operations, i.e., we can train in less than half the time compared to RLOR [116] since their environment is based on CPU parallelism only. We also made some improvements in sampling speed compared to TorchRL, i.e., one training epoch takes 284.9 seconds instead of 302.2 for one epoch in training TSP 200 (see Table 29) in TorchRL thanks to our more efficient memory usage explained in Appendix E.7.3. at page 63.
> > >
> > > We note that we have provided all the necessary details in the original submission, though we understand it might take some time for readers to locate them, which is fair given the extensive benchmark we're aiming to present. Additionally, we want to emphasize that settings such as training distributions, hyper-parameters choice, and comparison with OR solvers are standard practices within the NCO field. As an active researcher in both RL and AI4CO, we believe the reviewer could acknowledge that these settings are indeed common in recent NCO papers, as we have referenced in this paper. We should remind that the system will not allow us to respond any further since the author rebuttal is about to end. We will make every effort to further enrich the manuscript and documentation to clarify these details in response to your comments.

---

### Decision · Program_Chairs · 2024-09-26

**Decision:**

Reject

**Comment:**

The paper presents RL4CO, a comprehensive benchmark for reinforcement learning (RL) in combinatorial optimization (CO). It includes a wide range of CO problems, such as the Traveling Salesman Problem (TSP), Vehicle Routing Problem (VRP), and Job Shop Scheduling Problem (JSSP), and state-of-the-art RL methods, implemented in a modular and efficient manner using PyTorch and TorchRL.  The benchmark also features flexible configuration options and standardized evaluation protocols, enabling researchers to easily experiment with different combinations of algorithms, architectures, and environments.

The RL4CO benchmark addresses a crucial need in the NCO community by providing a unified and extensive framework for developing, evaluating, and comparing RL methods for CO problems. The benchmark's comprehensive coverage, modular design, and efficiency make it a valuable tool for researchers and practitioners. However, the paper's presentation could be improved by focusing more on technical contributions and providing clearer explanations of the benchmark's key features.

The review process highlighted some discrepancies in perspectives between the authors and reviewers. While the authors emphasized the benchmark's comprehensiveness and technical depth, some reviewers desired more explicit articulation of the key contributions and a clearer distinction from existing work. However, reviewers raised concerns about the limited scope and quality of the presentation of the paper.

The authors have addressed some of these concerns during the rebuttal, providing additional clarifications, examples, and insights into the benchmark's capabilities and potential impact. The authors have also demonstrated a willingness to further refine the paper's presentation and consider future extensions to incorporate other machine learning paradigms. I encourage the authors to continue engaging with the community and showcase the benchmark's capabilities, fostering collaboration and advancements in the field.

While the RL4CO benchmark is undoubtedly a valuable resource, the current submission does not fully meet the high bar for acceptance at NeurIPS at this point, maybe it could presented at a workshop focused on RL and CO.

Pros:
- Comprehensive: The benchmark covers a wide range of CO problems and RL methods, making it a valuable resource for researchers and practitioners. For instance, it includes 23 state-of-the-art RL methods and over 20 different CO problems across various domains.
- Modular and Flexible: The modular design allows for easy combination of different components, such as encoders, decoders, and feature embeddings, facilitating experimentation and customization. This enables researchers to build upon existing methods and develop novel solutions tailored to specific problems.
- Efficient: The implementation leverages hardware acceleration and optimized data structures, resulting in faster training and lower memory usage compared to similar libraries. This efficiency allows for experimentation with larger models and datasets, potentially leading to breakthroughs in solving complex CO problems.
- Well-documented: The benchmark is accompanied by extensive documentation and tutorials, making it accessible to both newcomers and experts in the field.
- Active Community: The benchmark has attracted a growing community of researchers and practitioners on Slack, fostering collaboration and knowledge sharing.

Cons:
- Presentation: The paper could benefit from a more focused presentation of its technical contributions, ensuring that key features and innovations are clearly highlighted and distinguished from existing work. Keeping the important technical contributions in the main paper and move the rest to appendix is key for clear communication.
- Limited Scope: The current focus is primarily on RL methods, potentially limiting its applicability to other machine learning paradigms, such as supervised or unsupervised learning, which could offer valuable insights for solving CO problems. However, the modular design allows for future extensions to incorporate these approaches.
- Figure Clarity: Some figures could be improved in terms of readability and clarity to enhance the overall presentation.